# Towards Nonlinear Disentanglement in Natural Data with Temporal Sparse Coding

**David Klindt**[*]
University of Tübingen
klindt.david@gmail.com

**Lukas Schott**[*]
University of Tübingen
lukas.schott@bethgelab.org

**Yash Sharma**[*]
University of Tübingen
yash.sharma@bethgelab.org

**Ivan Ustyuzhaninov**
University of Tübingen
ivan.ustyuzhaninov@bethgelab.org

**Wieland Brendel**
University of Tübingen
wieland.brendel@bethgelab.org

**Matthias Bethge**[‡]
University of Tübingen
matthias.bethge@bethgelab.org

**Dylan M Paiton**[‡]
University of Tübingen
dylan.paiton@bethgelab.org

## Abstract

Disentangling the underlying generative factors from data has so far been limited to carefully constructed scenarios. We propose a path towards natural data by first showing that the statistics of natural data provide enough structure to enable disentanglement, both theoretically and empirically. Specifically, we provide evidence that objects in natural movies undergo transitions that are typically small in magnitude with occasional large jumps, which is characteristic of a temporally sparse distribution. Leveraging this finding we provide a novel proof that relies on a sparse prior on temporally adjacent observations to recover the true latent variables up to permutations and sign flips, providing a stronger result than previous work. We show that equipping practical estimation methods with our prior often surpasses the current state-of-the-art on several established benchmark datasets without any impractical assumptions, such as knowledge of the number of changing generative factors. Furthermore, we contribute two new benchmarks, Natural Sprites and KITTI Masks, which integrate the measured natural dynamics to enable disentanglement evaluation with more realistic datasets. We test our theory on these benchmarks and demonstrate improved performance. We also identify non-obvious challenges for current methods in scaling to more natural domains. Taken together our work addresses key issues in disentanglement research for moving towards more natural settings.

## 1 Introduction

Natural scene understanding can be achieved by decomposing the signal into its underlying factors of variation. An intuitive approach for this problem assumes that a visual representation of the world can be constructed via a generative process that receives factors as input and produces natural signals as output (Bengio et al., 2013). This analogy is justified by the fact that our world is composed of distinct entities that can vary independently, but with regularity imposed by physics. What makes the approach appealing is that it formalizes representation learning by directly comparing representations to underlying ground-truth states, as opposed to the indirect evaluation of benchmarking against heuristic downstream tasks (e.g. object recognition). However, the core issue with this approach is *non-identifiability*, which means a set of possible solutions may all appear equally valid to the model, while only one identifies the true generative factors.

Our work is motivated by the question of whether the statistics of natural data will allow for the formulation of an identifiable model. Our core observation that enables us to make progress in

---

[*][‡]Equal contribution. Code: `https://github.com/bethgelab/slow_disentanglement`

addressing this question is that *generative factors of natural data have sparse transitions*. To estimate these generative factors, we compute statistics on measured transitions of area and position for object masks from large-scale, natural, unstructured videos. Specifically, we extracted over 300,000 object segmentation mask transitions from YouTube-VOS (Xu et al., 2018; Yang et al., 2019) and KITTI-MOTS (Voigtlaender et al., 2019; Geiger et al., 2012; Milan et al., 2016) (discussed in detail in Appendix D). We fit generalized Laplace distributions to the collected data (Eq. 2), which we indicate with orange lines in Fig. 1. We see empirically that all marginal distributions of temporal transitions are highly sparse and that there exist complex dependencies between natural factors (e.g. motion typically affects both position and apparent size). In this study, we focus on the sparse marginals, which we believe constitutes an important advance that sets the stage for solving further issues and eventually applying the technology to real-world problems. With this information at hand, we are able to provide a stronger proof for capturing the underlying generative factors of the data up to permutations and sign flips that is not covered by previous work (Hyvärinen and Morioka, 2016; 2017; Khemakhem et al., 2020a). Thus, we present the first work, to the best of our knowledge, which proposes a theoretically grounded solution that covers the statistics observed in real videos.

Our contributions are: With measurements from unstructured natural video annotations we provide evidence that natural generative factors undergo sparse changes across time. We provide a proof of identifiability that relies on the observed sparse innovations to identify nonlinearly mixed sources up to a permutation and sign-flips, which we then validate with practical estimation methods for empirical comparisons. We leverage the natural scene information to create novel datasets where the latent transitions between frames follow natural statistics. These datasets provide a benchmark to evaluate how well models can uncover the true latent generative factors in the presence of realistic dynamics. We demonstrate improved disentanglement over previous models on existing datasets and our

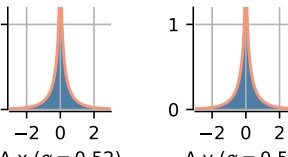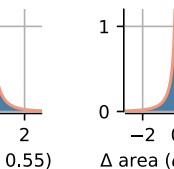

Figure 1: **Statistics of Natural Transitions**. The histograms show distributions over transitions of segmented object masks from natural videos for horizontal and vertical position as well as object size. The red lines indicate fits of generalized Laplace distributions (Eq. 2) with shape value $\alpha$. Data shown is for object masks extracted from YouTube videos. See Appendix G for 2D marginals and corresponding analysis from the KITTI self-driving car dataset.

contributed ones with quantitative metrics from both the disentanglement (Locatello et al., 2018) and the nonlinear ICA community (Hyvärinen and Morioka, 2016). We show via numerous visualization techniques that the learned representations for competing models have important differences, even when quantitative metrics suggest that they are performing equally well.

## 2  RELATED WORK – DISENTANGLEMENT AND NONLINEAR ICA

Disentangled representation learning has its roots in blind source separation (Cardoso, 1989; Jutten and Herault, 1991) and shares goals with fields such as inverse graphics (Kulkarni et al., 2015; Yildirim et al., 2020; Barron and Malik, 2012) and developing models of invariant neural computation (Hyvärinen and Hoyer, 2000; Wiskott and Sejnowski, 2002; Sohl-Dickstein et al., 2010) (see Bengio et al., 2013, for a review). A disentangled representation would be valuable for a wide variety of machine learning applications, including sample efficiency for downstream tasks (Locatello et al., 2018; Gao et al., 2019), fairness (Locatello et al., 2019; Creager et al., 2019) and interpretability (Bengio et al., 2013; Higgins et al., 2017; Adel et al., 2018). Since there is no agreed upon definition of disentanglement in the literature, we adopt two common measurable criteria: i) each encoding element represents a single generative factor and ii) the values of generative factors are trivially decodable from the encoding (Ridgeway and Mozer, 2018; Eastwood and Williams, 2018).

Uncovering the underlying factors of variation has been a long-standing goal in independent component analysis (ICA) (Comon, 1994; Bell and Sejnowski, 1995), which provides an identifiable solution for disentangling data mixed via an invertible linear generator receiving at most one Gaussian factor as input. Recent unsupervised approaches for nonlinear generators have largely been based on Variational Autoencoders (VAEs) (Kingma and Welling, 2013) and have assumed that the data is independent and identically distributed (*i.i.d.*) (Locatello et al., 2018), even though nonlinear methods that make this *i.i.d.* assumption have been proven to be *non-identifiable* (Hyvärinen and Pajunen,

1999; Locatello et al., 2018). Nonetheless, the bottom-up approach of starting with a nonlinear generator that produces well-controlled data has led to considerable achievements in understanding nonlinear disentanglement in VAEs (Higgins et al., 2017; Burgess et al., 2018; Rolinek et al., 2019; Chen et al., 2018), consolidating ideas from neural computation and machine learning (Khemakhem et al., 2020a), and seeking a principled definition of disentanglement (Ridgeway, 2016; Higgins et al., 2018; Eastwood and Williams, 2018).

Recently, Hyvärinen and colleagues (Hyvärinen and Morioka, 2016; 2017; Hyvärinen et al., 2018) showed that a solution to identifiable nonlinear ICA can be found by assuming that generative factors are conditioned on an additional observed variable, such as past states or the time index itself. This contribution was generalized by Khemakhem et al. (2020a) past the nonlinear ICA domain to any consistent parameter estimation method for deep latent-variable models, including the VAE framework. However, the theoretical assumptions underlying this branch of work do not account for the sparse transitions we observe in the statistics of natural scenes, which we discuss in further detail in appendix F.1.1. Another branch of work requires some form of supervision to demonstrate disentanglement (Szabó et al., 2017; Shu et al., 2019; Locatello et al., 2020). We select two of the above approaches, that are both different in their formulation and state-of-the-art in their respective empirical settings, Hyvärinen and Morioka (2017) and Locatello et al. (2020), for our experiments below. The motivation of our method and dataset contributions is to address the limitations of previous approaches and to enable unsupervised disentanglement learning in more naturalistic scenarios.[1]

The fact that physical processes bind generative factors in temporally adjacent natural video segments has been thoroughly explored for learning in neural networks (Hinton, 1990; Földiák, 1991; Mitchison, 1991; Wiskott and Sejnowski, 2002; Denton and Birodkar, 2017). We propose a method that uses time information in the form of an $L_1$-*sparse* temporal prior, which is motivated by the natural scene measurements presented above as well as by previous work (Simoncelli and Olshausen, 2001; Olshausen, 2003; Hyvärinen et al., 2003; Cadieu and Olshausen, 2012). Such a prior would intuitively allow for sharp changes in some latent factors, while most other factors remain unchanged between adjacent time-points. Almost all similar methods are variants of slow feature analysis (SFA, Wiskott and Sejnowski, 2002), which measure slowness in terms of the Euclidean (i.e. $L_2$, or log Gaussian) distance between temporally adjacent encodings. Related to our approach, a probabilistic interpretation of SFA has been previously proposed (Turner and Sahani, 2007), as well as extensions to variational inference (Grathwohl and Wilson, 2016). Additionally, Hashimoto (2003) suggested that a sparse (Cauchy) slowness prior improves correspondence to biological complex cells over the $L_2$ slowness prior in a two-layer model. However, to the best of our knowledge, an $L_1$ temporal prior has previously only been used in deep auto-encoder frameworks when applied to semi-supervised tasks (Mobahi et al., 2009; Zou et al., 2012), and was mentioned in Cadieu and Olshausen (2012), who used an $L_2$ prior, but claimed that an $L_1$ prior performed similarly on their task. Similar to Hyvärinen et al. (Hyvärinen and Morioka, 2016; Hyvärinen et al., 2018), we only assume that the latent factors are temporally dependent, thus avoiding assuming knowledge of the number of factors where the two observations differ (Shu et al., 2019; Locatello et al., 2020).

Most of the standard datasets for disentanglement (dSprites (Matthey et al., 2017), Cars3D (Reed et al., 2015), SmallNORB (LeCun et al., 2004), Shapes3D (Kim and Mnih, 2018), MPI3D (Gondal et al., 2019)) have been compiled into a disentanglement library (DisLib) by Locatello et al. (2018). However, all of the DisLib datasets are limited in that the data generating process is independent and identically distributed (*i.i.d.*) and all generative factors are assumed to be discrete. In a follow-up study, Locatello et al. (2020) proposed combining pairs of images such that only $k$ factors change, as this matches their modeling assumptions required to prove identifiability. Here, $k \in \mathcal{U}\{1, D-1\}$ and $D$ denotes the number of ground-truth factors, which are then sampled uniformly. We additionally use the measurements from Fig. 1 to construct datasets for evaluating disentanglement that have time transitions which directly correspond to natural dynamics.

---

[1]As in slow feature analysis, we consider learning from videos without labels as *unsupervised*.

## 3 THEORY

### 3.1 GENERATIVE MODEL

We have provided evidence to support the hypothesis that generative factors of natural videos have sparse temporal transitions (see Fig. 1). To model this process, we assume temporally adjacent input pairs $(\mathbf{x}_{t-1}, \mathbf{x}_t)$ coming from a nonlinear generator that maps factors to images $\mathbf{x} = g(\mathbf{z})$, where generative factors are dependent over time:

$$p(\mathbf{z}_t, \mathbf{z}_{t-1}) = p(\mathbf{z}_t|\mathbf{z}_{t-1})p(\mathbf{z}_{t-1}). \qquad (1)$$

Assume the observed data $(\mathbf{x}_t, \mathbf{x}_{t-1})$ comes from the following generative process, where different latent factors are assumed to be independent (cf. Appendix F.2):

$$\mathbf{x} = g(\mathbf{z}), \quad p(\mathbf{z}_{t-1}) = \prod_{i=1}^{d} p(z_{t-1,i}), \quad p(\mathbf{z}_t|\mathbf{z}_{t-1}) = \prod_{i=1}^{d} \frac{\alpha\lambda}{2\Gamma(1/\alpha)} \exp -(\lambda|z_{t,i} - z_{t-1,i}|^\alpha), \quad (2)$$

where $\lambda$ is the distribution rate, $p(\mathbf{z}_{t-1})$ is a factorized Gaussian prior $\mathcal{N}(\mathbf{0}, \mathbf{I})$ (as in Kingma and Welling, 2013) and $p(\mathbf{z}_t|\mathbf{z}_{t-1})$ is a factorized generalized Laplace distribution (Subbotin, 1923) with shape parameter $\alpha$, which determines the shape and especially the kurtosis of the function.[2] Intuitively, smaller $\alpha$ implies larger kurtosis and sparser temporal transitions of the generative factors (special cases are Gaussian, $\alpha = 2$, and Laplacian, $\alpha = 1$). Critically, for our proof we assume $\alpha < 2$ to ensure that temporal transitions are sparse. The novelty of our approach lies in our explicit modeling of sparse transitions that cover the statistics of natural data, which results in a stronger identifiability proof than previously achieved (see Appendix F.1.1 for a more detailed comparison with Hyvärinen and Morioka, 2017; Khemakhem et al., 2020a).

### 3.2 IDENTIFIABILITY PROOF

**Theorem 1** *For a ground-truth $(g^*, \lambda^*, \alpha^*)$ and a learned $(g, \lambda, \alpha)$ model as defined in Eq. (2), if the functions $g^*$ and $g$ are injective and differentiable almost everywhere, $\lambda^* = \lambda$, $\alpha^* = \alpha < 2$ (i.e. there is no model misspecification) and the distributions of pairs of images generated from the priors $\mathbf{z}^* \sim p^*(\mathbf{z})$ and $\mathbf{z} \sim p(\mathbf{z})$ generated as $(g^*(\mathbf{z}_{t-1}^*), g^*(\mathbf{z}_t^*))$ and $(g(\mathbf{z}_{t-1}), g(\mathbf{z}_t))$, respectively, are matched almost everywhere, then $g = g^* \circ \sigma$, where $\sigma$ is composed of a permutation and sign flips.*

The formal proof is provided in Appendix A.1. Similar to linear ICA, but in the temporal domain, we have to assume that the transitions of generative factors across time be non-Gaussian. Specifically, if the temporal changes of ground-truth factors are sparse, then the only generator consistent with the observations is the ground-truth one (up to a permutation and sign flips). The main idea behind the proof is to represent $g$ as $g^* \circ h$ and note that if $h$ were not a permutation, then the distributions $((g^* \circ h)(\mathbf{z}_{t-1}), (g^* \circ h)(\mathbf{z}_t))$ and $(g^*(\mathbf{z}_{t-1}^*), g^*(\mathbf{z}_t^*))$ would not match, due to the injectivity of $g^*$. Whether or not these distributions are the same is equivalent to whether or not the distributions of pairs $(z_{t-1}, z_t)$ and $(h(z_{t-1}), h(z_t))$ are the same. For these distributions to be the same, the function $h$ must preserve the Gaussian marginal for the first time step as well as the joint distribution, implying that it must preserve both the vector lengths and distances in the latent space. As we argue in the extended proof, this can only be the case if $h$ is a composition of permutations and sign flips.

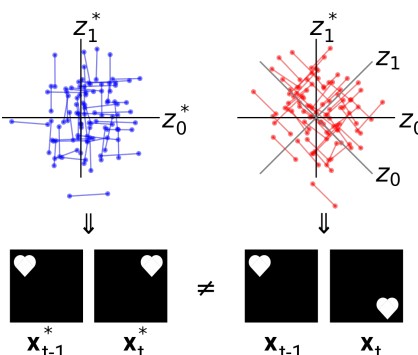

Figure 2: **Proof Intuition**. Latent representation and example generated image pairs for ground-truth (blue) and entangled (red) model. See text below for details.

**Intuition** Fig. 2 illustrates, by contradiction, why the model defined in Eq. (2) is identifiable. We consider temporal pairs of latents represented by connected points. A sparse transition prior encourages axis-alignment, as can be seen from the Laplace transition prior in the third image of Fig. 3.

---

[2]For a stationary stochastic process, $p(\mathbf{z}_{t-1})$ represents the instantaneous marginal distribution and $p(\mathbf{z}_t|\mathbf{z}_{t-1})$ the transition distribution. In case of an autoregressive process with non-Gaussian innovations with finite variance, it follows from the central limit theorem that the marginal distribution converges to a Gaussian in the limit of large $\lambda$.

This results in lines that are parallel with the axes in both the ground truth (left, blue, $\mathbf{z}^*$) and learned model (right, red, $\mathbf{z}$). In this example, $z_0^*$ corresponds to horizontal position, while $z_1^*$ corresponds to vertical position. The learned model must satisfy two criteria: (1) the latent factors should match the sparse prior (axis-aligned) and (2) the generated image pairs should match the ground-truth image pairs. If the learned latent factors were mismatched, for example by rotation, then the image pair distributions would not be matched. In this example, the ground truth model would produce image pairs with typically vertical or horizontal transitions, while the learned model pairs result in mostly diagonal transitions. Thus, the learned model cannot satisfy both criteria without aligning the latent axes with the ground-truth axes.

### 3.3 SLOW VARIATIONAL AUTOENCODER

In order to validate our proof, we must choose a probabilistic latent variable model for estimating the data density. We chose to build upon the framework of VAEs because of their efficiency in estimating a variational approximation to the ground truth posterior of a deep latent variable model (Kingma and Welling, 2013). We will refer to this model as *SlowVAE*. In Appendix B we note shortcomings of such an approach and test an alternative flow-based model.

The standard VAE objective assumes *i.i.d.* data and a standard normal prior with diagonal covariance on the learned latent representations $\mathbf{z} \sim \mathcal{N}(\mathbf{0}, \mathbf{I})$. To extend this to sequences, we assume the same functional form for our model prior as in Eq. (1) and Eq. (2). The posterior of our model is independent across time steps. Specifically,

$$q(\mathbf{z}_t, \mathbf{z}_{t-1}|\mathbf{x}_t, \mathbf{x}_{t-1}) = q(\mathbf{z}_t|\mathbf{x}_t)\, q(\mathbf{z}_{t-1}|\mathbf{x}_{t-1}), \quad q(\mathbf{z}|\mathbf{x}) = \prod_{i=1}^{d} \mathcal{N}(\mu_i(\mathbf{x}), \sigma_i^2(\mathbf{x})), \tag{3}$$

where $\mu_i(\mathbf{x})$ and $\sigma_i^2(\mathbf{x})$ are the input-dependent mean and variance of our model's posterior. We visualize this combination of priors and posteriors in Fig. 3. For a given pair of inputs $(\mathbf{x}_t, \mathbf{x}_{t-1})$, the full evidence lower bound (ELBO, which we derive in Appendix A.2) can be written as

$$\mathcal{L}(\mathbf{x}_t, \mathbf{x}_{t-1}) = E_{q(\mathbf{z}_t, \mathbf{z}_{t-1}|\mathbf{x}_t, \mathbf{x}_{t-1})}[\log p(\mathbf{x}_t, \mathbf{x}_{t-1}|\mathbf{z}_t, \mathbf{z}_{t-1})] - D_{KL}(q(\mathbf{z}_{t-1}|\mathbf{x}_{t-1})|p(\mathbf{z}_{t-1}))$$
$$- \gamma\, E_{q(\mathbf{z}_{t-1}|\mathbf{x}_{t-1})}[D_{KL}(q(\mathbf{z}_t|\mathbf{x}_t)|p(\mathbf{z}_t|\mathbf{z}_{t-1}))], \tag{4}$$

where $\gamma$ is a regularization term for the sparsity prior, analogous to $\beta$ in $\beta$-VAEs (Higgins et al., 2017) (technically, Eq. 4 is only an ELBO with $\gamma \leq 1$). The first term on the right-hand side is the log-likelihood (i.e. the negative reconstruction error, with $p(\mathbf{x}_t, \mathbf{x}_{t-1}|\mathbf{z}_t, \mathbf{z}_{t-1})$ parameterized by the decoder of the VAE), the second term is the KL to a normal prior as in the standard VAE and the last term is an expectation of the KL between the posterior at time step $t$ and the conditional prior $p(\mathbf{z}_t|\mathbf{z}_{t-1})$. The expectation in the last term is taken over samples from the posterior at the previous time step $q(\mathbf{z}_{t-1}|\mathbf{x}_{t-1})$. We observed empirically that taking the mean, $\mu(\mathbf{x}_{t-1})$, as a single sample produces good results, analogous to the log-likelihood that is typically evaluated at a single sample from the posterior (see Blei et al. (2017) for context).

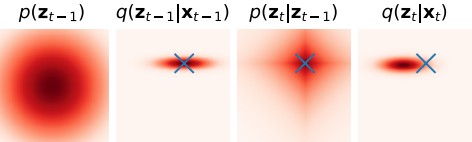

In practice, we need to choose $\alpha$, $\lambda$, and $\gamma$. For the latter two, we can perform a random search for hyper-parameters, as we discuss below. For the former, any $\alpha < 2$ would break the general rotation symmetry by having an optimum for axis-aligned representations, which theorem 1 includes as a requirement for identifiability. As can be seen in Figs. 1 and 11, $\alpha \approx 0.5$ provides the best fit to the ground-truth marginals. However, we used $\alpha = 1$ as a parsimonious choice for SlowVAE, since the Laplace is a well-understood distribution that allows us to derive a simple closed-form solution for the ELBO in Eq. 4, which we derive in Appendix A.2.

Figure 3: **SlowVAE illustration.** The prior and posterior for a two-dimensional latent space. Left to right: Normal prior for $t - 1$, posterior for $t - 1$, conditional Laplace prior for $t$, and posterior for $t$. The blue cross in the right three plots indicates the mean of the posterior for $t - 1$.

### 3.4 TOWARDS AN APPROXIMATE THEORY OF DISENTANGLEMENT

A number of our theoretical assumptions are violated in practice: After non-convex optimization, on a finite data sample, the distributions $p(\mathbf{x}_t, \mathbf{x}_{t-1})$ and $p^*(\mathbf{x}_t, \mathbf{x}_{t-1})$ are probably not perfectly

matched. In addition, the model assumptions on $p(\mathbf{z}_t, \mathbf{z}_{t-1})$ likely do not fully match the distribution of the ground truth factors. For example, the model may be misspecified such that $\alpha \neq \alpha^*$ or $\lambda \neq \lambda^*$, or the chosen family of distributions may be incorrect altogether. In the following section we will present results on several datasets where the marginal distributions $p(\mathbf{z}_{t-1})$ are drawn from a Uniform (not Normal) distribution, and some of them are over unordered sets (categories) or bounded periodic spaces (rotation). Also, in practice the model latent space is usually chosen to have more dimensions than the ground truth generative model. On real data, factors of variation may be dependent (Träuble et al., 2020; Yang et al., 2020). We show this is the case on YouTube-VOS and KITTI-MOTS in Appendix G and we provide evidence that breaking these dependencies has no clear consequence on disentanglement in Appendix F.2. A more formal treatment of dependence is done by Khemakhem et al. (2020b) who relax the independence assumption of ICA to Independently Modulated Components Analysis (IMCA) and introduce a family of conditional energy-based models that are identifiable up to simple transformations. Furthermore, the hypothesis class $\mathcal{G}$ of learnable functions in the VAE architecture may not contain the invertible ground truth generator $g^* \notin \mathcal{G}$, if it exists at all (e.g. occlusions may already lead to non-invertibility). Despite these violations, we consider it a strength of our method that the practical implementation still achieves improved disentanglement over previous approaches. However, we note understanding the impact of these violations as an important focus area for continued progress towards developing a practical yet theoretically supported method for disentanglement on natural scenes.

## 4  DATASETS WITH NATURAL TRANSITIONS

While the standard datasets compiled by DisLib are an important step towards real-world applications, they still assume the data is *i.i.d.*. As described in section 2, Locatello et al. (2020) proposed uniformly sampling the number of factors to be changed, $k = \text{Rnd}$, and changing said factors by uniformly sampling over the possible set of values. What we refer to as "UNI" is a dataset variant modeled after the described scheme (Locatello et al., 2020) (further details in Appendix D). Considering our natural data analysis presented in Figure 1, such transitions are certainly unnatural. Given the current state of evaluation, we provide a set of incrementally more natural datasets which are otherwise comparable to existing work. We propose that said datasets should be included in the standard benchmark suite to provide a step towards disentanglement in natural data.

**(1) Laplace Transitions** (LAP) is a procedure for constructing image pairs from DisLib datasets by sampling from a sparse conditional distribution. For each ground-truth factor, the first value in the pair is chosen *i.i.d.* from the dataset and the second is chosen by weighting nearby factor values using Laplace distributed probabilities. LAP is a step towards natural data that closely resembles previous extensions of DisLib datasets to the time domain, but in a way that matches the marginal distribution of natural transitions (see Appendix D.2 for more details).
**(2) Natural Sprites** consists of pairs of rendered sprite images with generative factors sampled from real YouTube-VOS transitions. For a given image pair, the position and scale of the sprites are set using measured values from adjacent time points in YouTube-VOS. The sprite shapes and orientations are simple, like dSprites, and are fixed for a given pair. While fixing shape follows the natural transitions of objects, it is unclear how to accurately estimate object orientation from the masks, and thus we fixed the factor to avoid introducing artificial transitions. We additionally consider a version that is discretized to the same number of object states as dSprites, which i) allows us to use the standard DisLib evaluation metrics and ii) helps isolate the effect of including natural transitions from the effect of increasing data complexity (see Appendix D.4 for more details).
**(3) KITTI Masks** is composed of pedestrian segmentation masks from the autonomous driving vision benchmark KITTI-MOTS, thus with natural shapes and continuous natural transitions in all underlying factors. We consider adjacent frames which correspond to mean$(\Delta t) = 0.05s$ in physical time (we report the mean because of variable sampling rates in the original data); as well as frames with a larger temporal gap of mean$(\Delta t) = 0.15s$, which corresponds to samples of pairs that are at most 5 frames apart. We show in Appendix G.3 that SlowVAE disentanglement performance increases and then plateaus as we continue to increase mean$(\Delta t)$.

In summary, we construct datasets with (1) imposed sparse transitions, (2) augmented with natural continuous generative factors using measurements from unstructured natural videos, as well as (3) data from unstructured natural videos themselves, but provided as segmentation masks to ensure visual complexity is manageable for current methods. For the provided datasets, the object categories

| Model | Data | BetaVAE | FactorVAE | MIG | MCC | DCI | Modularity | SAP |
|---|---|---|---|---|---|---|---|---|
| PCL | dSprites (Uniform) | 80.1 (0.4) | 62.1 (0.9) | 16.0 (7.4) | 41.6 (1.5) | 42.4 (1.2) | **99.7** (0.6) | 6.0 (2.7) |
| Ada-GVAE | dSprites (Uniform) | 88.0 (2.7) | 73.1 (3.9) | 17.3 (4.7) | 46.0 (4.8) | 32.3 (4.6) | 93.3 (1.8) | 6.6 (2.0) |
| SlowVAE | dSprites (Uniform) | 87.0 (5.1) | 75.2 (11.1) | **28.3** (11.5) | **58.8** (8.9) | 47.7 (8.5) | 86.9 (2.8) | 4.4 (2.0) |
| PCL | dSprites (Laplace) | 99.9 (0.1) | 94.7 (3.1) | 19.2 (3.1) | 67.9 (3.3) | 52.0 (3.5) | 93.2 (0.9) | 8.1 (1.6) |
| Ada-GVAE | dSprites (Laplace) | 91.4 (1.6) | 83.0 (5.9) | 21.8 (4.9) | 56.9 (4.2) | 39.0 (4.2) | 87.6 (1.8) | 7.2 (0.3) |
| SlowVAE | dSprites (Laplace) | 100.0 (0.0) | 97.5 (3.0) | **29.5** (9.3) | 69.8 (2.3) | **65.4** (3.6) | **96.5** (1.6) | 8.1 (3.0) |
| PCL | Natural (Discrete) | 82.4 (6.7) | 68.3 (8.0) | 7.8 (2.8) | 50.2 (4.2) | 14.3 (3.0) | 88.9 (3.1) | 2.5 (1.1) |
| Ada-GVAE | Natural (Discrete) | 83.4 (1.1) | 74.8 (4.4) | 14.5 (3.2) | 51.6 (2.5) | 21.8 (2.9) | 87.8 (2.5) | 5.3 (1.4) |
| SlowVAE | Natural (Discrete) | 82.6 (2.2) | 76.2 (4.8) | 11.7 (5.0) | 52.6 (4.1) | 18.9 (5.5) | 88.1 (3.6) | 4.4 (2.3) |

Table 1: Mean and standard deviation (s.d.) metric scores across 10 random seeds. PCL is a scaled-up implementation of the method described by Hyvärinen and Morioka (2017), leveraging the encoding architecture and training hyperparameters specified in appendix E. Ada-GVAE is the leading method proposed by Locatello et al. (2020). Bold indicates statistical significance above the next highest score (independent T-test, $p < 0.05$). Red indicates statistical significance below the next lowest score. Results for additional datasets and models are in Table 2 and Appendix G.

never change across transitions – reflecting natural object permanence. Finally, as (2) and (3) use factor transitions measured from natural videos, they exhibit any natural statistical structure present for those factors, such as natural dependencies (further discussion is in Appendix F.2).

## 5 EXPERIMENTS

### 5.1 EMPIRICAL STUDIES

We evaluate models using the DisLib implementation for the following supervised metrics: BetaVAE (Higgins et al., 2017); FactorVAE (Kim and Mnih, 2018); Mutual Information Gap (MIG; Chen et al., 2018); Disentanglement, Compactness, and Informativeness (DCI / Disentanglement; Eastwood and Williams, 2018); Modularity (Ridgeway and Mozer, 2018); and Separated Attribute Predictability (SAP; Kumar et al., 2018) (see Appendix C for metric details). None of the DisLib metrics support ground-truth labels with continuous variation, which is required for evaluation on the continuous Natural Sprites and KITTI Masks datasets. To reconcile this, we measure the Mean Correlation Coefficient (MCC), a standard metric in the ICA literature that is applicable to continuous variables. We report mean and standard deviation across 10 random seeds.

In order to select the conditional prior regularization and the prior rate in an unsupervised manner, we perform a random search over $\gamma \in [1, 16]$ and $\lambda \in [1, 10]$ and compute the recently proposed unsupervised disentanglement ranking (UDR) scores (Duan et al., 2020). We notice that the optimal values are close to $\gamma = 10$ and $\lambda = 6$ on most datasets, and thus use these values for all experiments. We leave finding optimal values for specific datasets to future work, but note that it is a strong advantage of our approach that it works well with the same model specification across 13 datasets (counting LAP and UNI for DisLib and optional discretization for Natural Sprites), addressing a concern posed in (Locatello et al., 2018). Additional details on model selection and training can be found in Appendix E. Although we train on image pairs, our model does not need paired data points at test time. For all visualizations, we pick the models with the highest average score across the DisLib metrics.

To compare our model fairly against other methods that also take image pairs as inputs, we also present performance for Permutation-Contrastive Learning from nonlinear ICA (PCL, Hyvärinen and Morioka, 2017) and Ada-GVAE, the leading method in the study by (Locatello et al., 2020). We scaled up the implementation of PCL for evaluation on our high-dimensional pixel inputs, and note this method does not have any hyperparameters. For Ada-GVAE, following the paper's recommendations, we select $\beta$ (per dataset) using the considered parameter set $[1, 2, 4, 6, 8, 16]$, and use the reconstruction loss as the unsupervised model selection criterion (Locatello et al., 2020).

Figure 4: **KITTI Masks** (mean($\Delta t$) = 0.15$s$). (Left) MCC correlation matrix of the top 3 latents corresponding to y-position, x-position and scale. (Right) Images produced by varying the SlowVAE latent unit that corresponds to the corresponding row in the MCC matrix.

## 5.2 RESULTS ON DISLIB AND NEW BENCHMARKS

In Table 1 we demonstrate favorable performance compared to PCL and Ada-GVAE across all applicable metrics for discrete ground-truth variable datasets. The relative improvement on UNI is particularly surprising given the drastic mismatch between UNI and SlowVAE's assumptions. In Appendix G, we report results for the remaining DisLib datasets, where the observed dSprites results largely transfer. We also outperform PCL with a (flow-based) exact likelihood implementation of our slow transition prior in Appendix F.1.1. In Appendix F.3, we show that a model with an $L_2$ transition ($\alpha = 2$) prior performs much worse, supporting our theoretical prediction.

On the KITTI Masks dataset, one source of variation in the data is the average temporal separation within pairs of images mean($\Delta t$). We present two settings (mean($\Delta t$) = 0.05$s$, mean($\Delta t$) = 0.15$s$) and observe a comparative increase in MCC for the latter (Table 2). Namely, the increase in performance for larger time gap is more pronounced with SlowVAE than the baselines, resulting in a statistically significant MCC gain. We provide details on the settings and ablate over the mean($\Delta t$) parameter in Appendix G.3, where we observe a positive trend between mean($\Delta t$) and MCC (reflecting Table 2, in Oord et al., 2018). Finally, we also verify that the

| Model | Data | MCC |
|---|---|---|
| PCL | Natural (Continuous) | 51.7 (3.0) |
| Ada-GVAE | Natural (Continuous) | 48.4 (4.8) |
| SlowVAE | Natural (Continuous) | 49.1 (4.0) |
| PCL | Kitti (mean($\Delta t$) = 0.05$s$) | 52.6 (5.1) |
| Ada-GVAE | Kitti (mean($\Delta t$) = 0.05$s$) | 62.6 (7.5) |
| SlowVAE | Kitti (mean($\Delta t$) = 0.05$s$) | 66.1 (4.5) |
| PCL | Kitti (mean($\Delta t$) = 0.15$s$) | 58.5 (3.3) |
| Ada-GVAE | Kitti (mean($\Delta t$) = 0.15$s$) | 67.6 (6.7) |
| SlowVAE | Kitti (mean($\Delta t$) = 0.15$s$) | **79.6** (5.8) |

Table 2: Continuous ground-truth variable datasets. See Table 1 for details.

transition distributions remain sparse despite the increase in this parameter (Appendix G.3). In Fig. 4, we can see that SlowVAE has learned latent dimensions which have correspondence with the estimated ground truth factors of x/y-position and scale.

Locatello et al. (2018) showed that all *i.i.d.* models performed similarly across the DisLib datasets and metrics when testing was carefully controlled. However, in Fig. 5 we observe that the different modeling assumptions result in differences in representation quality. To construct the visuals, we first compute the sorted correlation matrix between the latents (rows) and generative factors (columns), which we visualize as a correlation matrices. The matrices are sorted via linear sum assignment such that each ground-truth factor is non-greedily associated with the latent variable with highest correlation (Hyvärinen and Morioka, 2016). Below the matrices are scatter plots that reveal the decodability of the assigned latent factors. In each scatter plot, the horizontal axis indicates the ground truth value, the vertical axis indicates the corresponding latent value, and the colors indicate object shape. The models displayed are those with the maximum average score across evaluated metrics.

The latent space visualizations use the known ground-truth factors to aid in understanding how each factor is encoded in a way that is more informative than exclusively visualizing latent traversals or embeddings of pairs of latent units (Cheung et al., 2014; Chen et al., 2016; Szabó et al., 2017; Ma et al., 2018). For example, in the third row, we observe that several models have a sinusoidal variation with frequencies $\sim \omega, 2\omega$, and $4\omega$, which correspond to the three distinct rotational symmetries of the shapes: heart, ellipse and square. This directly impacts MCC performance (third row in the MCC matrix), which measures rank correlation between the matching latent factor (an angular variable) and the ground truth, which encodes the angles with monotonically increasing indices. Furthermore, the square has a four-fold rotational symmetry and repeats after $90°$, but it is represented in a full $360°$ rotation in the DisLib ground truth encoding format, resulting in different ground truth labels for identical input images.

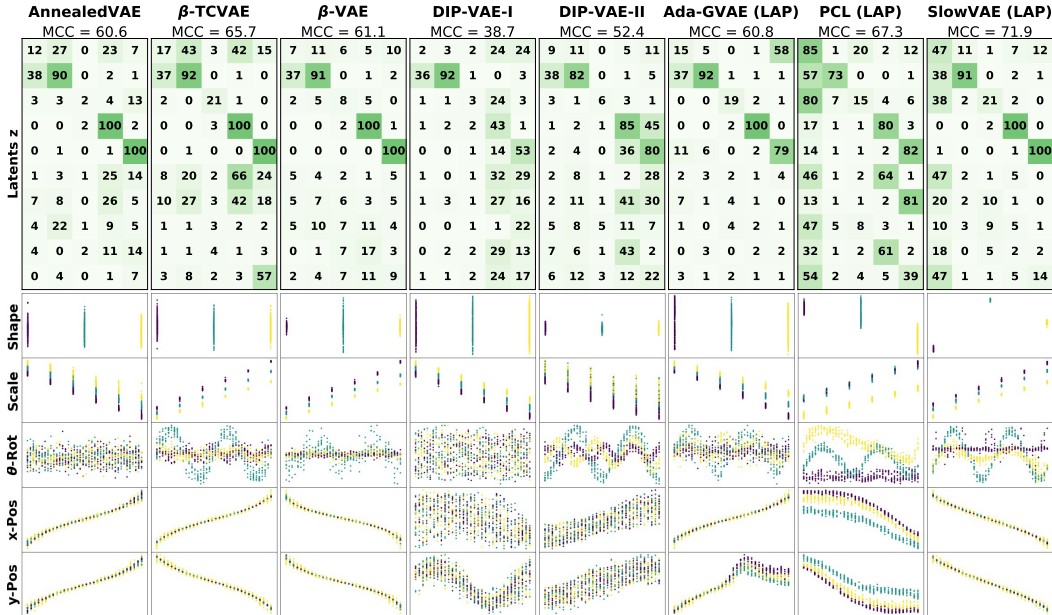

Figure 5: **DSprites Latent Representations:** (Top) shows absolute MCC between generative and model factors (rows are rearranged for maximal correlation on the main diagonal). The columns correspond to generative factors (shape, scale, rotation, x/y-position) and the values correspond to percent correlation. A more diagonal structure in the upper half corresponds to a better one-to-one mapping between generative and latent factors. (Bottom) shows individual latent dimensions (y-axis) over the matched generative factors (x-axis). Colors encode shapes: heart/yellow, ellipse/turquoise, and square/purple.

A similar observation can be made with respect to the categorical factors, which are also represented as ordinal ground truth variables. For example, the PCL correlation score (top left element in the PCL MCC matrix) is quite high, while the corresponding shape correlation score for SlowVAE is quite low. However, if we consider the shape scatter plots, we clearly see that SlowVAE separates the three shapes more distinctively than PCL, only in an order that differs from the ground truth. One solution is to modify MCC to report the maximum correlation over all permutations of the ground truth assignments, although brute force methods for this would scale poorly with the number of categories. We also note that datasets where we see small performance differences among models (e.g., Cars3D) have significantly more discrete categories (e.g., 183) than the other datasets ($3 - 6$). This could also explain why all models considered in Table 1 and 2 perform comparably on the Natural Sprites datasets, where unlike KITTI Masks the ground truth evaluation includes categorical and angular variables. We note that properly evaluating disentanglement is an ongoing area of research (Duan et al., 2020), with notable preliminary results in recent work (Higgins et al., 2018; Bouchacourt et al., 2021; Tonnaer et al., 2020).

## 6 CONCLUSION

We provide evidence to support the hypothesis that natural scenes exhibit highly sparse marginal transition probabilities. Leveraging this finding, we contribute a novel nonlinear ICA framework that is provably identifiable up to permutations and sign-flips — a stronger result than has been achieved previously. With the SlowVAE model we provide a parsimonious implementation that is inspired by a long history of learning visual representations from temporal data (Sutton, 1988; Hinton, 1990; Földiák, 1991). We apply this model to current metric-based disentanglement benchmarks to demonstrate that it outperforms existing approaches (Locatello et al., 2020; Hyvärinen and Morioka, 2017) on aggregate without any tuning of hyperparameters to individual datasets. We also provide novel video dataset benchmarks to guide disentanglement research towards more natural domains.

We observe that these datasets have complex dependencies that our theory will have to be extended to account for, although we demonstrate with empirical comparisons the efficacy of our approach. In addition to Natural Sprites and KITTI Masks, we suggest that YouTube-VOS will be valuable as a large-scale dataset that is unconstrained by object type and scenario for more advanced models. Variance in such categorical factors is problematic for evaluation due to the cited drawbacks of existing quantitative metrics, which should be addressed in tandem with scaling to natural data. Taken together, our dataset and model proposals set the stage for utilizing knowledge of natural scene statistics to advance unsupervised disentangled representation learning.

In our experiments we see that approximate identification as measured by the different disentanglement metrics increases despite violations of theoretical assumptions, which is in line with prior studies (Shu et al., 2019; Khemakhem et al., 2020a; Locatello et al., 2020). Nevertheless, future work should address gaining a better understanding of the theoretical and empirical consequences of such model misspecifications, in order to make the theory of disentanglement more predictive about empirically found solutions.

## ACKNOWLEDGEMENTS

The authors would like to thank Francesco Locatello for valuable discussions and providing numerical results to facilitate our experimental comparisons. Additionally, we thank Luigi Gresele, Matthias Tangemann, Roland Zimmermann, Robert Geirhos, Matthias Kümmerer, Cornelius Schröder, Charles Frye, and Sarah Master for helpful feedback in preparing the manuscript. Finally, the authors would like to thank Johannes Ballé, Jon Shlens and Eero Simoncelli for early discussions related to the ideas developed in this paper.

This work was supported by the Deutsche Forschungsgemeinschaft (DFG) in the priority program 1835 under grant BR2321/5-2 and by SFB 1233, Robust Vision: Inference Principles and Neural Mechanisms (TP3), project number: 276693517. We thank the International Max Planck Research School for Intelligent Systems (IMPRS-IS) for supporting LS and YS. DP was supported by the German Federal Ministry of Education and Research (BMBF) through the Tübingen AI Center (FKZ: 01IS18039A). IU, WB, and MB are supported by the Intelligence Advanced Research Projects Activity (IARPA) via Department of Interior/Interior Business Center (DoI/IBC) contract number D16PC00003. The U.S. Government is authorized to reproduce and distribute reprints for Governmental purposes notwithstanding any copyright annotation thereon. Disclaimer: The views and conclusions contained herein are those of the authors and should not be interpreted as necessarily representing the official policies or endorsements, either expressed or implied, of IARPA, DoI/IBC, or the U.S. Government.

The authors declare no conflicts of interests.

## BROADER IMPACT

Representation learning is at the heart of model building for cognition. Our specific contribution is focused on core methods for modeling natural videos and the datasets used are more simplistic than real-world examples. However, foundational research on unsupervised representation learning has potentially large impact on AI for advancing the power of self-learning systems.

The broader field of representation learning has a large number of focused research directions that span machine learning and computational neuroscience. As such, the application space for this work is vast. For example, applications in unsupervised analysis of complicated and unintuitive data, such as medical imaging and gene expression information, have great potential to solve fundamental problems in health sciences. A future iteration of our disentangling approach could be used to encode such complicated data into a lower-dimensional and more understandable space that might reveal important factors of variation to medical researchers. Another important and complex modeling space that could potentially be improved by this line of research is in environmental sciences and combating global climate change.

Nonetheless, we acknowledge that any machine learning method can be used for nefarious purposes, which can be mitigated via effective, scientifically informed communication, outreach, and policy direction. We unconditionally denounce the use of derivatives of our work for weaponized or wartime applications. Additionally, due to the lack of interpretability generally found in modern deep learning approaches, it is possible for practitioners to inadvertently introduce harmful biases or errors in machine learning applications. Although we certainly do not solve this problem, our focus on providing identifiable solutions to representation learning is likely beneficial for both interpretability and fairness in machine learning.

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

APPENDIX

## A  FORMAL METHODS

| Function / variable | Description |
| --- | --- |
| $g$ | Generator |
| $\alpha$ | Prior shape |
| $\lambda$ | Prior rate |
| $p(\mathbf{z})$ | Prior |
| $\mathbf{z} \sim p(\mathbf{z})$ | Latent variables |
| $\mathbf{x} = g(\mathbf{z})$ | Generated images |
| $q(\mathbf{z}|\mathbf{x})$ | Variational posterior |

Table 3: Glossary of terms. We use a $^*$ (i.e. $g^*$) when necessary to highlight that we are referring to the ground truth model.

### A.1  PROOF OF IDENTIFIABILITY

To study disentanglement, we assume that the generative factors $\mathbf{z} \in \mathbb{R}^D$ are mapped to images $\mathbf{x} \in \mathbb{R}^N$ (usually $D \ll N$, but see section B) by a nonlinear ground-truth generator $g^* : \mathbf{z} \mapsto \mathbf{x}$.

**Theorem 1** *Let $(g^*, \lambda^*, \alpha^*)$ and $(g, \lambda, \alpha)$ respectively be ground-truth and learned generative models as defined in Eq. (2). If the following conditions are satisfied:*

  *(i) The generators $g^*$ and $g$ are defined everywhere in the latent space. Moreover, they are injective and differentiable almost everywhere,*

  *(ii) There is no model misspecification i.e. $\alpha = \alpha^*$ and $\lambda = \lambda^*$, so $\mathbf{z} \sim p(\mathbf{z}) = p^*(\mathbf{z})$,*

  *(iii) Pairs of images are generated as $(\mathbf{x}^*_{t-1}, \mathbf{x}^*_t) = (g^*(\mathbf{z}_{t-1}), g^*(\mathbf{z}_t))$ and $(\mathbf{x}_{t-1}, \mathbf{x}_t) = (g(\mathbf{z}_{t-1}), g(\mathbf{z}_t))$,*

  *(iv) The distributions of $(\mathbf{x}^*_{t-1}, \mathbf{x}^*_t)$ and $(\mathbf{x}_{t-1}, \mathbf{x}_t)$ are the same (i.e. the corresponding densities are equal almost everywhere: $p^*(\mathbf{x}_{t-1}, \mathbf{x}_t) = p(\mathbf{x}_{t-1}, \mathbf{x}_t)$,*

*then $g = g^* \circ \sigma$, where $\sigma$ is a composition of a permutation and sign flips.*

*Proof.* Since $\mathbf{x} = g(\mathbf{z})$ can be written as $\mathbf{x} = (g^* \circ (g^*)^{-1} \circ g)(\mathbf{z})$, we can assume that $g = g^* \circ h$ for some function $h$ on the latent space.

We first show that the function $h$ is a bijection on the latent space. It is injective, since both $g$ and $g^*$ are injective. Because of continuity of $h$, if it were not surjective, there would be some neighborhood $\mathbf{U}_{\tilde{\mathbf{z}}}$ of $\tilde{\mathbf{z}}$ that would not have a pre-image under $h$. This would mean that images generated by $g^*$ from $\mathbf{U}_{\tilde{\mathbf{z}}}$ would have zero density under the distribution of images generated by $g$ (i.e. $p(g^*(\mathbf{U}_{\tilde{\mathbf{z}}})) = 0$). This density would be non-zero under the distribution of images directly generated by the ground-truth generator $g^*$ (i.e. $p^*(g^*(\mathbf{U}_{\tilde{\mathbf{z}}})) \neq 0$), which contradicts the assumption that these distributions are equal. It follows that $h$ is bijective.

In the next step, we show that the distribution of latent space pairs $(h(\mathbf{z}_{t-1}), h(\mathbf{z}_t))$ matches the latent space prior distribution (i.e. $h$ preserves the prior distribution in the latent space). Indeed, using the assumption that the distributions of $(g^*(\mathbf{z}_{t-1}), g^*(\mathbf{z}_t))$ and $((g^* \circ h)(\mathbf{z}_{t-1}), (g^* \circ h)(\mathbf{z}_t))$ are the same, we can write the following equality using the change of variables formula:

$$
\begin{aligned}
p^*(\mathbf{x}_{t-1}, \mathbf{x}_t) &= p((g^*)^{-1}(\mathbf{x}_{t-1}), (g^*)^{-1}(\mathbf{x}_t)) \left| \det \left( \frac{\mathrm{d}(g^*)^{-1}}{\mathrm{d}(\mathbf{x}_{t-1}, \mathbf{x}_t)} \right) \right| \\
&= p_h((g^*)^{-1}(\mathbf{x}_{t-1}), (g^*)^{-1}(\mathbf{x}_t)) \left| \det \left( \frac{\mathrm{d}(g^*)^{-1}}{\mathrm{d}(\mathbf{x}_{t-1}, \mathbf{x}_t)} \right) \right| \\
&= p(\mathbf{x}_{t-1}, \mathbf{x}_t),
\end{aligned}
\tag{5}
$$

where $p$ and $p_h$ are densities of $(\mathbf{z}_{t-1}, \mathbf{z}_t)$ and $(h(\mathbf{z}_{t-1}), h(\mathbf{z}_t))$. Since the determinants above cancel, these densities are equal at the pre-image of any pair of images $(\mathbf{x}_{t-1}, \mathbf{x}_t)$. Because $g^*$ is defined

everywhere in the latent space, $p$ and $p_h$ are equal for any pair of latent space points. Applying the change of variables formula again, we obtain the following equation:

$$
\begin{aligned}
p(\mathbf{z}_{t-1}, \mathbf{z}_t) &= p(h^{-1}(\mathbf{z}_{t-1}), h^{-1}(\mathbf{z}_t)) \left| \det \left( \frac{\mathrm{d}h^{-1}}{\mathrm{d}(\mathbf{z}_{t-1}, \mathbf{z}_t)} \right) \right| \\
&= p(h^{-1}(\mathbf{z}_{t-1}))\, p(h^{-1}(\mathbf{z}_t) \mid h^{-1}(\mathbf{z}_{t-1})) \left| \det \left( \frac{\mathrm{d}h^{-1}(\mathbf{z}_{t-1})}{\mathrm{d}\mathbf{z}_{t-1}} \right) \right| \left| \det \left( \frac{\mathrm{d}h^{-1}(\mathbf{z}_t)}{\mathrm{d}\mathbf{z}_t} \right) \right| \\
&= p(\mathbf{z}_{t-1})\, p(\mathbf{z}_t \mid \mathbf{z}_{t-1}).
\end{aligned}
\tag{6}
$$

Note that the probability measure $p$ is the same before and after the change of variables, since we showed that the prior distribution in the latent space must be invariant under the function $h$. The same condition for the marginal $p(\mathbf{z}_{t-1})$ is as follows:

$$
p(\mathbf{z}_{t-1}) = p(h^{-1}(\mathbf{z}_{t-1})) \left| \det \left( \frac{\mathrm{d}h^{-1}(\mathbf{z}_{t-1})}{\mathrm{d}\mathbf{z}_{t-1}} \right) \right|.
\tag{7}
$$

Solving for the determinant of the Jacobian in (7) and plugging it into (6), we obtain

$$
p(\mathbf{z}_t \mid \mathbf{z}_{t-1}) = p(h^{-1}(\mathbf{z}_t) \mid h^{-1}(\mathbf{z}_{t-1})) \frac{p(\mathbf{z}_t)}{p(h^{-1}(\mathbf{z}_t))}.
\tag{8}
$$

Taking logs of both sides, we arrive at the following equation:

$$
A(||\mathbf{z}_t - \mathbf{z}_{t-1}||_\alpha^\alpha - ||h^{-1}(\mathbf{z}_t) - h^{-1}(\mathbf{z}_{t-1})||_\alpha^\alpha) = B(||\mathbf{z}_t||_2^2 - ||h^{-1}(\mathbf{z}_t)||_2^2),
\tag{9}
$$

where $A$ and $B$ are the constants appearing in the exponentials in $p(\mathbf{z}_{t-1})$ and $p(\mathbf{z}_t \mid \mathbf{z}_{t-1})$. The logs of normalization constants cancel out.

For any $\mathbf{z}_t$ we can choose $\mathbf{z}_{t-1} = \mathbf{z}_t$ making the left hand side in (9) equal to zero. This implies that $||\mathbf{z}_t||_2^2 = ||h^{-1}(\mathbf{z}_t)||_2^2$ for any $\mathbf{z}_t$, i.e. function $h^{-1}$ preserves the 2-norm. Moreover, the preservation of the 2-norm implies that $p(\mathbf{z}_{t-1}) = p(h^{-1}(\mathbf{z}_{t-1}))$ and therefore it follows from (7) that for any $\mathbf{z}$

$$
\left| \det \left( \frac{\mathrm{d}h^{-1}(\mathbf{z})}{\mathrm{d}\mathbf{z}} \right) \right| = 1.
\tag{10}
$$

Thus, the left hand side of (9) can be re-written as

$$
||\mathbf{z}_t - \mathbf{z}_{t-1}||_\alpha^\alpha - ||h^{-1}(\mathbf{z}_t) - h^{-1}(\mathbf{z}_{t-1})||_\alpha^\alpha = 0.
\tag{11}
$$

This means that $h^{-1}$ preserves the $\alpha$-distances between points. Moreover, because $h$ is bijective, the Mazur-Ulam theorem (Mazur and Ulam, 1932) tells us that $h$ must be an affine transform.

In the next step, to prove that $h$ must be a permutation and sign flip, let us choose an arbitrary point $\mathbf{z}_{t-1}$ and $\mathbf{z}_t = \mathbf{z}_{t-1} + \varepsilon\, \mathbf{e}_k = (z_{1,1}, \ldots, z_{1,k} + \varepsilon, \ldots, z_{1,D})$. Using (11) and performing a Taylor expansion around $\mathbf{z}_{t-1}$, we obtain the following:

$$
\begin{aligned}
\varepsilon^\alpha &= ||\mathbf{z}_t - \mathbf{z}_{t-1}||_\alpha^\alpha \\
&= ||h^{-1}(\mathbf{z}_{t-1} + \varepsilon\, \mathbf{e}_k) - h^{-1}(\mathbf{z}_{t-1})||_\alpha^\alpha \\
&= \left|\left| \varepsilon \cdot \left( \frac{\partial h_1^{-1}(\mathbf{z}_{t-1})}{\partial z_{t-1,k}}, \ldots, \frac{\partial h_D^{-1}(\mathbf{z}_{t-1})}{\partial z_{t-1,k}} \right) + O(\varepsilon^2) \right|\right|_\alpha^\alpha.
\end{aligned}
\tag{12}
$$

The higher-order terms $O(\varepsilon^2)$ are zero since $h$ is affine, therefore dividing both sides of the above equation by $\varepsilon^\alpha$ we find that

$$
\left|\left| \left( \frac{\partial h_1^{-1}(\mathbf{z}_{t-1})}{\partial z_{t-1,k}}, \ldots, \frac{\partial h_D^{-1}(\mathbf{z}_{t-1})}{\partial z_{t-1,k}} \right) \right|\right|_\alpha^\alpha = 1.
\tag{13}
$$

The vectors of $k$-th partial derivatives of components of $h^{-1}$ are columns of the Jacobian matrix $\left( \frac{\mathrm{d}h^{-1}(\mathbf{z})}{\mathrm{d}\mathbf{z}} \right)$. Using the fact that the determinant of that matrix is equal to one and applying Hadamard's inequality, we obtain that

$$
\left| \det \left( \frac{\mathrm{d}h^{-1}(\mathbf{z})}{\mathrm{d}\mathbf{z}} \right) \right| = 1 \le \prod_{k=1}^D \left|\left| \left( \frac{\partial h_1^{-1}(\mathbf{z}_{t-1})}{\partial z_{t-1,k}}, \ldots, \frac{\partial h_D^{-1}(\mathbf{z}_{t-1})}{\partial z_{t-1,k}} \right) \right|\right|_2.
\tag{14}
$$

Since $\alpha < 2$, for any vector $\mathbf{v}$ it holds that $||\mathbf{v}||_2 \leq ||\mathbf{v}||_\alpha$, with equality only if at most one component of $\mathbf{v}$ is non-zero. This inequality implies that both (13) and (14) hold at the same time if and only if

$$\left\|\left(\frac{\partial h_1^{-1}(\mathbf{z}_{t-1})}{\partial z_{t-1,k}}, \ldots, \frac{\partial h_D^{-1}(\mathbf{z}_{t-1})}{\partial z_{t-1,k}}\right)\right\|_2 = \left\|\left(\frac{\partial h_1^{-1}(\mathbf{z}_{t-1})}{\partial z_{t-1,k}}, \ldots, \frac{\partial h_D^{-1}(\mathbf{z}_{t-1})}{\partial z_{t-1,k}}\right)\right\|_\alpha = 1, \quad (15)$$

meaning that only one element of these vectors of $k$-th partial derivatives is non-zero, and it is equal to 1 or -1. Thus, the function $h$ is a composition of a permutation and sign flips at every point. Potentially, this permutation might be input-dependent, but we argued above that $h$ is affine, therefore the permutation must be the same for all points. □

## A.2  Kullback Leibler Divergence of Slow Variational Autoencoder

The VAE learns a variational approximation to the true posterior by maximizing a lower bound on the log-likelihood of the empirical data distribution $\mathcal{D}$

$$\begin{aligned} E_{\mathbf{x}_{t-1},\mathbf{x}_t\sim\mathcal{D}}[\log p(\mathbf{x}_{t-1},\mathbf{x}_t)] \geq \\ E_{\mathbf{x}_{t-1},\mathbf{x}_t\sim\mathcal{D}}[E_{q(\mathbf{z}_t,\mathbf{z}_{t-1}|\mathbf{x}_t,\mathbf{x}_{t-1})}[\log p(\mathbf{x}_{t-1},\mathbf{x}_t,\mathbf{z}_{t-1},\mathbf{z}_t) - \log q(\mathbf{z}_t,\mathbf{z}_{t-1}|\mathbf{x}_t,\mathbf{x}_{t-1})]]. \end{aligned} \quad (16)$$

For this, we need to compute the Kullback-Leibler divergence (KL) between the posterior $q(\mathbf{z}_t,\mathbf{z}_{t-1}|\mathbf{x}_t,\mathbf{x}_{t-1})$ and the prior $p(\mathbf{z}_t,\mathbf{z}_{t-1})$. Since all of these distributions are per design factorial, we will, for simplicity, derive the KL below for scalar variables (log-probabilities will simply have to be summed to obtain the full expression). Recall that the model prior and posterior factorize like

$$\begin{aligned} p(z_t,z_{t-1}) &= p(z_t|z_{t-1})\,p(z_{t-1}) \\ q(z_t,z_{t-1}|\mathbf{x}_t,\mathbf{x}_{t-1}) &= q(z_t|\mathbf{x}_t)\,q(z_{t-1}|\mathbf{x}_{t-1}). \end{aligned} \quad (17)$$

Then, given a pair of inputs $(\mathbf{x}_{t-1},\mathbf{x}_t)$, the KL can be written

$$\begin{aligned} D_{KL}(q(z_t,z_{t-1}|\mathbf{x}_t,\mathbf{x}_{t-1})|p(z_t,z_{t-1})) &= E_{z_t,z_{t-1}\sim q(z_t,z_{t-1}|\mathbf{x}_t,\mathbf{x}_{t-1})}\left[\log\frac{q(z_t|\mathbf{x}_t)\,q(z_{t-1}|\mathbf{x}_{t-1})}{p(z_t|z_{t-1})\,p(z_{t-1})}\right] \\ &= E_{z_{t-1}\sim q(z_{t-1}|\mathbf{x}_{t-1})}\left[\log\frac{q(z_{t-1}|\mathbf{x}_{t-1})}{p(z_{t-1})}\right] + E_{z_t,z_{t-1}\sim q(z_t,z_{t-1}|\mathbf{x}_t,\mathbf{x}_{t-1})}\left[\log\frac{q(z_t|\mathbf{x}_t)}{p(z_t|z_{t-1})}\right] \\ &= D_{KL}(q(z_{t-1}|\mathbf{x}_{t-1})|p(z_{t-1})) - H(q(z_t|\mathbf{x}_t)) + E_{z_{t-1}\sim q(z_{t-1}|\mathbf{x}_{t-1})}\left[H(q(z_t|\mathbf{x}_t),p(z_t|z_{t-1}))\right] \end{aligned} \quad (18)$$

Where we use the fact that KL divergences decompose like $D_{KL}(X,Y) = H(X,Y) - H(X)$ into (differential) cross-entropy $H(X,Y)$ and entropy $H(X)$. The first term of the last line in (18) is the same KL divergence as in the standard VAE, namely between a Gaussian distribution $q(z_{t-1}|\mathbf{x}_{t-1})$ with some $\mu(\mathbf{x}_{t-1})$ and $\sigma(\mathbf{x}_{t-1})$ and a standard Normal distribution $p(z_{t-1})$. The solution of the KL is given by $D_{KL}(q(z_{t-1}|\mathbf{x}_{t-1})|q(z_{t-1})) = -\log\sigma(\mathbf{x}_{t-1}) + \frac{1}{2}(\mu(\mathbf{x}_{t-1})^2 + \sigma(\mathbf{x}_{t-1})^2 - 1)$ (Bishop, 2006). The second term on the RHS, i.e. the entropy of a Gaussian is simply given by $H(q(z_t|\mathbf{x}_t)) = \log(\sigma(\mathbf{x}_t)\sqrt{2\pi e})$.

To compute the last term on the RHS, let us recall the Laplace form of the conditional prior

$$p(z_t|z_{t-1}) = \frac{\lambda}{2}\exp{-\lambda|z_t - z_{t-1}|}. \quad (19)$$

Thus the cross-entropy becomes

$$\begin{aligned} H(q(z_t|\mathbf{x}_t),p(z_t|z_{t-1})) &= -E_{z_t\sim q(z_t|\mathbf{x}_t)}[\log p(z_t|z_{t-1})] \\ &= -\log\left(\frac{\lambda}{2}\right) + \lambda E_{z_t\sim q(z_t|\mathbf{x}_t)}[|z_t - z_{t-1}|]. \end{aligned} \quad (20)$$

Now, if some random variable $X \sim \mathcal{N}(\mu, \sigma^2)$, then $Y = |X|$ follows a *folded normal distribution*, for which the mean is defined as

$$E[|x|] = \sigma \sqrt{\frac{2}{\pi}} \exp\left(-\frac{\mu^2}{2\sigma^2}\right) - \mu\left(1 - 2\,\Phi\left(\frac{\mu}{\sigma}\right)\right), \tag{21}$$

where $\Phi$ is the cumulative distribution function of a standard normal distribution (mean zero and variance one). Thus, denoting $\mu(\mathbf{x}_t)$ and $\sigma(\mathbf{x}_t)$ the mean and variance of $q(z_t|\mathbf{x}_t)$, and defining $\mu(\mathbf{x}_t, z_{t-1}) = \mu(\mathbf{x}_t) - z_{t-1}$, we can rewrite further

$$H(q(z_t|\mathbf{x}_t), p(z_t|z_{t-1})) =$$

$$-\log\left(\frac{\lambda}{2}\right) + \lambda\left(\sigma(\mathbf{x}_t)\sqrt{\frac{2}{\pi}} \exp\left(-\frac{\mu(\mathbf{x}_t, z_{t-1})^2}{2\sigma(\mathbf{x}_t)^2}\right) - \mu(\mathbf{x}_t, z_{t-1})\left(1 - 2\,\Phi\left(\frac{\mu(\mathbf{x}_t, z_{t-1})}{\sigma(\mathbf{x}_t)}\right)\right)\right). \tag{22}$$

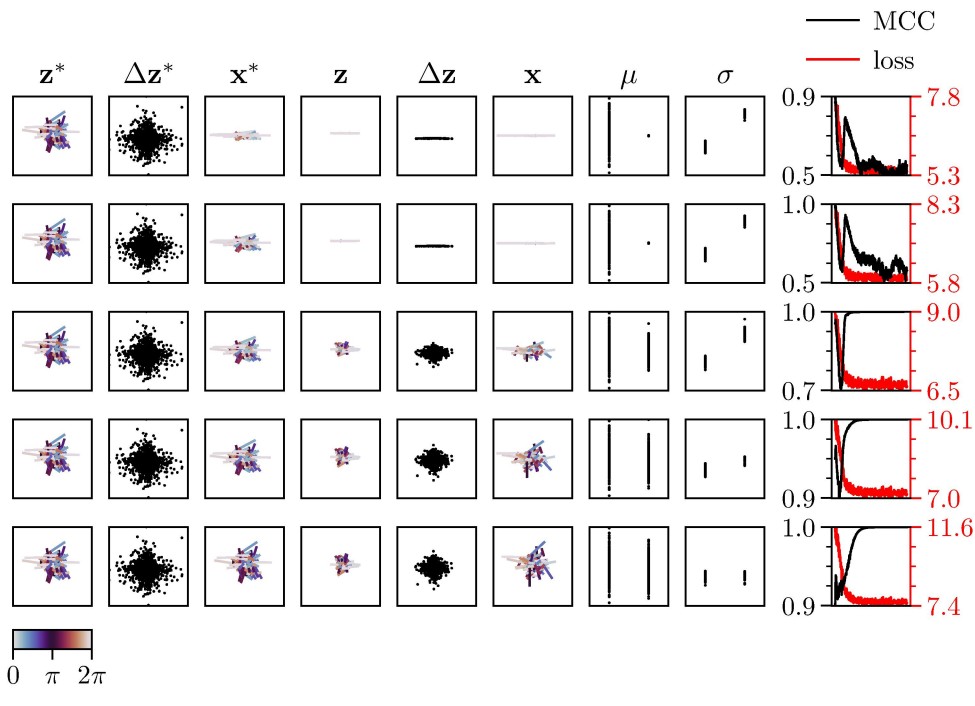

(a) SlowVAE performance.

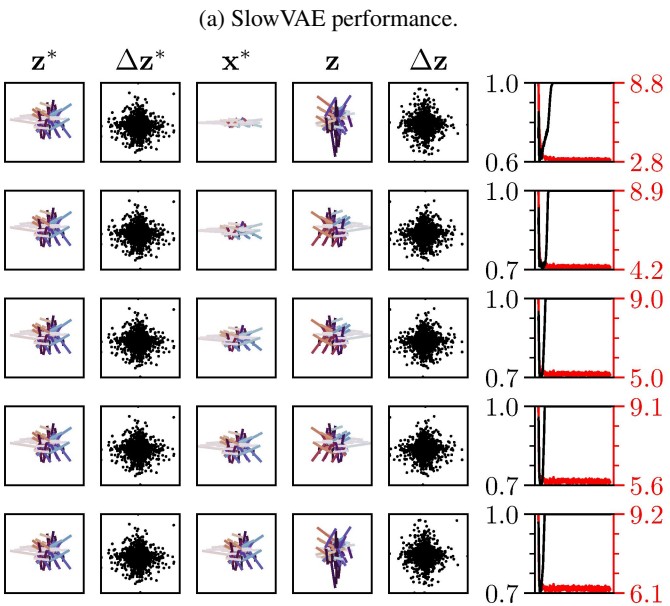

(b) SlowFlow performance.

Figure 6: **VAE failure modes.** Rows respectively indicate $\kappa = 0.2, 0.4, 0.6, 0.8, 1.0$ from Eq. (24). The left five columns show values for 100 randomly chosen examples, while the $\mu$ and $\sigma$ columns show values for the full training set. Columns in the sets ($\mathbf{z}^*$, $\mathbf{z}$), ($\Delta\mathbf{z}^*$, $\Delta\mathbf{z}$), ($\mathbf{x}^*$, $\mathbf{x}$) all have the same (arbitrary) scale factors the axes. Lines indicate trajectories from time-point $t$ to $t + 1$, and color indicates the angle of the trajectory vector with respect to the canonical variable axes. The $\mu$ axes is scaled from $-4$ to $4$, and $\sigma$ axes are scaled from $0$ to $1$, where individual dots represent latent encoding values from test images. The rightmost plots show a shift in the relationship between the mean correlation coefficient (MCC) (black, higher is better) and training loss (red, lower is better) as one increases $\kappa$.

## B   CHOOSING A LATENT VARIABLE MODEL

Our proposed method for disentanglement can be implemented in conjunction with different probabilistic latent variable models. In this section, we compare VAEs and normalizing flows as possible candidates.

Variational Autoencoders (VAEs) (Kingma and Welling, 2013) are a widely used probabilistic latent variable model. Despite their simple structure and empirical success, VAEs can converge to a pathological solution called *posterior collapse* (Lucas et al., 2019; Bowman et al., 2016; He et al., 2019). This solution results in the encoder's variational posterior approximation matching the prior, which is typically chosen to be a multivariate standard normal $q(\mathbf{z}|\mathbf{x}) \approx p(\mathbf{z}) = \mathcal{N}(\mathbf{0}, \mathbf{I})$. This disconnects the encoder from the decoder, making them approximately independent, i.e. $p(\mathbf{x}|\mathbf{z}) \approx p(\mathbf{x})$. The failure mode is often observed when the decoder architecture is overly expressive, i.e. with autoregressive models, or when the likelihood $p(\mathbf{x})$ is easy to estimate. Approaches that alleviate this problem rely on modifying the ELBO training objective (Bowman et al., 2016; Kingma et al., 2016) or restricting the decoder structure (Dieng et al., 2019; Maaløe et al., 2019). However, these approaches come with various drawbacks, including optimization issues (Lucas et al., 2019).

Another approach to estimate latent variables are normalizing flows which describe a sequence of invertible mappings by iteratively applying the change of variables rule (Dinh et al., 2017b). Unlike VAEs, flow based latent variable models allow for a direct optimization of the likelihood (Dinh et al., 2017b). Most normalizing flow models rely on a fast and reliable calculation of the determinant of the Jacobian of the outputs with respect to the inputs, which constrains the architectural design and limits the capacity of the network (Tabak et al., 2010; Tabak and Turner, 2013; Dinh et al., 2017b). Thus, competitive flows require very deep architectures in practice (Kingma and Dhariwal, 2018). Furthermore, flows are not directly suited for a scenario where the observation space is higher dimensional than the generating latent factors, $\dim(\mathbf{z}) < \dim(\mathbf{x})$, as the computation of the determinant requires a square Jacobian matrix. We tried setting $\dim(\mathbf{z}) = \dim(\mathbf{x}) > \dim(\mathbf{z}^*)$, but observed instability while optimizing the objective defined below.

It is straightforward to derive a flow-based objective based on the assumptions in Eq. (2). We consider a normalizing flow with with $K$ blocks $f(\mathbf{x}) = f_K \circ ... \circ f_1 : \mathbf{x} \mapsto \mathbf{z}$. The coupling blocks can refer to nonlinear mixing similar to Kingma and Dhariwal (2018), or in the linear case ($K = 1$) to an invertible de-mixing matrix. This leads to the following estimation of the likelihood

$$p(\mathbf{x}_{t-1}, \mathbf{x}_t) = p(f(\mathbf{x}_{t-1})) \, p(f(\mathbf{x}_t)|f(\mathbf{x}_{t-1})) \prod_{k=1}^{K} \left| \det \frac{\partial f_k}{\partial \mathbf{z}_{k-1,t-1}} \right|^{-1} \prod_{k=1}^{K} \left| \det \frac{\partial f_k}{\partial \mathbf{z}_{k-1,t}} \right|^{-1}.$$

(23)

Note that $p(f(\mathbf{x}_{t-1}))$ is Gaussian and $p(f(\mathbf{x}_t)|f(\mathbf{x}_{t-1}))$ is a Laplacian, similar to Eq. (2). During optimization we take the $-\log$ of both sides and minimize w.r.t. the parameters of $f$. We refer to this estimator as *SlowFlow*. Our SlowFlow model is very similar to the flow described in (Pineau et al., 2020), who use a Gaussian transition prior and therefore would have weaker identifiability guarantees. Next, we compare SlowFlow and SlowVAE in the context of disentanglement.

To demonstrate the posterior collapse in VAEs, we generate data points $(\mathbf{x}_t, \mathbf{x}_{t-1})$ according to Eq. (2) with a two dimensional latent space $\dim(\mathbf{z}^*) = 2$. We consider a trivial linear mixing of $\mathbf{x}^* = \mathbf{W}^*\mathbf{z}^* = g^*(\mathbf{z}^*)$ with

$$\mathbf{W}^* = \mathrm{diag}(1, \kappa) \tag{24}$$

and $\kappa \in [0.1, 1]$. As can be seen by looking at the $\sigma$ and $\mu$ outputs of the encoder in Fig 6a, for $\kappa < 0.4$, the encoder for the minor axis collapses to the prior. The decoder then tries to minimize the reconstruction loss by solely covering the first principal component of the data, which is also described in Rolinek et al. (2019). Despite the collapse and decrease in MCC, the SlowVAE loss from Eq. (4) still improves during training. On the other hand, a simple linear SlowFlow model $f(\mathbf{x}) = \mathbf{W}\mathbf{x}$, which directly optimizes the likelihood, recovers the latents consistently as seen by the MCC measure (Fig 6b).

To show the strength of the VAE model we increase the complexity of the data-distribution by using a non-linear expanding decoder such that $\dim(\mathbf{x}) \gg \dim(\mathbf{z}^*)$. In Fig. 7 we observe that increasing the input dimensionality is sufficient for SlowVAE to find the corresponding latents and achieve high MCC with low loss.

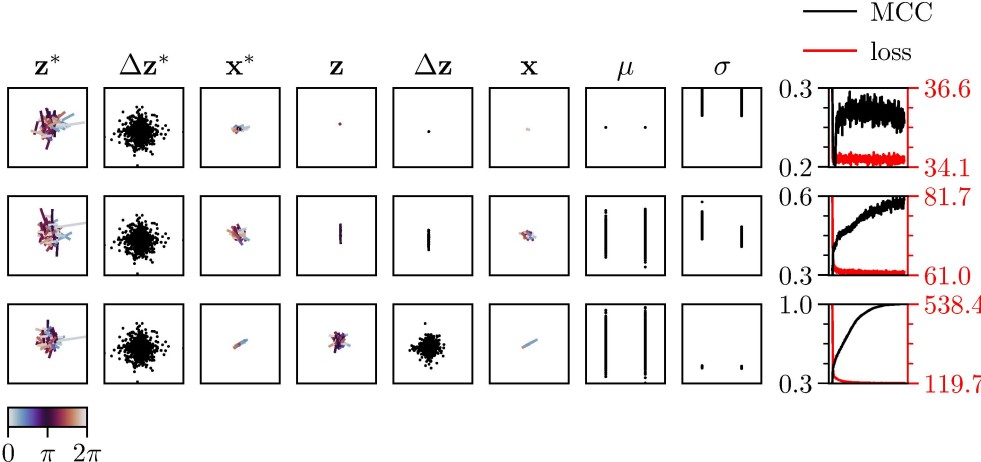

Figure 7: **VAEs perform better when data dimensionality exceeds the latent dimensionality.** VAEs prefer data dimensions to be greater than latent dimensions. Individual subplots are as described in Fig. 6. For all data in this experiment we used a 20-dimensional latent space, $\dim(\mathbf{z}^*) = 20$. Each row corresponds to the dimensionality of the $\mathbf{x}^*$, with values of 20, 200, and 2000. The first two dimensions of $\mathbf{z}^*$ are plotted as well as the two dimensions of $\mathbf{z}$ with the highest corresponding mean correlation coefficient (MCC). The $\mathbf{x}^*$ and $\mathbf{x}$ data are projected onto their first two principal component axes before plotting. A two-layer mixing matrix was used to transform data from $Z_{\text{gt}}$ to $X_{\text{gt}}$. As one increases the data dimensionality, the SlowVAE network performs increasingly better in terms of MCC, although worse in terms of total training loss.

Each estimation method is practically useful in different experimental settings. In the case when the mixing operation is trivially defined (Eq. (24), or when the number of dimensions in $\mathbf{z}^*$ match those in $\mathbf{x}^*$), the VAE estimator tends to learn a pathological solution. On the other hand, the normalizing flow estimator does not scale well to high dimensional data due to the requirement of computing the network Jacobian. Additionally, the framework for constructing normalizing flow estimators assumes the latent dimensionality is equal to the data dimensionality to allow for an invertible transform. Together these results lead us to choose an estimator based on the nature of the problem. For our contributed datasets and the DisLib experiments we adopt the VAE framework. However, if one aims to perform simplified experiments such as those typically conducted in the nonlinear ICA literature, it will often make practical sense to switch to a flow-based estimator.

## C  DISENTANGLEMENT METRICS

Several recent studies have brought to light shortcomings in a number of proposed disentanglement metrics (Kim and Mnih, 2018; Eastwood and Williams, 2018; Chen et al., 2018; Higgins et al., 2018; Mathieu et al., 2019), many of which have been compiled in the DisLib benchmark. In addition to the concerns they raise, it is important to note that none of the supervised metrics implemented in DisLib allow for continuous ground-truth factors, which is necessary for evaluating with the Natural Sprites and KITTI Masks datasets, as factors such as position and scale are effectively continuous in reality. To rectify this issue without introducing novel metrics, we include the Mean Correlation Coefficient (MCC) in our evaluations, using the implementation of Hyvärinen and Morioka (2016), which is described below.

We measure all metrics presented below between $10,000$ samples of latent factors $\mathbf{z}$ and the corresponding encoded means of our model $\mu(g^*(\mathbf{z}))$. We increase this sample size to $100,000$ for Modularity and MIG to stabilize the entropy estimates.

## C.1 Mean Correlation Coefficient

In addition to the DisLib metrics, we also compute the Mean Correlation Coefficient (MCC) in order to perform quantitative evaluation with continuous variables. Because of Theorem 1, perfect disentanglement in the noiseless case should always lead to a correlation coefficient of 1 or $-1$, although note that we report 100 times the absolute value of the correlation coefficient. In our experiments, MCC is used without modification from the authors' open-sourced code (Morioka, 2018). The method first measures correlation between the ground-truth factors and the encoded latent variables. The initial correlation matrix is then used to match each latent unit with a preferred ground-truth factor. This is an assignment problem that can be solved in polynomial time via the Munkres algorithm, as described in the code release from Morioka (2018). After solving the assignment problem, the correlation coefficients are computed again for the vector of ground-truth factors and the resulting permuted vector of latent encodings, where the output is a matrix of correlation coefficients with $D$ columns for each ground-truth factor and $D'$ rows for each latent variable. We use the (absolute value of the) Spearman coefficient as our correlation measure which assumes a monotonic relationship between the ground-truth factors and latent encodings but tolerates deviations from a strictly linear correspondence.

In the existing implementation for MCC, the ground truth factors, latent encodings, and mixed signal inputs are assumed to have the same dimensionality, i.e. $D = D' = N$. However, in our case, the ground-truth generating factors are much lower dimensional than the signal, $N \ll D$, and the latent encoding is higher dimensional than the ground-truth factors $D' > D$ (see Appendix E for details). To resolve this discrepancy, we add $D' - D$ standard Gaussian noise channels to the ground-truth factors. To compute the MCC score, we take the mean of the absolute value of the upper diagonal of the correlation matrix. The upper diagonal is the diagonal of the square matrix of $D$ ground-truth factors by the top $D$ most correlated latent dimensions after sorting. In this way, we obtain an MCC estimate which averages only over the $D$ correlation coefficients of the $D$ ground truth factors with their corresponding best matching latent factors.

## C.2 DisLib Metrics

**BetaVAE** (Higgins et al., 2017)

The BetaVAE metric uses a biased estimator with tunable hyperparameters, although we follow the convention established in (Locatello et al., 2018) of using the *scikit-learn* defaults. For a sample in a batch, a pair of images, $(\mathbf{x}_1, \mathbf{x}_2)$, is generated by fixing the value of one of the data generative factors while uniformly sampling the rest. The absolute value of the difference between the latent codes produced from the image pairs is then taken, $\mathbf{z}_{\text{diff}} = |\mathbf{z}_1 - \mathbf{z}_2|$. A logistic classifier is fit with batches of $\mathbf{z}_{\text{diff}}$ variables and the corresponding index of the fixed ground-truth factor serves as the label. Once the classifier is trained, the metric itself is the mean classifier accuracy on a batch of held-out test data. The training minimizes the following loss:

$$L = \frac{1}{2}\mathbf{w}^T\mathbf{w} + \sum_{i=1}^{n} \log(\exp(-\mathbf{y}_i(\mathbf{z}_{\text{diff},i}^T\mathbf{w} + c)) + 1), \tag{25}$$

where $\mathbf{w}$ and $c$ are the learnable weight matrix and bias, respectively, and $\mathbf{y}$ is the index of the fixed ground-truth factor for the batch. The network is trained using the `lbfgs` optimizer (Byrd et al., 1995), which is implemented via the *scikit-learn* Python package (Pedregosa et al., 2011) in the Disentanglement Library (DisLib, Locatello et al., 2018). In the original work, the authors argue that their metric improves over a correlation metric such as the mean correlation coefficient by additionally measuring interpretability. However, the linear operation of $\mathbf{z}_{\text{diff},i}^T\mathbf{w} + c$ can perform demixing, which means the measure gives no direct indication of identifiability and thus does not guarantee that the latent encodings are interpretable, especially in the case of dependent factors. Additionally, as noted by Kim and Mnih (2018), BetaVAE can report perfect accuracy when all but one of the ground-truth factors are disentangled, since the classifier can trivially attribute the remaining factor to the remaining latents.

**FactorVAE** (Kim and Mnih, 2018)

For the FactorVAE metric, the variance of the latent encodings is computed for a large (10,000 in DisLib) batch of data where all factors could possibly be changing. Latent dimensions with variance

below some threshold (0.05 in DisLib) are rejected and not considered further. Next, the encoding variance is computed again on a smaller batch (64 in DisLib) of data where one factor is fixed during sampling. The quotient of these two quantities (with the larger batch variance as the denominator) is then taken to obtain a normalized variance estimate per latent factor. Finally, a majority-vote classifier is trained to predict the index of the ground-truth factor with the latent unit that has the lowest normalized variance. The FactorVAE score is the classification accuracy for a batch of held-out data.

**Mutual Information Gap** (Chen et al., 2018)

The Mutual Information Gap (MIG) metric was introduced as an alternative to the classifier-based metrics. It provides a normalized measure of the mean difference in mutual information between each ground truth factor and the two latent codes that have the highest mutual information with the given ground truth factor. As it is implemented in DisLib, MIG measures entropy by discretizing the model's latent code using a histogram with 20 bins equally spaced between the representation minimum and maximum. It then computes the discrete mutual information between the ground-truth values and the discretized latents using the *scikit-learn* `metrics.mutual_info_score` function (Pedregosa et al., 2011). For the normalization it divides this difference by the entropy of the discretized ground truth factors.

**Modularity** (Ridgeway and Mozer, 2018)

Ridgeway and Mozer (2018) measure disentanglement in terms of three factors: modularity, compactness, and explicitness. For modularity, they first measure the mutual information between the discretized latents and ground-truth factors using the same histogram procedure that was used for the MIG, resulting in a matrix, $M \in \mathbb{R}^{D' \times D}$ with entries for each mutual information pair. Their measure of modularity is then

$$\text{modularity} = \frac{1}{D'} \sum_{i=1}^{D'} \Theta \left( 1 - \frac{\sum_{j=1}^{D} M_{i,j}^2 - \max(M_i^2)}{\max(M_i^2)(D-1)} \right), \tag{26}$$

where $\max(M_i^2)$ returns the maximum of the vector of squared mutual information measurements between ground truth $i$ and each latent factor. Additionally, $\Theta$ is a selection function that returns zero for any $i$ where $\max(M_i^2) = 0$ and otherwise acts as the identity function.

**DCI Disentanglement** (Eastwood and Williams, 2018)

The DCI scores measure disentanglement, completeness, and informativeness, which have intuitive correspondence to the modularity, compactness, and explicitness of (Ridgeway and Mozer, 2018), respectively. To measure DCI Disentanglement, $D$ regressors are trained to predict each ground truth factor state given the latent encoding. The DisLib implementation uses the `ensemble.GradientBoostingClassifier` function from *scikit-learn* with default parameters, which trains $D$ gradient boosted logistic regression tree classifiers. Importance is assigned to each latent factor using the built-in `feature_importance_` property of the classifier, which computes the normalized total reduction of the classifier criterion loss contributed by each latent. Disentanglement is then measured as

$$\sum_{i=1} D(1 - H(I_i))\tilde{I}_i, \tag{27}$$

where $H$ is the entropy computed with the `stats.entropy` function from *scikit-learn*, $I \in \mathbb{R}^{D \times D'}$ is a matrix of the absolute value of the feature importance between each factor and each ground truth, and $\tilde{I}$ is a normalized version of the matrix

$$\tilde{I}_i = \frac{\sum_{j=1}^{D'} I_{i,j}}{\sum_{k=1}^{D} \sum_{j=1}^{D'} I_{k,j}} \tag{28}$$

**SAP Score** (Kumar et al., 2018)

To compute the SAP score, Kumar et al. (2018) first train a linear support vector classifier with squared hinge loss and $L_2$ penalty to predict each ground truth factor from each latent variable. In DisLib this is implemented with the `svm.LinearSVC` function with default parameters from *scikit-learn*. They construct a score matrix $S \in \mathbb{R}^{D' \times D}$, where each entry in the matrix is the

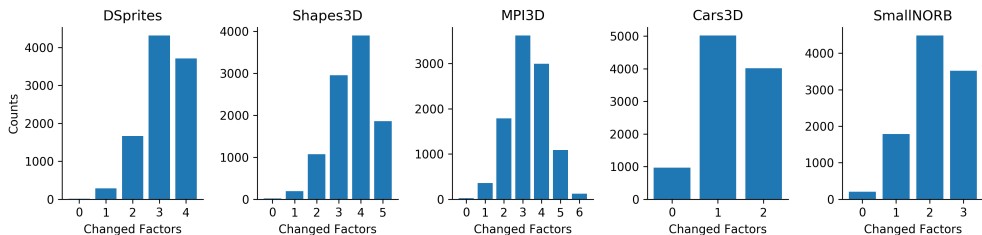

Figure 8: **Number of changing factors in LAP dataset**. For each dataset we sample 10,000 transitions and record the number of changing factors. These are indicated in the histograms. $\lambda = 1$, see Appendix D.

batch-mean classifier accuracy for predicting each ground truth given each individual latent encoding. For each generative factor, they compute the difference between the top two most predictive latent dimensions, which are the two highest scores in a given column of $S$. The mean (across ground-truth factors) of these differences is the SAP score.

# D  NATURAL DATASETS

We introduce several datasets to investigate disentanglement in more natural scenarios. Here, we provide an overview on the motivation and design of each dataset.

We have chosen to work with pairs of inputs as minimal sequences because we are interested in the first temporal derivative, more specifically in the sparsity of the transitions between pairs of images. Other methods that look at the second temporal derivative, such as work from Hénaff et al. (2019) on straightening, would require triplets as minimal sequences. Extending our approach beyond this minimal requirement would be simple in terms of the resulting ELBO (which would still factorise like in Eq. 4 because of the Markov property). The only additional complexity would be in the data and loss handling.

An issue with evaluating disentanglement on natural datasets is the fact that the existing disentanglement metrics require knowledge of the underlying generative process of the given data. Although we can observe that the world is composed of distinct entities that vary according to rules imposed by physics, we are unable to determine the appropriate "factors" that generate such scenes. To mitigate this problem, we compile object measurements by calculating the $x$ and $y$ coordinates of the center of mass as well as the area of object masks in natural video frames. We use these measurements to a) inform new disentanglement benchmarks with natural transitions that have similar complexity to existing benchmarks (Natural Sprites) and b) evaluate the ability of algorithms to decode intrinsic object properties (KITTI Masks). We additionally propose a simple extension to the existing DisLib datasets in the form of collecting images into pairs that exhibit sparse (i.e. Laplace) transition probabilities.

## D.1  UNIFORM TRANSITIONS (UNI)

The UNI extension is based on the description given by Locatello et al. (2020), where the number of changing factors is determined using draws from a uniform distribution. The key differences between our implementation and theirs is: (i) their code[3] randomly (with 50% probability) sets $k = 1$ even in the $k = $ Rnd setting, and (ii) we ensure that exactly k factors change. Though we consider these discrepancies minor, we nonetheless label all results reported directly from Locatello et al. (2020) with "LOC", as opposed to "UNI", for clarity.

## D.2    LAPLACE TRANSITIONS (LAP)

For each of the datasets in DisLib, we collect pairs of images. For each ground-truth factor, the first value in the pair is chosen from a uniform distribution across all possible values in latent space, while the second is chosen by weighting nearby values in latent space using Laplace distributed probabilities (see Eq. 2). We reject samples that would push a factor outside of the preset range provided by the dataset. We call this the *LAP* DisLib extension. Although the sparse prior indicates that any individual factor is more likely to remain constant, the number of factors that change in a given transition is still typically greater than one. To show this in Fig. 8, we sampled 10,000 transitions from each DisLib dataset with LAP transitions and computed the number of factors that had changed within a pair. This extension of the DisLib datasets provides a bridge from i.i.d. data to natural data by explicitly modeling the observed sparse marginal transition distributions. When training models on the LAP dataset it is possible to reject samples without transitions (i.e. all factors remain constant) since the pair would not result in any temporal learning signal. However, it would arguably be more natural to leave these samples as they would more accurately reflect occurrences of stationary objects in real data. We report the rejection setting in the main text, but found no significant difference between the two settings (see Appendix G).

This dataset also introduces a hyper-parameter $\lambda$ that controls the rate of the Laplace sampling distribution, while the location is set by the initial factor value. Effectively, when this rate is $\lambda = 1$ most of the factors change most of the time, whereas for a rate of $\lambda = 10$ most of the factors will not change most of the time. Note that this means $\lambda$ (inversely) changes the scale, which results in larger or smaller movements, but does not affect the distribution itself. In other words, the sparsity is unchanged, as the sparsity is controlled by the shape $\alpha$. We fix $\lambda = 1$, which yields multiple changes, thus making this dataset fundamentally different both in spirit and in practice, from the UNI dataset.

## D.3    YOUTUBE-VOS

For the YouTube dataset, we download annotations from the 2019 version of the video instance segmentation (Youtube-VIS) dataset (Yang et al., 2019)[4], which is built on top of the video object segmentation (Youtube-VOS) dataset (Xu et al., 2018). The dataset has multi-object annotations for every five frames in a 30fps video, which results in a 6fps sampling rate. The authors state that the temporal correlation between five consecutive frames is sufficiently strong that annotations can be omitted for intermediate frames to reduce the annotation efforts. Such a skip-frame annotation strategy enables scaling up the number of videos and objects annotated under the same budget, yielding 131,000 annotations for 2,883 videos, with 4,883 unique video object instances. Although we do not evaluate against YouTube-VOS in this study, we see it as the logical next step in transitioning to natural data. The large scale, lack of environmental constraints, and abundance of object types makes it the most challenging of the datasets considered herein.

The original image size of the YouTube-VOS dataset is $720 \times 1280$. In order to preserve the statistics of the transitions, we choose not to directly downsample to $64 \times 64$, but instead preserve the aspect ratio by downsampling to $64 \times 128$. In order to minimize the bias yielded by the extraction method, noting the center bias typically present in human videos, we extract three overlapping, equally spaced $64 \times 64$ pixel windows with a stride of 32. For each resulting $64 \times 64 \times T$ sequence, where $T$ denotes the number of time steps in the sequence, we filter out all pairs where the given object instance is not present in adjacent frames, resulting in 234,652 pairs.

## D.4    NATURAL SPRITES

The benchmark is available at `https://zenodo.org/record/3948069`.

Without a metric for disentanglement that can be applied to unknown data generating processes, we are limited to synthetic datasets with known ground-truth factors. Let us take dSprites (Matthey et al., 2017) as an example. The dataset consists of all combinations of a set of latent factor values, namely,

- Color: white

---

[3] `https://github.com/google-research/disentanglement_lib/blob/master/disentanglement_lib/methods/weak/train_weak_lib.py#L48`

[4] `https://competitions.codalab.org/competitions/20127`

| Config | Scale | X | Y | (R, G, B) | Shape | Orientation |
|---|---|---|---|---|---|---|
| Continuous | YT [2375] | YT [197342] | YT [187112] | (1.0, 1.0, 1.0) | (square, triangle, star_4, spoke_4) | (0,9,...,342,351) |
| Discrete | YT [6] | YT [32] | YT [32] | (1.0, 1.0, 1.0) | (square, triangle, star_4, spoke_4) | (0,9,...,342,351) |

Table 4: Natural Sprite Configs. Values in brackets refer to the number of unique values. Shapes presented are predefined in Spriteworld (Watters et al., 2019).

- Shape: square, ellipse, heart
- Scale: 6 values linearly spaced in $[0.5, 1]$
- Orientation: 40 values in $[0, 2\pi]$
- Position $X$: 32 values in $[0, 1]$
- Position $Y$: 32 values in $[0, 1]$

Given the limited set of discrete values each factor can take on, all possible samples can be described by a tractable dataset, compiled and released to the public. But, in reality, all of these factors should be continuous: a spectrum of possible colors, shapes, scales, orientations, and positions exist. We address this by constructing a dataset that is augmented with natural and continuous ground truth factors, using the mask properties measured from the YouTube dataset described in Appendix D.3.

We can choose the complexity of the dataset by discretizing the 234,652 transition pairs of position and scale into an arbitrary number of bins. In this study, we discretize to match the number of possible object states as dSprites, which we present in Table 4. This helps isolate the effect of including natural transitions from the effect of increasing data complexity. We produce a pair by fixing the color, shape, and orientation, but updating the position and scale with transitions sampled from the YouTube measurements. We motivate fixing shape and color by noting that this is consistent with object permanence in the real world. We decided to fix the orientation because we do not currently have a way to approximate it from object masks and we did not want to introduce artificial transition probabilities. To minimize the effect of extreme outliers, we filter out 10% of the data by removing frames if the mask area falls below the 5% or above the 95% quantiles, which reduces the number of pairs to 207,794. Finally, we use the Spriteworld (Watters et al., 2019) renderer to generate the images. Spriteworld allows us to render entirely new sprite objects at the precise position and scale as was measured from YouTube. For example, if one would want to apply YouTube-VOS transitions to MPI3D (Gondal et al., 2019), this option is unavailable without the associated renderer.

In relation to the Laplace transitions described in section D.2, this update i) produces pairs that correspond to transitions observed in real data, ii) allows for smooth transitions by defining the data generation process as opposed to being limited by the given collected dataset (e.g. dSprites), and iii) includes complex dependencies among factors that are present in natural data. We generate the data online, thus training the model to fit the underlying distribution as opposed to a sampled finite dataset.

However, as noted previously, all supervised metrics aggregated in DisLib are inapplicable to continuous factors, which is problematic as the generating distribution is effectively continuous with respect to a subset of the factors. Therefore, we limit our quantitative evaluation to MCC for continuous datasets. However, we are able to evaluate disentanglement with the standard metrics on the discretized version.

### D.5 KITTI MOTS Pedestrian Masks (KITTI Masks)

The benchmark is available at `https://zenodo.org/record/3931823`.

While Natural Sprites enables evaluation of disentanglement with natural transitions, we note that any disentanglement framework that requires knowledge of the underlying generative factors is unrealistic for real-world data. Measurements such as scale and position correspond to object properties that are ecologically relevant to the observer and can serve as suitable alternatives to the typical generative factors. We directly test this using our KITTI Masks dataset.

To create the dataset, we download annotations from the Multi-Object Tracking and Segmentation (MOTS) Evaluation Benchmark (Voigtlaender et al., 2019; Geiger et al., 2012; Milan et al., 2016),

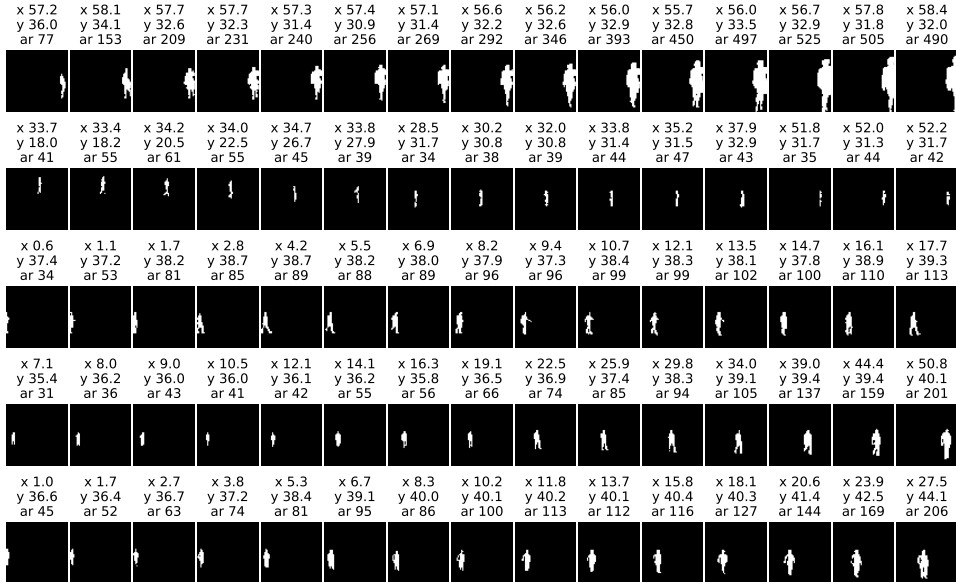

Figure 9: **KITTI Masks**. Each row corresponds to sequential frames from random sequences in the KITTI Mssks dataset. Above each image we denote measured object properties where $x, y$ correspond the center of mass position and $ar$ corresponds to the area.

which is split into KITTI MOTS and MOTSChallenge[5]. Both datasets contain sequences of pedestrians with their positions densely annotated in the time and pixel domains. For simplicity, we only consider the instance segmentation masks for pedestrians and do not use the raw data.

The resulting KITTI Masks dataset consists of 2,120 sequences of individual pedestrians with lengths between 2 and 710 frames each, resulting in a total of 84,626 individual frames. As we did with YouTube-VOS, we estimate ground truth factors by calculating the $x$ and $y$ coordinates of the center of mass of each pedestrian mask in each frame. We define the object size as the area of the mask, i.e. the total number of pixels. We consider the disentanglement performance for different mean time gaps between image pairs in table 2 and Appendix G.3. For samples and the corresponding ground truth factors see Fig. 9.

The original KITTI image sizes are $1080 \times 1920$ or $480 \times 640$ resolution for MOTSChallenge and between 370 and 374 pixels tall by 1224 and 1242 pixels wide for KITTI MOTS. The frame rates of the videos vary from 14 to 30 fps, which can be seen in Table 2 of Milan et al. (2016). We use nearest neighbor down-sampling for each frame such that the height was 64 pixels and the width is set to conserve the aspect ratio. After down-sampling, we use a horizontal sliding window approach to extract six equally spaced windows of size $64 \times 64$ (with overlap) for each sequence in both datasets. This results in a $64 \times 64 \times T$ sequence, where $T$ denotes the number of time steps in the sequence. Note that here we make reasonable assumptions on horizontal translation and scale invariance of the dataset. We justify the assumed scale invariance by observing that the data is collected from a camera mounted onto a car which has varying distance to pedestrians. To confirm the translation invariance, we performed an ablation study on the number of horizontal images. Instead of six horizontal, equally spaced sliding windows, we only use two which leads to differently placed windows. We do not observe significant changes in the reported data statistics (e.g. the kurtosis of the fit stays within $\pm 10\%$ of the previous value for $\Delta x$ transitions). The values of $\Delta y$ and $\Delta area$ do not change significantly compared to Table 7.

For each resulting $64 \times 64 \times T$ sequence, where $T$ denotes the number of time steps in the sequence, we extract all individual pedestrian masks based on their object instance identity and create a new sequence for each pedestrian such that each resulting sequence only contains a single pedestrian. We ignore images with masks that have less than 30 pixels as they are too far away or occluded and were

---

[5]https://www.vision.rwth-aachen.de/page/mots

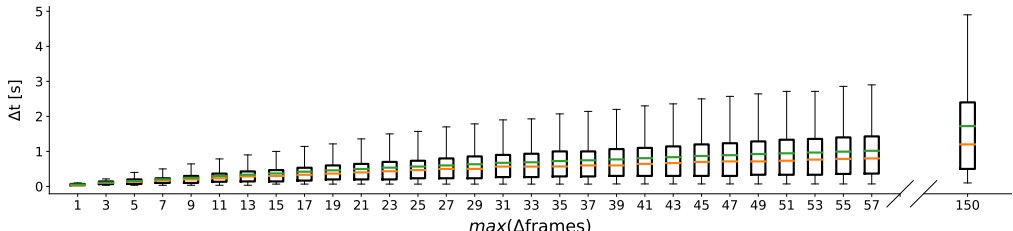

Figure 10: **KITTI Masks $\Delta$t**. Boxes indicate correspondence to physical time for different $\max(\Delta\text{frames})$ in the KITTI Masks datasets. The orange line denotes the median and the green line the mean. The whiskers cover the 5th and 95th percentile of data.

not recognizable by the authors. We keep all sequences of two or more frames, as the algorithm only requires pairs of frames for training.

We leave the maximum distance between time frames within a pair, $\max(\Delta\text{frames})$, as a hyper-parameter. For a given $\max(\Delta\text{frames})$, we report the mean change in physical time in seconds (denoted by $\text{mean}(\Delta t)$). We test adjacent frames ($\max(\Delta\text{frames}) = 1$), which corresponds to a $\text{mean}(\Delta t = 0.05)$ and $\max(\Delta\text{frames}) = 5$, which corresponds to a $\text{mean}(\Delta t = 0.15)$. This procedure is motivated by the fact that different sequences were recorded with different frame rates and reporting the $\text{mean}(\Delta t)$ in seconds allows for a physical interpretation. The relationship between $\max(\Delta\text{frames})$ and $\text{mean}(\Delta t)$ is in Fig. 10. We show results for testing additional values of $\text{mean}(\Delta t)$ in Appendix G.3.

During training, we augment the data by applying horizontal and vertical translations of $\pm 5$ pixels and rotations of $\pm 2°$ degree. We apply the exact same data augmentation to both images within a pair to not change any transition statistics.

We note that both YouTube-VOS (Xu et al., 2018; Yang et al., 2019) and KITTI-MOTS (Voigtlaender et al., 2019; Geiger et al., 2012; Milan et al., 2016) are multi-object datasets, although we consider each unique object (mask) separately. Multi-object representation learning and disentanglement are highly connected, in fact they have recently begun to be used interchangeably (Wulfmeier et al., 2020).

To briefly comment on possible extensions in this direction, we see no reason why our prior would not be beneficial to multi-object methods such as MONet (Burgess et al., 2019) and IODINE (Greff et al., 2019), or video extensions such as ViMON (Weis et al., 2020) and OP3 (Veerapaneni et al., 2019).

# E  MODEL TRAINING AND SELECTION

We train all models on all datasets provided in DisLib with the UNI and LAP variants.

All models are implemented in PyTorch (Paszke et al., 2019). To facilitate comparison, the training parameters, e.g. optimizer, batch size, number of training steps, as well as the VAE encoder and decoder architecture are identical to those reported in (Locatello et al., 2018; 2020). We use this architecture for all datasets, only adjusting the number of input channels (greyscale for dSprites, smallNORB, and KITTI Masks; three color channels for all other datasets).

The model formulation is agnostic to the direction of time. Therefore, to increase the temporal training signal at a fixed computational cost for each batch of input pairs $(\mathbf{x}_0, \mathbf{x}_1)$, we optimize the model in both directions i.e. optimizing the model objective for both $t_0 = 0$, $t_1 = 1$ as well as $t_0 = 1$, $t_1 = 0$.

## F    EXTENDED COMPARISONS AND CONTROLS

### F.1    COMPARISON TO NONLINEAR ICA

#### F.1.1    THEORETICAL COMPARISON

Nonlinear ICA has recently been advanced significantly by several papers from Hyvärinen and colleagues. Of these studies, the two that are most comparable to our work is Hyvärinen and Morioka (2017), which uses an unsupervised contrastive loss for nonlinear demixing and Khemakhem et al. (2020a), which extends the nonlinear ICA framework to include variational autoencoders (VAEs). However, our theory covers an important class of transitions relevant for natural data that is not covered by the identifiability proofs of either of the aforementioned studies.

As a specific comparison to the first paper, the non-Gaussian autoregressive model that their identifiability proof rests upon (Eq. 8 in Hyvärinen and Morioka, 2017) assumes that the second derivative of the innovation probability density function is less than zero to satisfy *uniform dependence*, which is only met for $\alpha > 1$ for generalized Laplace transition distributions. While they denote (footnote 3) that Laplace distributions ($\alpha = 1$) are not covered by their theory, they offer a suggestion for a smooth approximation. However, they do not demonstrate that this approximation is useful in practice, or offer a solution to a general class of sparse distributions for $\alpha \le 1$. We chose a generalized Laplacian to fit our data and for our model assumption as it allows for simple parameterization of fits to data (e.g. $\alpha = 0.5$ for natural movie transitions), but is simultaneously quite expressive (Sinz et al., 2009). Though we use $\alpha = 1$ in practice for our estimation method, we prove identifiability up to permutations and sign flips for any $\alpha < 2$, covering all sparse distributions under the expressive generalized Laplacian model. In addition, we assume a Gaussian marginal distribution that allows us to derive a fundamentally stronger proof of identifiability – where we identify up to permutation and sign-flips. Hyvärinen and Morioka (2017) only identify the sources up to arbitrary non-linear element-wise transformations. Thus they require a subsequent step of ICA (under the typical assumption that at most one marginal source distribution is Gaussian) to recover the signal up to permutations and sign flips for a class of distributions where it is unclear whether they account for temporal sparsity.

The work of Khemakhem et al. (2020a) has a couple of differences from our own, most notable of which is the form of the conditional prior, $p(\mathbf{z}_t|\mathbf{z}_{t-1})$. They assume that the conditional posterior is part of the exponential family, which does not include Laplacian conditionals. Though the exponential family contains the Laplace distribution with fixed mean as its member, it does not allow their approach to model sparse transitions. They assume that the natural parameters of the exponential family distribution are conditioned on $\mathbf{z}_{t-1}$, meaning that only the scale but not the mean of the Laplace prior for $\mathbf{z}_t$ can be modulated by the previous time step, thus not allowing for sparse transition probabilities. Additionally, their implementation requires the number of classes (i.e. states of the conditioning variable) to equal the number of stationary segments, which is impractical for the datasets we consider.

Thus, we provide a closer match to natural data transitions, with a stronger identifiability result. We provide validation by performing an extensive evaluation leveraging our contributed datasets as well as the models, metrics, and datasets provided by the Disentanglement Library (DisLib, discussed in section 4). We consider methods from the disentanglement literature (Locatello et al., 2020) as well as nonlinear ICA (Hyvärinen and Morioka, 2017), that are functionally capable of processing transitions.

#### F.1.2    EMPIRICAL COMPARISON

Hyvärinen and Morioka (2017) conducted a simulation where the sources in the nonlinear ICA model come from a linear autoregressive (AR) model with non-Gaussian innovations. Specifically, temporally dependent 20-dimensional source signals were randomly generated according to $\log p(s(t)|s(t-1)) = -|s(t) - 0.7s(t-1)|$. Though this generative process was noted to not be covered by the theory presented in (Hyvärinen and Morioka, 2017), the authors demonstrated that PCL could reconstruct the source signals reasonably well even for the nonlinear mixture case. Given our practical use of a Laplacian conditional, we found it a valuable comparison to evaluate our theory in this artificial setting.

| Method | L=1 | L=2 | L=3 | L=4 | L=5 |
|---|---|---|---|---|---|
| PCL | **0.998** | 0.960 | 0.950 | 0.917 | 0.902 |
| PCL (NF) | 0.946 | 0.918 | 0.918 | 0.917 | 0.876 |
| SlowFlow | 0.997 | **0.987** | **0.982** | **0.975** | **0.975** |

Table 5: MCC using linear correlation where $L$ denotes the number of mixing layers.

Given the discussion in Appendix B, we use SlowFlow for these experiments. For computational tractability in demixing highly nonlinear transformations, we consider normalizing flows (Dinh et al., 2017a;b; Kingma and Dhariwal, 2018), namely volume-preserving flows (Sorrenson et al., 2017), as we find constraining the Jacobian determinant stabilizes learning. To ensure sufficient expressivity, we consider 6 coupling blocks, each containing a 2-layer MLP with 500 hidden units and ReLU nonlinearities. We compare to the PCL implementation presented in (Hyvärinen and Morioka, 2017), where an MLP with the same number of hidden layers as the mixing MLP was adopted. We use 100 hidden units as we did not find increasing the value improved performance. To account for the architectural difference serving as a possible confounder, we use the same normalizing flow encoder for optimizing the PCL objective, which we term "PCL (NF)".

While (Hyvärinen and Morioka, 2017) used leaky ReLU nonlinearities to make the mixing invertible, said mixing is non-differentiable. This is problematic for SlowFlow, as it involves gradient optimization of the Jacobian term, and more importantly, unlike PCL, aims to explicitly recover the mixing process. We thus use a a smooth version of the leaky-ReLU activation function with a hyperparameter $\alpha$ (Gresele et al., 2020),

$$s_L(x) = \alpha x + (1 - \alpha) \log(1 + e^x). \tag{29}$$

By ensuring the mixing process is smooth, we find that SlowFlow performs favorably relative to PCL (Table 5) when evaluated in the same setting, converging to a better optimum at higher levels of mixing.

## F.2 JOINT FACTOR DEPENDENCE EVALUATION

In order to consider joint dependencies among natural generative factors, we leverage Natural Sprites to construct modified datasets where time-pairs of factors are shuffled per-factor (e.g. combining the x transition from one clip with the y transition from a different clip). This destroys dependencies between the factors, while maintaining the sparse marginal distributions. In Fig. 11 (right), we show 2D marginals before (blue) and after (orange) this shuffling. The additional density on the diagonals in the unshuffled data reveals dependencies between pairs of factors on both datasets. As mentioned in section 3.4, the observed dependency is mismatched from the theoretical assumptions of our model.

We test how robust SlowVAE is to such a mismatch by training it on the *permuted* data and re-evaluating disentanglement. In Table 22, we highlight that the improvement of SlowVAE on the permuted (i.e. independent) continuous Natural Sprites is not significant. In Table 21, we surprisingly find an overall improved score with non-permuted transitions (i.e. with dependencies), with three out of seven metrics showing a significant improvement. This is in line with Fig. 1f in Khemakhem et al. (2020b), where, at least for simple mixing, a model (Khemakhem et al., 2020a) that does not account for dependencies performs as well as one that does (Khemakhem et al., 2020b). We conclude that these preliminary results do not support the hypothesis that SlowVAE's disentanglement is reliant upon the model assumption that the factors are independent, but do acknowledge that the empirical effect of statistical dependence in natural video warrants further exploration (Träuble et al., 2020; Yang et al., 2020).

## F.3 TRANSITION PRIOR ABLATION

We consider an ablated model which minimizes a KL-divergence term between the posteriors at time-step $t$ and time-step $t - 1$. This encourages the model to match the posteriors of both time points as closely as possible, and resembles a probabilistic variant of Slow Feature Analysis (Turner and Sahani, 2007). Specifically, we set $p(\mathbf{z}_t|\mathbf{z}_{t-1}) = q(\mathbf{z}_{t-1}|\mathbf{x}_{t-1})$, replacing the Laplace prior with the

posterior of the previous time step. This is equivalent to a Gaussian ($\alpha = 2$) transition prior, where the mean and variance are specified by the previous time step. We ablate over the regularization parameter $\gamma$ and provide results in Tables 14 and 15, although we note that we still use the same hyperparameter values for SlowVAE as in all other experiments. As predicted by our theoretical result, $\alpha = 2$ leads to *entangled* representations in aggregate across evaluated datasets and metrics, even when considering a spectrum of $\gamma$ values, resulting in a drastic reduction in scores, particularly on dSprites and Natural Sprites.

# G    ADDITIONAL RESULTS

## G.1    EXTENDED DATA ANALYSIS

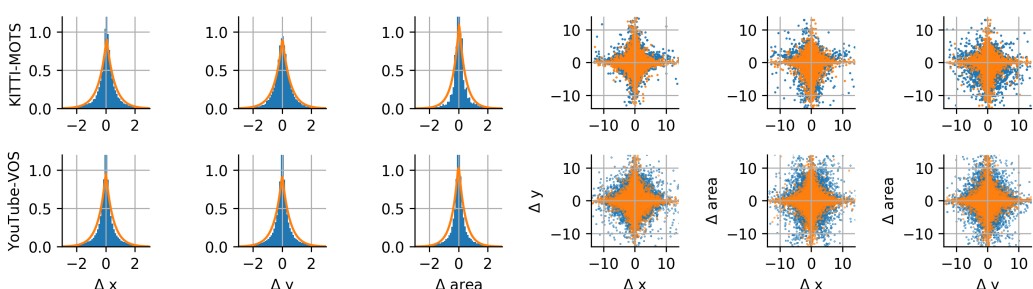

Figure 11: **Statistics of Natural Transitions**. Left) Distribution over transitions for horizontal ($\Delta x$) and vertical ($\Delta y$) position as well as mask/object size ($\Delta area$) for both datasets. Orange lines indicate fits of generalized Laplace distributions (Eq. 2). Right) 2D marginal distribution over pairs of factor transitions (blue) and permuted pairs (orange) that indicate the marginal distributions when made independent.

| dataset | N | $\Delta$ area | $\Delta$ x | $\Delta$ y |
|---|---|---|---|---|
| KITTI-MOTS | 82506 | 0.45 | 0.59 | 0.69 |
| YouTube-VOS | 234652 | 0.44 | 0.52 | 0.55 |

Table 6: Shape parameters ($\alpha$) of the fitted generalized Laplace distributions in Fig. 11.

We report the empirical estimates of Kurtosis in Table 7. We report the log-likelihood scores for the $\Delta$ area, $\Delta$ x, $\Delta$ y statistics in Tables 8, 9, and 10, respectively for a Normal, a Laplace and a generalized Laplace/Normal distribution. For these distributions, we also report the fit parameters for the $\Delta$ area, $\Delta$ x, $\Delta$ y statistics in Tables 11, 12, and 13, respectively, where the shape parameter $\alpha$ of the generalized Laplacian is in bold face. As a higher likelihood indicates a better fit, we can see further evidence that natural transitions are highly leptokurtic; a Laplace distribution ($\alpha = 1$) is a better fit than a Gaussian ($\alpha = 2$), while the generalized Laplacian yields the highest likelihood consistently with $\alpha \approx 0.5$ for all measurements, as indicated in the main paper. For the plots in Figs. 1 and 11, we set the standard deviation of each component to 1 and clipped the minimum ($-5$) and maximum (5) values.

We note that while the marginal transitions appear sparse in metrics computed from the given object masks, our analysis considers 2D projections of objects instead of the transition statistics in their 3D environment. Understanding the relationship between 3D and 2D transition statistics is a compelling question from a broader perspective of visual processing, but unfortunately, the KITTI-MOTS masks (Voigtlaender et al., 2019; Geiger et al., 2012; Milan et al., 2016) lack the associated depth data required to answer it. Nonetheless, the natural scene statistics we compute are relevant, given that most computer vision models and vision-based animals see the 3D world as projected onto their 2D receptor arrays.

| dataset | N | $\Delta$ area | $\Delta$ x | $\Delta$ y |
|---------|-----|-------|-------|-------|
| KITTI | 82506 | 68.92 | 38.50 | 65.39 |
| YouTube | 234652 | 76.49 | 39.98 | 35.59 |

Table 7: Empirical estimates of Kurtosis for mask transitions per metric for each dataset.

| dataset | N | genlaplace | normal | laplace |
|---------|-----|-----------|--------|---------|
| KITTI | 82506 | -3.21e+05 | -3.79e+05 | -3.35e+05 |
| YouTube | 234652 | -1.29e+06 | -1.45e+06 | -1.33e+06 |

Table 8: Maximum likelihood scores for the considered distributions on $\Delta$ **area** for each dataset.

| dataset | N | genlaplace | normal | laplace |
|---------|-----|-----------|--------|---------|
| KITTI | 82506 | -8.72e+04 | -1.20e+05 | -9.25e+04 |
| YouTube | 234652 | -4.50e+05 | -5.64e+05 | -4.74e+05 |

Table 9: Maximum likelihood scores for the considered distributions on $\Delta x$ for each dataset.

| dataset | N | genlaplace | normal | laplace |
|---------|-----|-----------|--------|---------|
| KITTI | 82506 | -7.59e+04 | -1.07e+05 | -7.86e+04 |
| YouTube | 234652 | -4.40e+05 | -5.45e+05 | -4.60e+05 |

Table 10: Maximum likelihood scores for the considered distributions on $\Delta y$ for each dataset.

| dataset | N | genlaplace | normal | laplace |
|---------|-----|-----------|--------|---------|
| KITTI | 82506 | [**4.55e-01**, 1.00e+00, 1.01e+00] | [4.53e-01, 2.39e+01] | [1.00e+00, 1.07e+01] |
| YouTube | 234652 | [**4.44e-01**, 1.47e-16, 5.04e+00] | [2.25e-01, 1.16e+02] | [7.73e-09, 5.28e+01] |

Table 11: Parameter fits for the considered distributions on $\Delta$ **area** for each dataset. The parameters are (alpha, location, scale) for generalized Laplace/Normal, (location, scale) for the other two distributions.

| dataset | N | genlaplace | normal | laplace |
|---------|-----|-----------|--------|---------|
| KITTI | 82506 | [**5.87e-01**, 4.76e-02, 1.69e-01] | [5.34e-02, 1.04e+00] | [5.49e-02, 5.64e-01] |
| YouTube | 234652 | [**5.15e-01**, 1.15e-14, 2.57e-01] | [2.32e-03, 2.68e+00] | [7.54e-09, 1.38e+00] |

Table 12: Parameter fits for the considered distributions on $\Delta$ **x** for each dataset. The parameters are (alpha, location, scale) for generalized Laplace/Normal, (location, scale) for the other two distributions.

| dataset | N | genlaplace | normal | laplace |
|---------|-----|-----------|--------|---------|
| KITTI | 82506 | [**6.94e-01**, 1.02e-02, 2.32e-01] | [3.84e-02, 8.86e-01] | [1.71e-02, 4.77e-01] |
| YouTube | 234652 | [**5.48e-01**, 2.93e-13, 3.08e-01] | [8.81e-03, 2.47e+00] | [9.15e-04, 1.30e+00] |

Table 13: Parameter fits for the considered distributions on $\Delta$ **y** for each dataset. The parameters are (alpha, location, scale) for generalized Laplace/Normal, (location, scale) for the other two distributions.

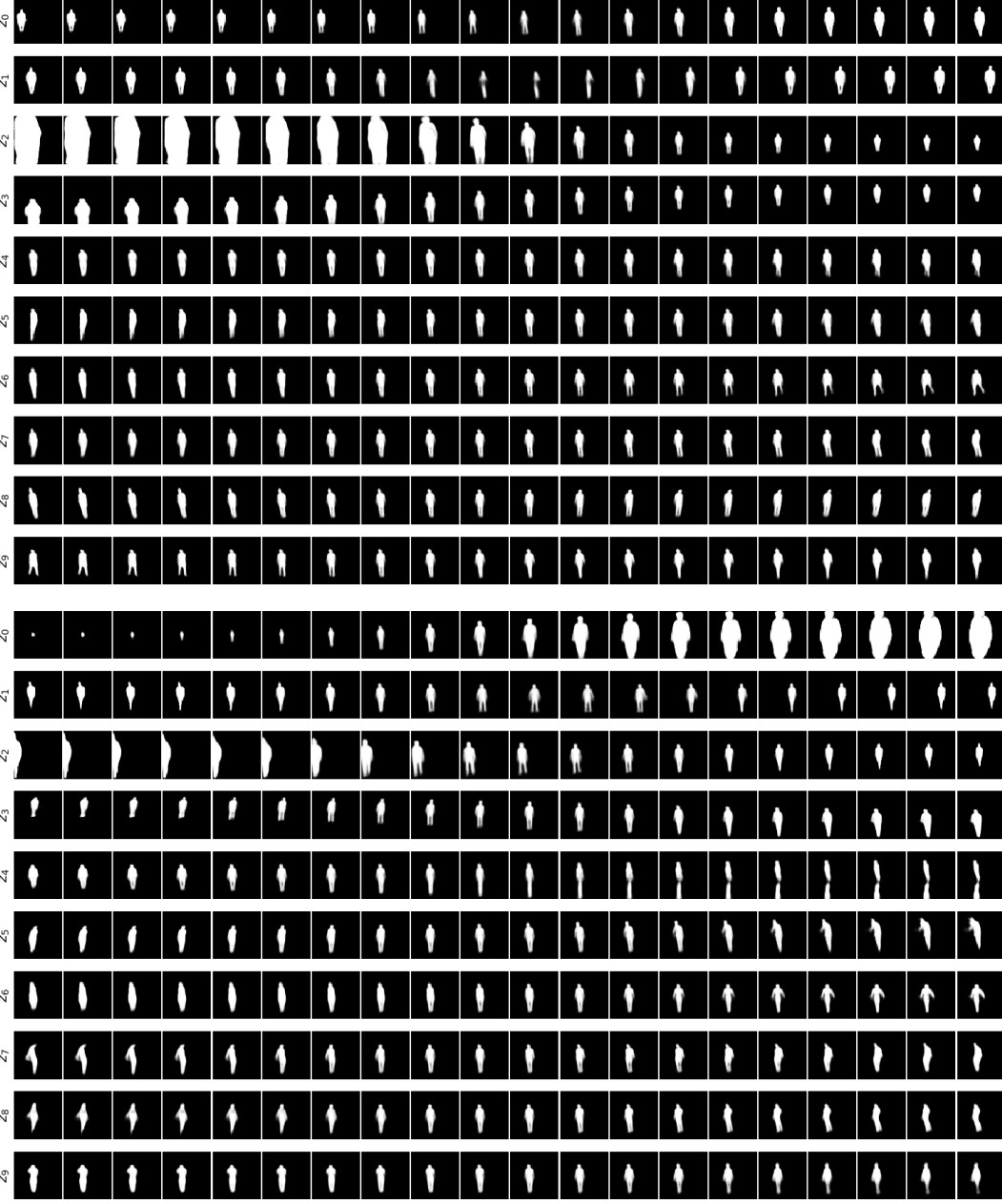

Figure 12: **KITTI Masks Latent Representations.** We show axis latent traversals along each dimension for the $\beta$-VAE (top) and SlowVAE (bottom). Here, the latents $z_i$ are sorted from top to bottom in ascending order according to the mean variance output of the encoder. With MCC correlation (see e.g. Fig. 20) the known ground truth factors are matched as following: $\beta$-VAE: scale$\sim z_2$, x-position$\sim z_1$ and y-position$\sim z_3$; SlowVAE: scale$\sim z_0$, x-position$\sim z_1$ and y-position$\sim z_3$. With these latent visualizations alone, there is no significant difference visible between $\beta$-VAE and SlowVAE. However, we see a quantitative difference with the MCC score (see Table 2) and a qualitative difference when directly observing latent embeddings (see Fig. 20).

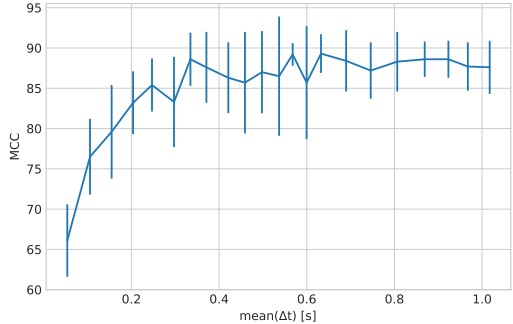

Figure 13: Ablation over mean($\Delta t$) for SlowVAE. Mean and standard deviation (s.d.) MCC scores

| Model | Data | BetaVAE | FactorVAE | MIG | MCC | DCI | Modularity | SAP |
|-------|------|---------|-----------|-----|-----|-----|------------|-----|
| SlowVAE | dSprites (Laplace) | 100.0 (0.0) | 97.5 (3.0) | 29.5 (9.3) | 69.8 (2.3) | 65.4 (3.6) | 96.5 (1.6) | 8.1 (3.0) |
| PM-VAE (16) | dSprites (Laplace) | 64.1 (7.0) | 44.8 (13.0) | 5.2 (2.3) | 45.0 (5.5) | 5.9 (3.9) | 93.5 (1.9) | 1.7 (0.8) |
| PM-VAE (10) | dSprites (Laplace) | 78.8 (7.5) | 59.4 (11.2) | 5.9 (1.8) | 49.2 (4.3) | 13.6 (5.6) | 92.7 (3.0) | 3.9 (1.7) |
| PM-VAE (8) | dSprites (Laplace) | 82.9 (2.8) | 61.2 (5.7) | 7.1 (2.6) | 49.6 (3.3) | 14.5 (3.5) | 91.6 (3.0) | 4.3 (1.6) |
| PM-VAE (4) | dSprites (Laplace) | 86.6 (2.7) | 64.1 (7.2) | 11.6 (5.0) | 52.0 (3.8) | 22.9 (3.7) | 90.9 (2.7) | 5.7 (2.8) |
| PM-VAE (2) | dSprites (Laplace) | 86.3 (2.4) | 62.9 (7.7) | 10.9 (3.2) | 50.0 (3.5) | 21.2 (5.3) | 92.3 (1.9) | 5.5 (2.0) |
| PM-VAE (1) | dSprites (Laplace) | 82.5 (5.4) | 58.4 (6.0) | 7.6 (3.6) | 45.9 (4.9) | 14.4 (5.1) | 92.1 (4.0) | 4.0 (2.0) |
| SlowVAE | Natural (Discrete) | 82.6 (2.2) | 76.2 (4.8) | 11.7 (5.0) | 52.6 (4.1) | 18.9 (5.5) | 88.1 (3.6) | 4.4 (2.3) |
| PM-VAE (16) | Natural (Discrete) | 72.7 (2.8) | 49.2 (3.7) | 2.8 (1.2) | 38.3 (3.2) | 6.9 (1.8) | 85.3 (1.8) | 1.2 (0.7) |
| PM-VAE (10) | Natural (Discrete) | 76.6 (3.6) | 52.0 (4.9) | 3.8 (2.2) | 39.0 (3.9) | 7.3 (1.8) | 87.0 (2.2) | 2.0 (1.0) |
| PM-VAE (8) | Natural (Discrete) | 74.6 (3.4) | 49.3 (4.4) | 3.1 (1.8) | 38.9 (3.2) | 7.1 (1.8) | 87.8 (1.7) | 1.6 (1.0) |
| PM-VAE (4) | Natural (Discrete) | 73.8 (3.8) | 48.8 (5.3) | 2.7 (1.5) | 35.7 (3.5) | 6.7 (2.0) | 87.4 (2.2) | 1.6 (0.9) |
| PM-VAE (2) | Natural (Discrete) | 73.4 (3.1) | 47.0 (5.3) | 2.2 (1.1) | 36.8 (2.4) | 6.2 (1.5) | 87.4 (1.9) | 1.1 (0.6) |
| PM-VAE (1) | Natural (Discrete) | 73.5 (3.3) | 49.7 (5.4) | 3.1 (1.6) | 36.9 (3.2) | 6.9 (1.8) | 86.9 (2.2) | 1.8 (0.7) |

Table 14: Mean and standard deviation (s.d.) metric scores across 10 random seeds. PM-VAE ($\gamma$) refers to replacing the Laplace prior with a KL-divergence term between the (Gaussian) posteriors at time-step $t$ and time-step $t - 1$, with conditional prior regularization, $\gamma$.

## G.2 ALL DISLIB RESULTS

We include results on all DisLib datasets, dSprites (Matthey et al., 2017), Cars3D (Reed et al., 2015), SmallNORB (LeCun et al., 2004), Shapes3D (Kim and Mnih, 2018), MPI3D (Gondal et al., 2019), in Tables 16, 17, 18, 19, and 20, respectively. We report both median (a.d.) to compare to the previous median scores reported in (Locatello et al., 2020), as well as the the more common mean (s.d.) scores for future comparisons and straightforward statistical estimates of significant differences between models. We also consider allowing for static transitions, which we denote with "NC", e.g. "LAP-NC", in the tabular results. As mentioned in Section 5, we use the same parameter settings for SlowVAE in all experiments, while model selection was performed not only per dataset, but per seed, for results from (Locatello et al., 2020).

## G.3 KITTI MASKS $\Delta t$ ABLATION

As seen in the main text, considering image pairs separated further apart in time appears beneficial. Here we evaluate a wider range by taking frames which are further apart in a sequence. $\max(\Delta\text{frames}) = N$ indicates that all pairs differ by *at most* $N$ frames. We chose an upper bound of $N$, rather than sampling pairs with a fixed separation, to account for the variable frame rates and sequence lengths in the original dataset (Milan et al., 2016) without introducing a confounding factor of varying dataset size. We report in Fig. 10 how the $\max(\Delta\text{frames})$ criterion corresponds to the mean time gap between image pairs (mean($\Delta t$)) in seconds. For further details, we refer to Appendix D.5.

In Fig. 13 we visualize an ablation over mean($\Delta t$). We find that model performance increased initially with larger temporal separation between data points, then plateaued. We also observe in Fig. 14 that the measured factor marginals remain sparse, with $\alpha < 1$, for all tested settings of mean($\Delta t$).

| Model | Data | MCC |
|---|---|---|
| SlowVAE | Natural (Continuous) | 49.1 (4.0) |
| PM-VAE (16) | Natural (Continuous) | 35.2 (3.7) |
| PM-VAE (10) | Natural (Continuous) | 33.2 (2.1) |
| PM-VAE (8) | Natural (Continuous) | 32.7 (3.1) |
| PM-VAE (4) | Natural (Continuous) | 33.7 (2.3) |
| PM-VAE (2) | Natural (Continuous) | 32.4 (3.2) |
| PM-VAE (1) | Natural (Continuous) | 34.2 (3.4) |
| SlowVAE | Kitti (mean$(\Delta t) = 0.05s$) | 66.1 (4.5) |
| PM-VAE (16) | Kitti (mean$(\Delta t) = 0.05s$) | 63.1 (9.3) |
| PM-VAE (10) | Kitti (mean$(\Delta t) = 0.05s$) | 57.4 (8.5) |
| PM-VAE (8) | Kitti (mean$(\Delta t) = 0.05s$) | 59.0 (5.6) |
| PM-VAE (4) | Kitti (mean$(\Delta t) = 0.05s$) | 51.8 (9.2) |
| PM-VAE (2) | Kitti (mean$(\Delta t) = 0.05s$) | 50.3 (7.4) |
| PM-VAE (1) | Kitti (mean$(\Delta t) = 0.05s$) | 38.4 (6.8) |
| SlowVAE | Kitti (mean$(\Delta t) = 0.15s$) | 79.6 (5.8) |
| PM-VAE (16) | Kitti (mean$(\Delta t) = 0.15s$) | 69.6 (5.9) |
| PM-VAE (10) | Kitti (mean$(\Delta t) = 0.15s$) | 78.2 (6.0) |
| PM-VAE (8) | Kitti (mean$(\Delta t) = 0.15s$) | 73.8 (10.0) |
| PM-VAE (4) | Kitti (mean$(\Delta t) = 0.15s$) | 67.9 (10.4) |
| PM-VAE (2) | Kitti (mean$(\Delta t) = 0.15s$) | 60.7 (8.8) |
| PM-VAE (1) | Kitti (mean$(\Delta t) = 0.15s$) | 60.9 (9.1) |

Table 15: Continuous ground-truth variable datasets. See Table 14 for details.

| Model (Data) | BetaVAE | FactorVAE | MIG | DCI | Modularity | SAP |
|---|---|---|---|---|---|---|
| $\beta$-VAE (*i.i.d.*) | 82.3 | 66.0 | 10.2 | 18.6 | 82.2 | 4.9 |
| Ada-ML-VAE (LOC) | 89.6 | 70.1 | 11.5 | 29.4 | 89.7 | 3.6 |
| Ada-GVAE (LOC) | 92.3 | 84.7 | 26.6 | 47.9 | 91.3 | 7.4 |
| SlowVAE (UNI) | 89.7 (3.8) | 81.4 (8.4) | 34.5 (9.6) | 50.0 (6.9) | 87.1 (2.0) | 5.1 (1.5) |
| SlowVAE (LAP) | 100.0 (0.0) | 99.2 (2.3) | 28.2 (8.2) | 65.5 (3.1) | 96.8 (1.4) | 6.0 (2.4) |
| SlowVAE (LAP-NC) | 100.0 (0.2) | 97.4 (4.4) | 29.1 (7.1) | 62.0 (4.2) | 97.4 (1.6) | 8.2 (2.9) |
| SlowVAE (UNI) | 87.0 (5.1) | 75.2 (11.1) | 28.3 (11.5) | 47.7 (8.5) | 86.9 (2.8) | 4.4 (2.0) |
| SlowVAE (LAP) | 100.0 (0.0) | 97.5 (3.0) | 29.5 (9.3) | 65.4 (3.6) | 96.5 (1.6) | 8.1 (3.0) |
| SlowVAE (LAP-NC) | 99.8 (0.6) | 95.2 (6.0) | 27.6 (8.6) | 61.5 (5.3) | 96.8 (1.8) | 8.4 (3.4) |

Table 16: **dSprites.** Median and absolute deviation (a.d.) metric scores across 10 random seeds (first three rows are from (Locatello et al., 2020)). The bottom three rows give mean and standard deviation (s.d.) for the models presented in this paper.

Increasing mean$(\Delta t)$ leads to increased diversity, and thus more information in the learning signal. However, it is worth noting that since SlowVAE assumes $\alpha = 1$ in the transitions, an increase in $\alpha$ from increasing the temporal gap leads to a reduction in mismatch.

Our results on increasing the temporal difference within pairs of inputs is in agreement with recent work by Oord et al. (2018, Table 2), who show increased performance in representation learning for larger separation between positive samples in a contrastive objective function. Additional related work from Tschannen et al. (2019) shows that temporal separation between frame embeddings influences the representation that is learned from videos.

## G.4 LATENT SPACE VISUALIZATIONS

We visualize differences in learned latent representations using image embedding in Figures 15- 28. We show four different plots for each dataset considered and include all available models. Each figure corresponds to a different dataset.

| Model (Data) | BetaVAE | FactorVAE | MIG | DCI | Modularity | SAP |
|---|---|---|---|---|---|---|
| $\beta$-VAE (*i.i.d.*) | 100.0 | 87.9 | 8.8 | 22.5 | 90.2 | 1.0 |
| Ada-ML-VAE (LOC) | 100.0 | 87.4 | 14.7 | 45.6 | 94.6 | 2.8 |
| Ada-GVAE (LOC) | 100.0 | 90.2 | 15.0 | 54.0 | 93.9 | 9.4 |
| SlowVAE (UNI) | 100.0 (0.0) | 90.4 (0.4) | 15.7 (1.5) | 48.9 (1.7) | 95.7 (1.0) | 1.6 (0.4) |
| SlowVAE (LAP) | 100.0 (0.0) | 91.0 (2.5) | 9.7 (1.1) | 51.0 (2.2) | 94.4 (1.1) | 1.7 (0.9) |
| SlowVAE (LAP-NC) | 100.0 (0.0) | 90.8 (1.1) | 9.3 (1.1) | 50.0 (2.0) | 94.6 (0.9) | 0.9 (0.9) |
| SlowVAE (UNI) | 100.0 (0.0) | 90.4 (0.5) | 15.4 (2.2) | 48.0 (2.4) | 95.4 (1.5) | 1.6 (0.5) |
| SlowVAE (LAP) | 100.0 (0.0) | 90.2 (3.5) | 10.4 (1.8) | 50.9 (2.7) | 94.1 (1.2) | 2.0 (1.1) |
| SlowVAE (LAP-NC) | 100.0 (0.0) | 90.9 (1.2) | 9.5 (1.4) | 50.2 (2.7) | 95.0 (1.2) | 1.7 (1.4) |

Table 17: **Cars3D.** Median and absolute deviation (a.d.) metric scores across 10 random seeds (first three rows are from (Locatello et al., 2020)). The bottom three rows give mean and standard deviation (s.d.) for the models presented in this paper.

| Model (Data) | BetaVAE | FactorVAE | MIG | DCI | Modularity | SAP |
|---|---|---|---|---|---|---|
| $\beta$-VAE (*i.i.d.*) | 74.0 | 49.5 | 21.4 | 28.0 | 89.5 | 9.8 |
| Ada-ML-VAE (LOC) | 91.0 | 72.1 | 31.1 | 34.1 | 86.1 | 15.3 |
| Ada-GVAE (LOC) | 87.9 | 55.5 | 25.6 | 33.8 | 78.8 | 10.6 |
| SlowVAE (UNI) | 78.8 (2.1) | 46.2 (1.9) | 23.7 (1.3) | 28.8 (0.6) | 92.1 (1.6) | 7.8 (1.0) |
| SlowVAE (LAP) | 86.0 (0.2) | 72.9 (0.7) | 25.8 (0.5) | 42.7 (0.9) | 97.7 (0.3) | 6.5 (0.4) |
| SlowVAE (LAP-NC) | 86.1 (0.7) | 73.7 (0.6) | 26.3 (0.5) | 42.5 (0.6) | 97.6 (0.3) | 6.5 (0.9) |
| SlowVAE (UNI) | 78.2 (3.8) | 47.0 (2.9) | 23.8 (1.8) | 28.7 (0.7) | 90.9 (2.1) | 7.8 (1.1) |
| SlowVAE (LAP) | 85.9 (0.3) | 73.1 (0.9) | 25.7 (0.6) | 42.6 (0.9) | 97.5 (0.3) | 6.8 (0.5) |
| SlowVAE (LAP-NC) | 85.7 (1.0) | 73.3 (0.8) | 26.2 (0.7) | 42.6 (0.8) | 97.6 (0.5) | 6.6 (1.3) |

Table 18: **SmallNORB.** Median and absolute deviation (a.d.) metric scores across 10 random seeds (first three rows are from (Locatello et al., 2020)). The bottom three rows give mean and standard deviation (s.d.) for the models presented in this paper.

In Figures 15- 21 we display the mean correlation coefficient matrix and the latent representations for each ground-truth, as described in the main text for Fig. 5.

The top row is the sorted absolute correlation coefficient matrix between the latents (rows) and the ground truth generating factors (columns). The latent dimensions are permuted such that the sum on the diagonal is maximal. This is achieved by an optimal, non-greedy matching process for each ground truth factor with its corresponding latent, as described in appendix C. As such, a more

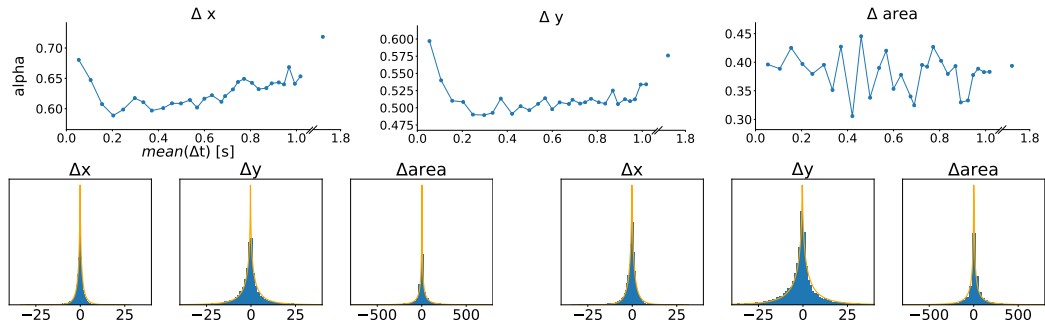

Figure 14: **KITTI Masks Sparseness**. We show the sparseness over time of the transitions for horizontal ($\Delta x$), vertical ($\Delta y$) as well as mask/object size ($\Delta$area) in KITTI Masks by plotting the $\alpha$ of a generalized Laplace fit for different mean($\Delta$t) (top). To display the quality of the fits, we show two exemplary fits at mean($\Delta$t) = 0.63 (bottom-left) and mean($\Delta$t) = 1.02 (bottom-right).

| Model (Data) | BetaVAE | FactorVAE | MIG | DCI | Modularity | SAP |
|---|---|---|---|---|---|---|
| $\beta$-VAE (*i.i.d.*) | 98.6 | 83.9 | 22.0 | 58.8 | 93.8 | 6.2 |
| Ada-ML-VAE (LOC) | 100.0 | 100.0 | 50.9 | 94.0 | 98.8 | 12.7 |
| Ada-GVAE (LOC) | 100.0 | 100.0 | 56.2 | 94.6 | 97.5 | 15.3 |
| SlowVAE (UNI) | 100.0 (0.1) | 97.3 (4.0) | 64.4 (8.4) | 82.6 (4.4) | 95.5 (1.6) | 5.8 (0.9) |
| SlowVAE (LAP) | 100.0 (0.0) | 95.9 (2.6) | 62.5 (3.1) | 85.6 (4.0) | 98.1 (0.6) | 8.2 (1.7) |
| SlowVAE (LAP-NC) | 100.0 (1.6) | 97.0 (2.0) | 63.6 (5.4) | 86.7 (4.1) | 98.4 (1.4) | 7.0 (2.1) |
| SlowVAE (UNI) | 99.9 (0.3) | 95.4 (5.2) | 58.8 (13.0) | 82.3 (5.4) | 95.2 (2.0) | 5.7 (1.4) |
| SlowVAE (LAP) | 100.0 (0.0) | 95.0 (3.2) | 61.5 (4.5) | 85.0 (4.7) | 98.3 (0.8) | 8.9 (2.6) |
| SlowVAE (LAP-NC) | 98.4 (4.9) | 97.4 (2.4) | 61.6 (10.6) | 86.1 (5.2) | 98.2 (1.6) | 8.2 (2.6) |

Table 19: **Shapes3D.** Median and absolute deviation (a.d.) metric scores across 10 random seeds (first three rows are from (Locatello et al., 2020)). The bottom three rows give mean and standard deviation (s.d.) for the models presented in this paper.

| Model (Data) | BetaVAE | FactorVAE | MIG | DCI | Modularity | SAP |
|---|---|---|---|---|---|---|
| $\beta$-VAE (*i.i.d.*) | 54.6 | 32.2 | 7.2 | 19.5 | 87.4 | 3.7 |
| Ada-ML-VAE (LOC) | 72.6 | 47.6 | 24.1 | 28.5 | 87.5 | 7.4 |
| Ada-GVAE (LOC) | 78.9 | 62.1 | 28.4 | 40.1 | 91.6 | 21.5 |
| SlowVAE (UNI) | 58.5 (0.9) | 38.6 (2.3) | 32.2 (1.0) | 29.9 (1.3) | 89.2 (2.0) | 8.8 (0.8) |
| SlowVAE (LAP) | 67.6 (6.1) | 42.4 (6.1) | 32.0 (1.8) | 35.9 (2.2) | 89.5 (1.5) | 9.7 (0.8) |
| SlowVAE (LAP-NC) | 60.1 (2.7) | 39.2 (1.7) | 30.6 (0.7) | 34.3 (0.7) | 85.9 (1.1) | 9.3 (0.9) |
| SlowVAE (UNI) | 58.6 (1.1) | 38.5 (3.2) | 32.2 (1.2) | 30.1 (1.6) | 89.4 (2.6) | 8.7 (1.0) |
| SlowVAE (LAP) | 66.6 (6.9) | 45.5 (8.3) | 32.9 (2.6) | 35.5 (2.7) | 89.2 (1.9) | 9.7 (1.2) |
| SlowVAE (LAP-NC) | 61.0 (3.6) | 40.3 (2.5) | 30.4 (0.8) | 34.2 (1.0) | 86.6 (1.7) | 9.3 (1.0) |

Table 20: **MPI3D.** Median and absolute deviation (a.d.) metric scores across 10 random seeds (first three rows are from (Locatello et al., 2020)). The bottom three rows give mean and standard deviation (s.d.) for comparison with other tables.

prevalent diagonal structure corresponds to a better mapping between the ground-truth factors and latent encoding.

The middle set of plots are latent embeddings of random training data samples. The x-axis denotes the ground truth generating factor and the y-axis denotes the corresponding latent factor as matched according to the main diagonal of the correlation matrix. For each dataset, we further color-code the latents by a categorical variable as denoted in each figure.

The bottom set of plots show the ground truth encoding compared to the second best latent as opposed to the diagonally matched latent. This plot can be used to judge how much the correspondence between latents is one-to-one or rather one-to-many.

To further investigate the latent representations, we show a scatter plot over the best and second best latents in figures 22-28. Here, the color-coding is matched by the ground truth factor denoted in each row.

When comparing the correlation matrix with the corresponding scatter plots, one can see that embeddings with sinusoidal curves have low correlation, which illustrates a shortcoming of the metric. Another limitation is that categorical variables which have no natural ordering have an order-dependent MCC score, indicating the permutation variance of MCC. With SlowVAE, we can infer three different types of embeddings. First, we have simple ordered ground truth factors with non-circular boundary conditions. Here, SlowVAE models often show a clear one-to-one correspondence

| Model | $\gamma$ | $\lambda$ | Data | Permuted? | BetaVAE | FactorVAE | MIG | MCC | DCI | Modularity | SAP |
|---|---|---|---|---|---|---|---|---|---|---|---|
| SlowVAE | 10 | 6 | Natural (Discrete) | Yes | 77.6 (4.1) | 69.7 (6.5) | 8.5 (4.4) | 49.9 (3.5) | 17.6 (2.8) | 89.8 (3.2) | 1.8 (0.9) |
| SlowVAE | 10 | 6 | Natural (Discrete) | No | **82.6** (2.2) | **76.2** (4.8) | 11.7 (5.0) | 52.6 (4.1) | 18.9 (5.5) | 88.1 (3.6) | **4.4** (2.3) |

Table 21: Impact of removing natural dependence on Discrete Natural Sprites.

| Model | $\gamma$ | $\lambda$ | Data | Permuted? | MCC |
|-------|------|------|------|-----------|-----|
| SlowVAE | 10 | 6 | Natural (Continuous) | Yes | 52.9 (4.2) |
| SlowVAE | 10 | 6 | Natural (Continuous) | No | 49.1 (4.0) |

Table 22: Impact of removing natural dependence on Continuous Natural Sprites.

(e.g. Fig 22 scale, x-position and y-position; Fig 25 $\theta$-rotation; Fig 26 $\Phi$-rotation). Second, we observe circular embeddings due to boundary conditions for certain factors (e.g. Fig 15, 22 3rd row; Fig 16, 23 2nd row). Note that not all datasets with orientations exhibit full rotations and thus do not have circular boundary conditions, e.g. smallNORB. Finally, we have categorical variables, where no order exists (e.g. Fig. 16, 23 top row, Fig 17, 24 top row, Fig 18, 25 top row) resulting in separated but not necessarily ordered clusters.

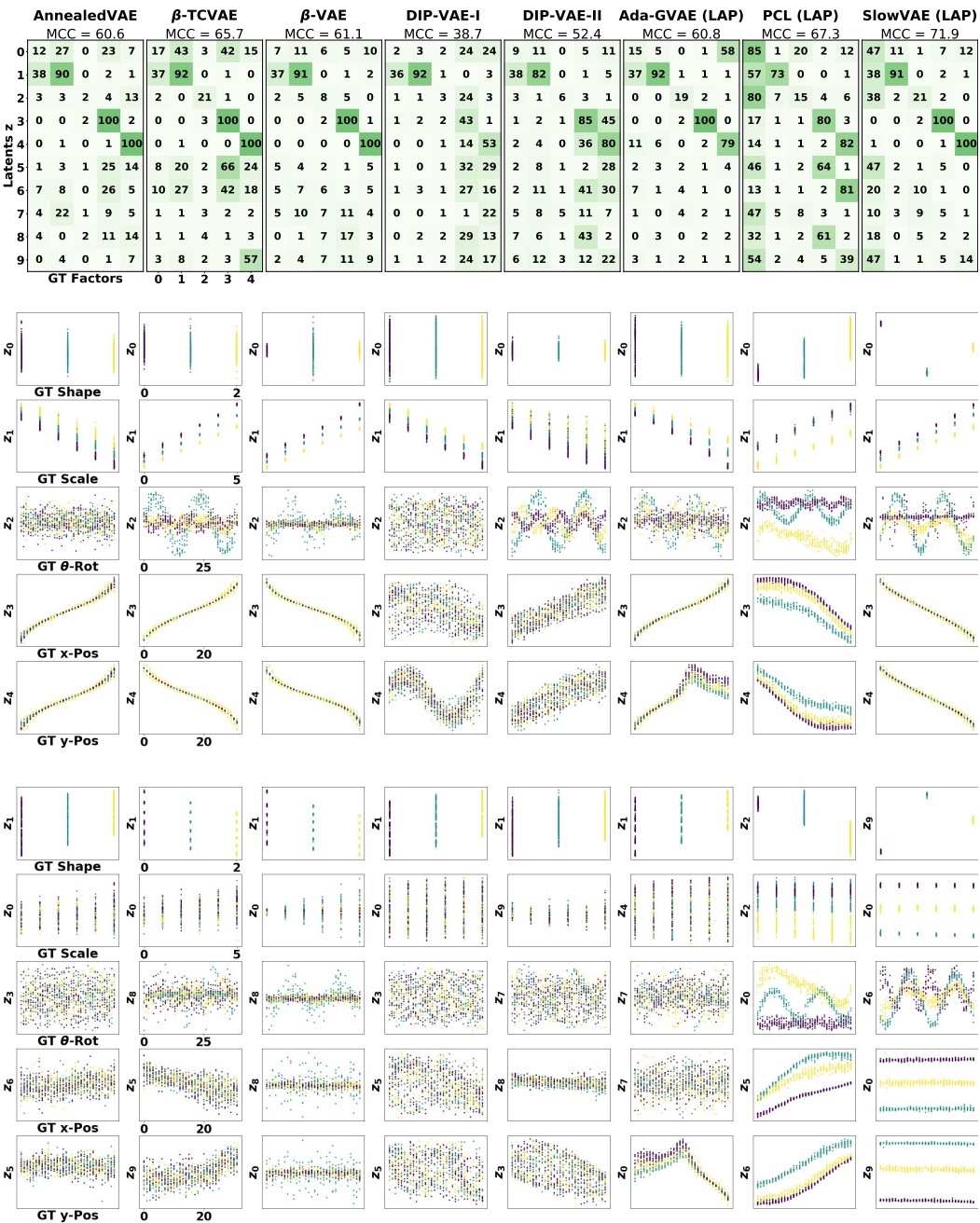

Figure 15: **DSprites Latent Representations.** Top, MCC correlation matrices. Middle five rows, model latent over highest correlating ground truth factor. Bottom five rows, model latent over second highest correlating ground truth factor. The color-coding corresponds to the shapes: heart/yellow, ellipse/turquoise and square/purple.

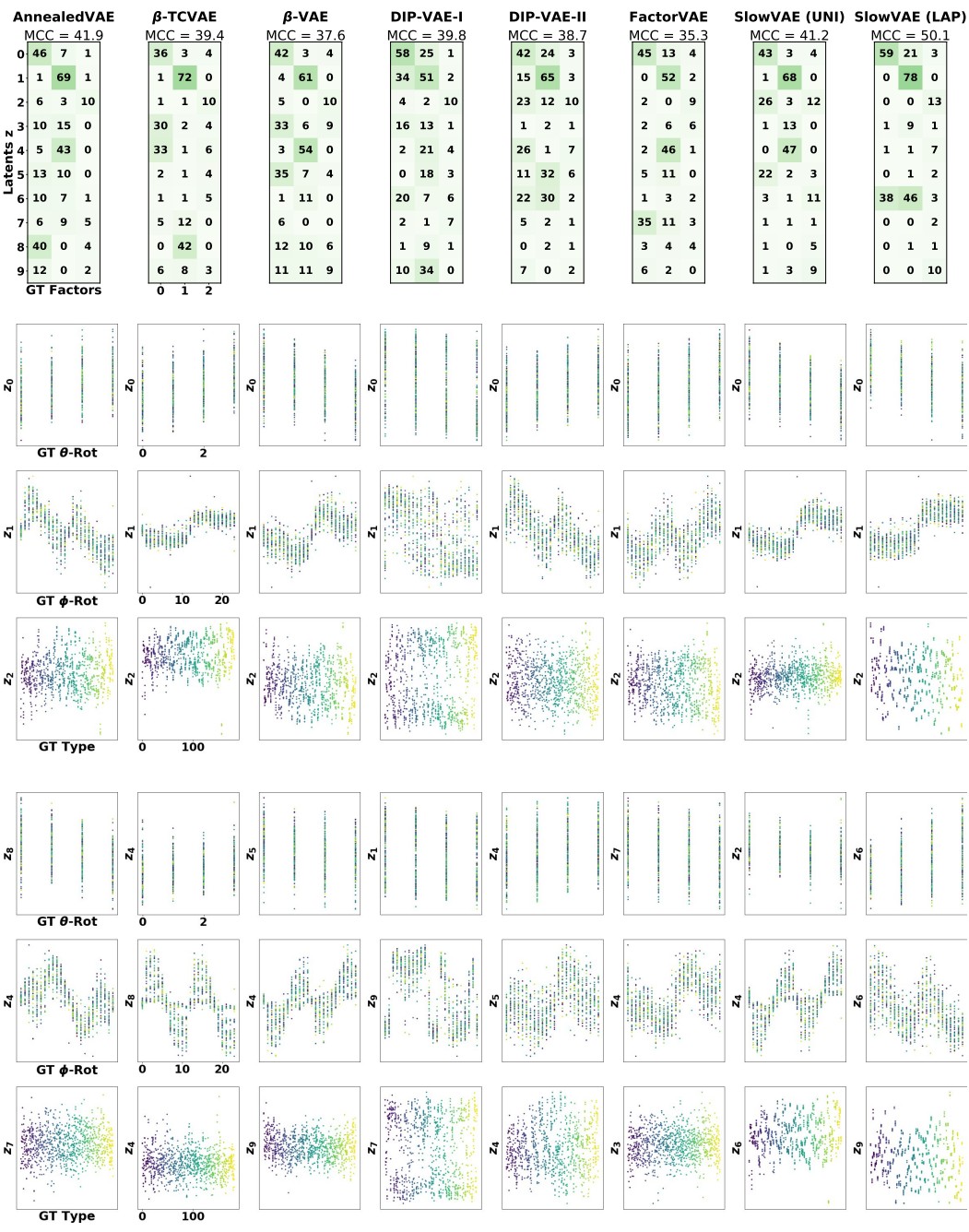

Figure 16: **Cars3D Latent Representations.** Top, MCC correlation matrices. Middle three rows, model latent over highest correlating ground truth factor. Bottom three rows, model latent over second highest correlating ground truth factor. The color-coding corresponds to the 183 different car types (GT Types) in the dataset.

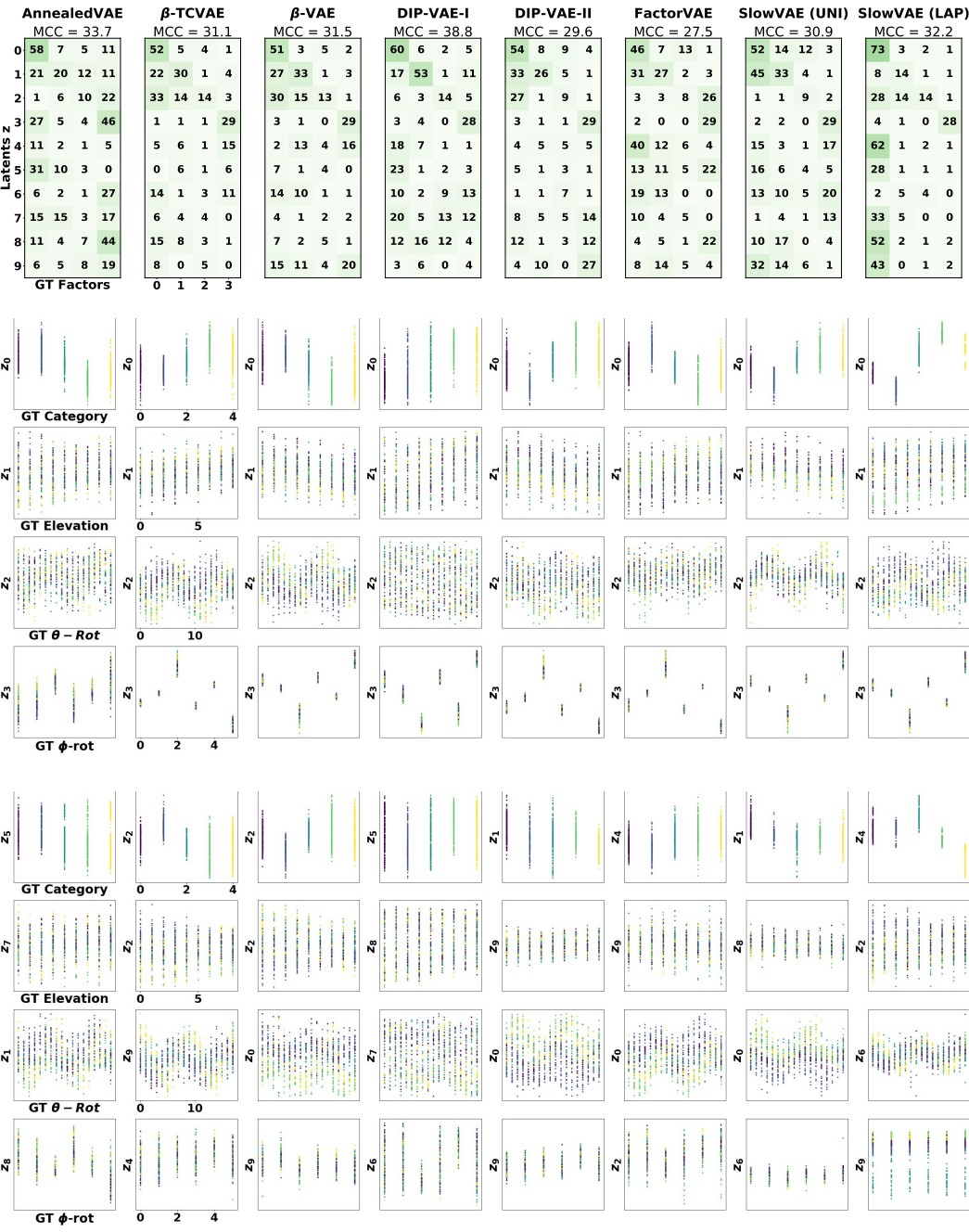

Figure 17: **SmallNorb Latent Representations.** Top, MCC correlation matrices. Middle four rows, model latent over highest correlating ground truth factor. Bottom four rows, model latent over second highest correlating ground truth factor. The color-coding corresponds to the five different GT categories in the dataset.

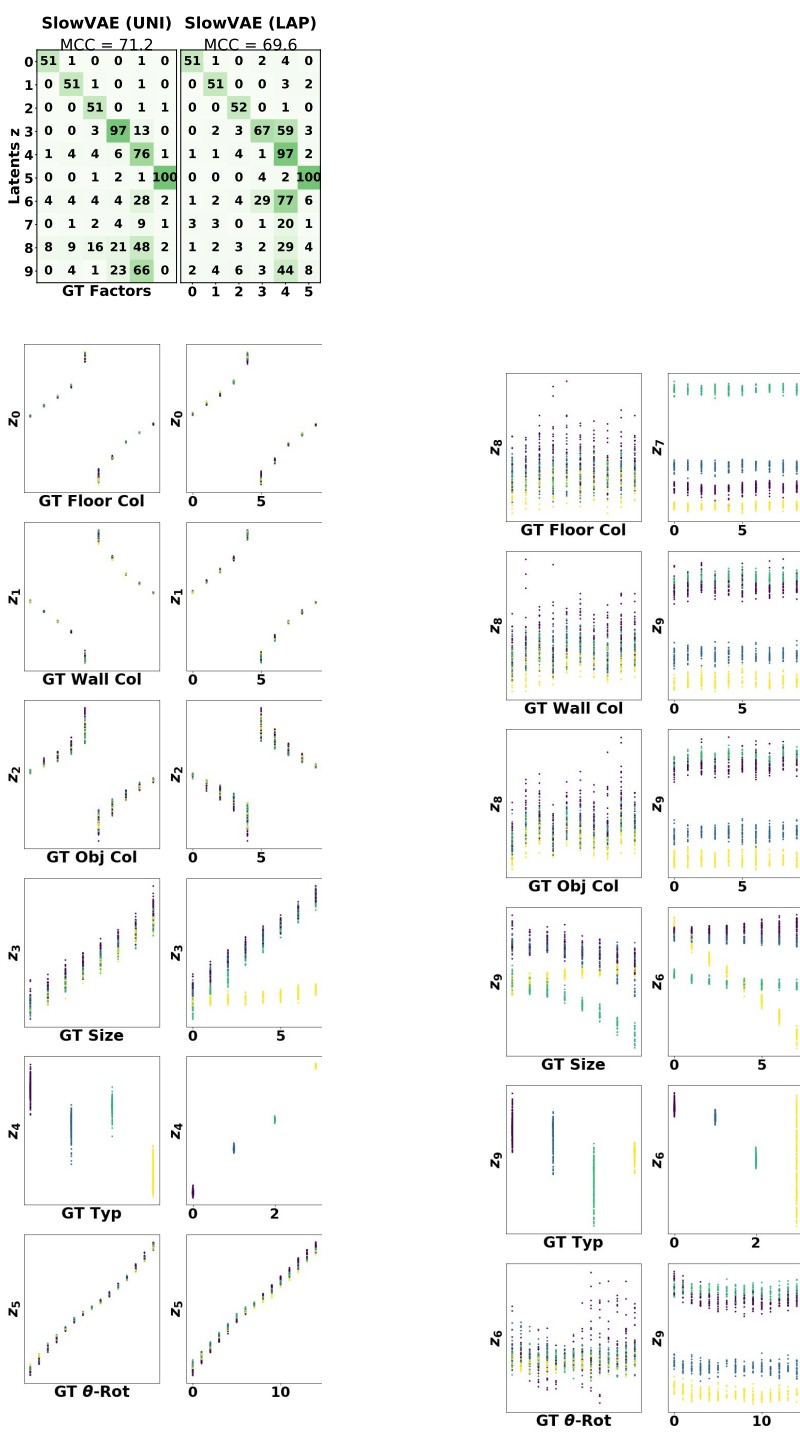

Figure 18: **Shapes3D Latent Representations.** Top, MCC correlation matrices. Left two columns, model latent over highest correlating ground truth factor. Right two columns, model latent over second highest correlating ground truth factor. The color-coding corresponds to the four different object types (GT-Type) in the dataset.

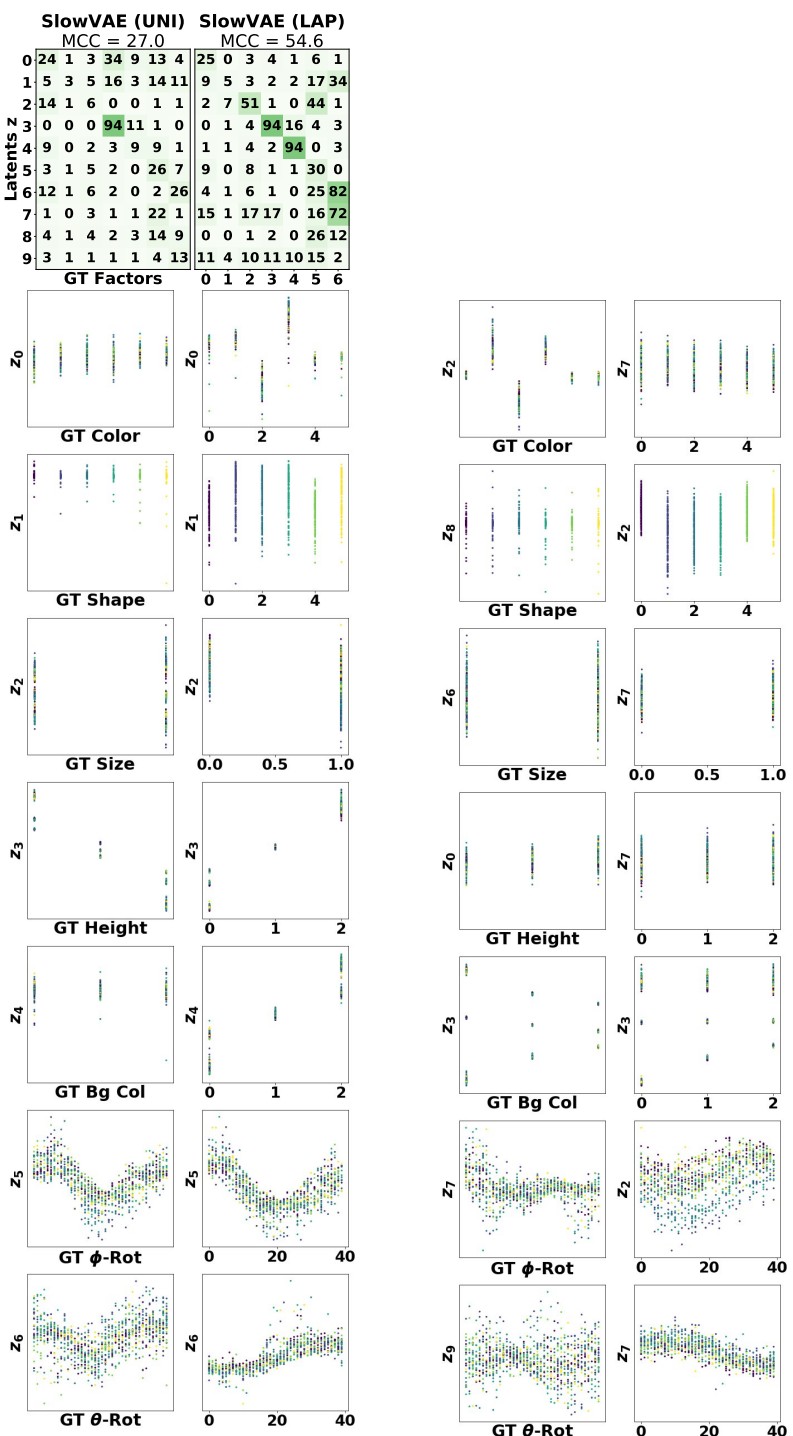

Figure 19: **MPI3DReal Latent Representations.** Top, MCC correlation matrices. Left two columns, model latent over highest correlating ground truth factor. Right two columns, model latent over second highest correlating ground truth factor. The color-coding corresponds to the six different object shapes (GT Shape) in the dataset.

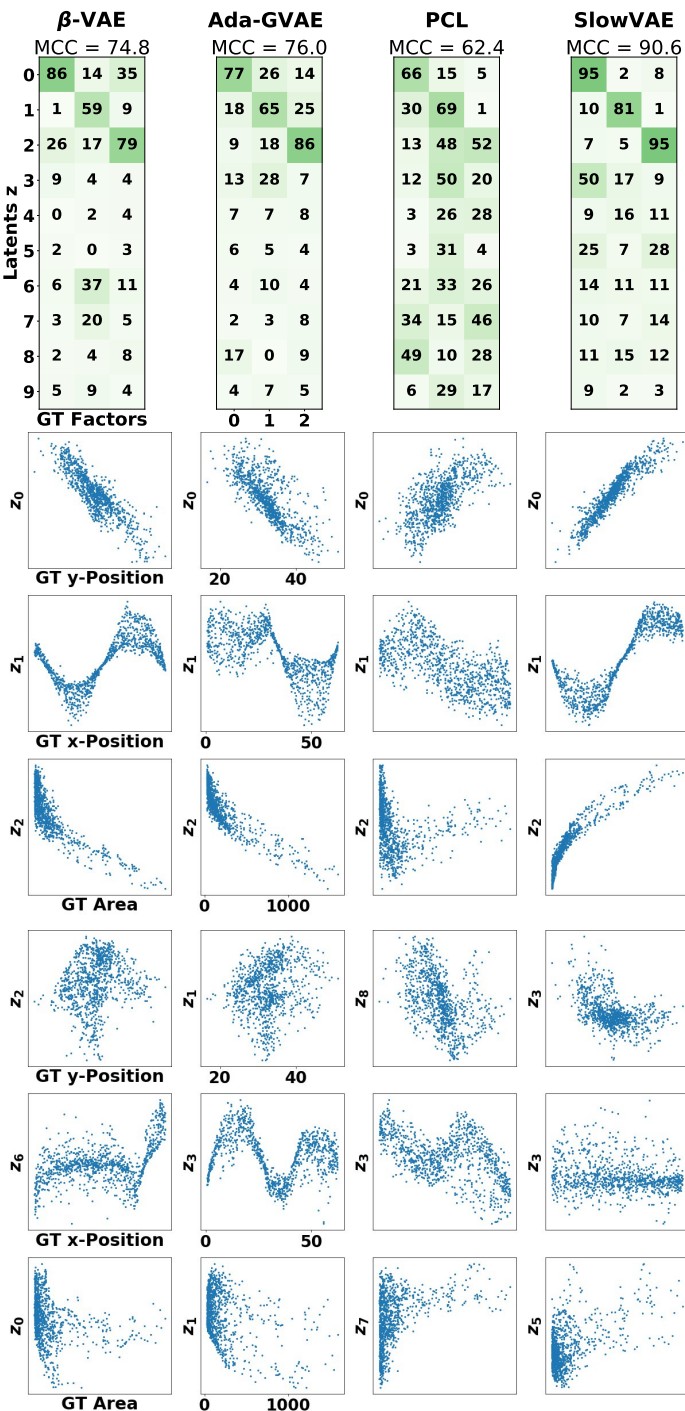

Figure 20: **KITTI Masks Latent Representations.** Top, MCC correlation matrices. Middle three rows, model latent over highest correlating ground truth factor. Bottom three rows, model latent over second highest correlating ground truth factor.

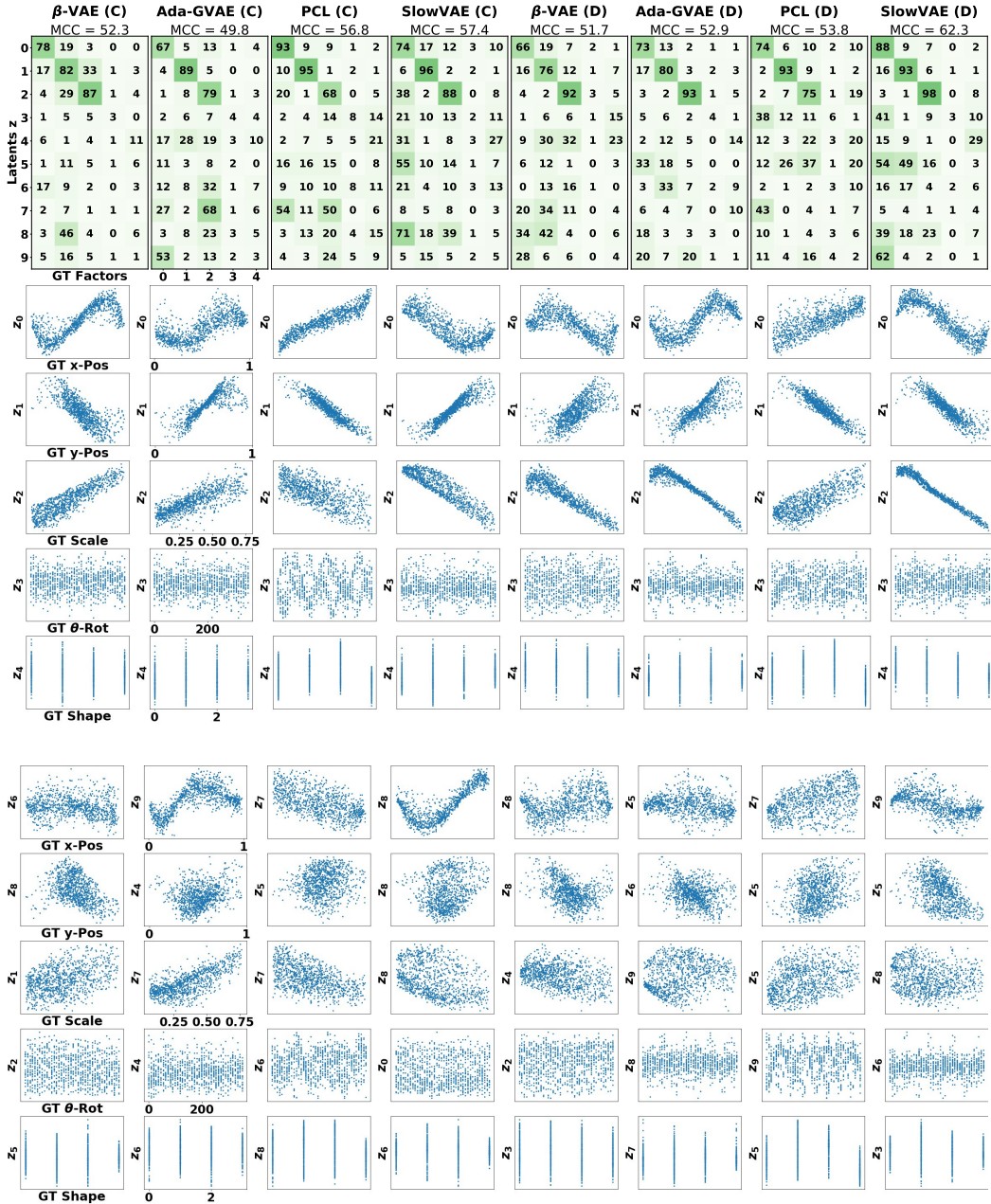

Figure 21: **Natural Sprites Latent Representations.** Top, MCC correlation matrices. Middle five rows, model latent over highest correlating ground truth factor (colored by category). Bottom five rows, model latent over second highest correlating ground truth factor. The left two columns denote the continuous (C) version of Natural Sprites, whereas the right two columns correspond to the discretized (D) version.

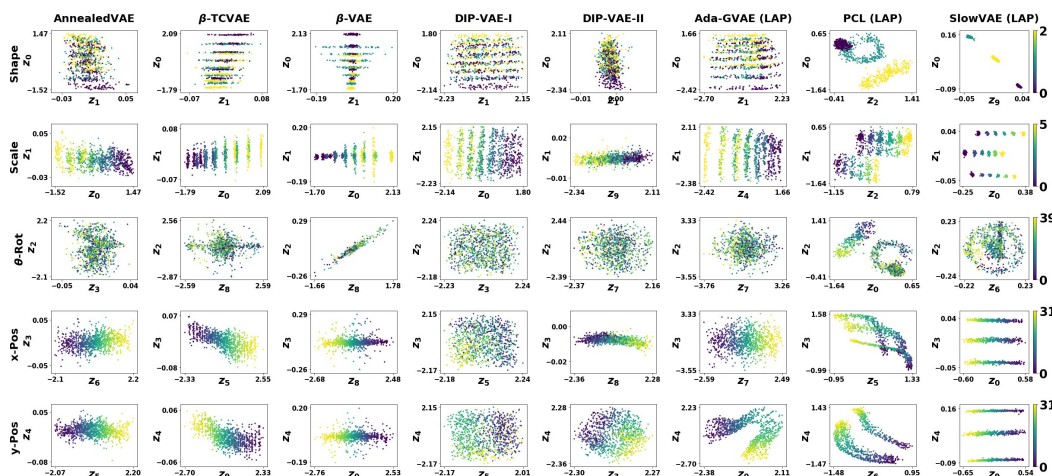

Figure 22: **DSprites Latent Representations**. Best two latents selected from Fig 15. Color-coded by the corresponding ground truth factor.

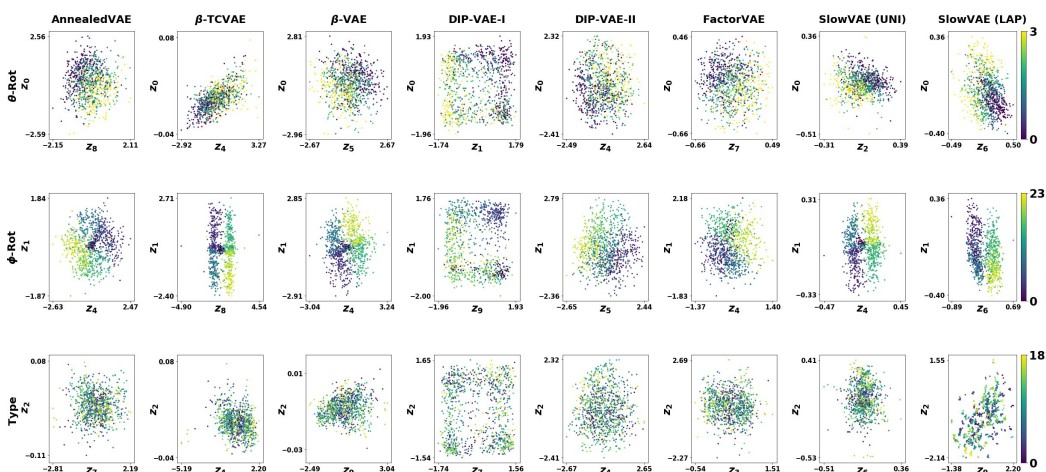

Figure 23: **Cars3D Latent Representations**. Best two latents selected from Fig 16. Color-coded by ground truth.

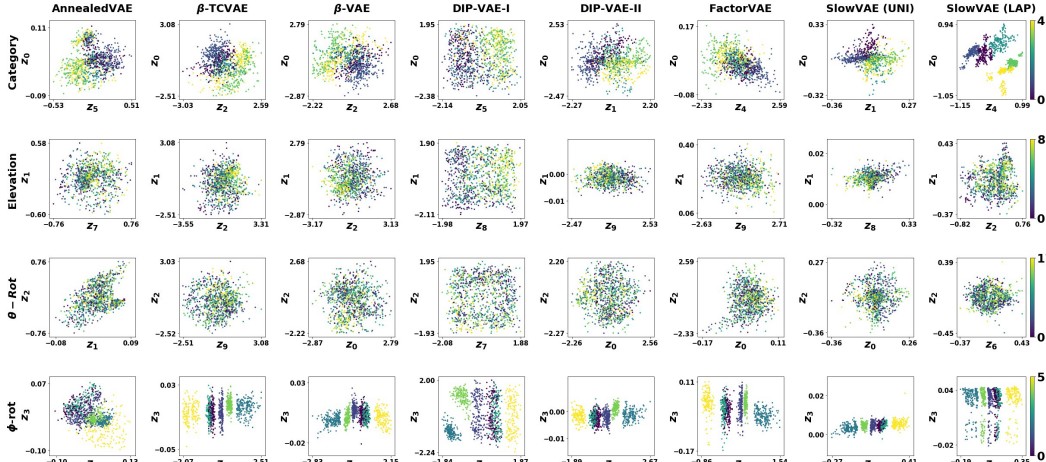

Figure 24: **SmallNorb Latent Representations**. Best two latents selected from Fig 17. Color-coded by ground truth.

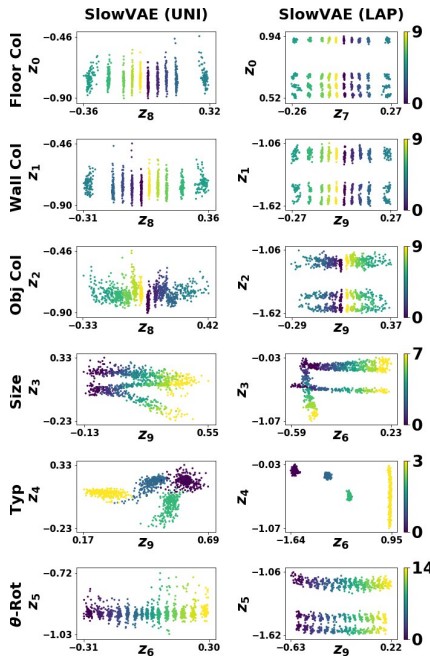

Figure 25: **Shapes3D Latent Representations**. Best two latents selected from Fig 18. Color-coded by ground truth.

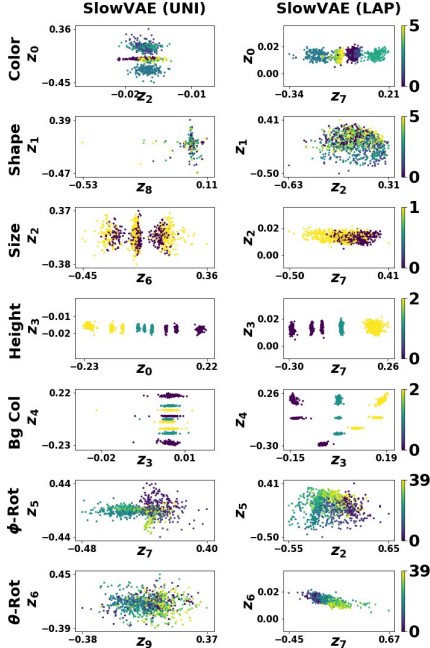

Figure 26: **MPI3DReal Latent Representations**. Best two latents selected from Fig 19. Color-coded by ground truth.

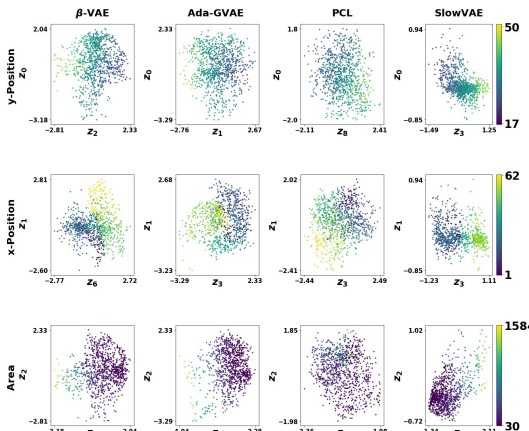

Figure 27: **KITTI Masks Latent Representations**. Best two latents selected from Fig 20. Color-coded by ground truth.

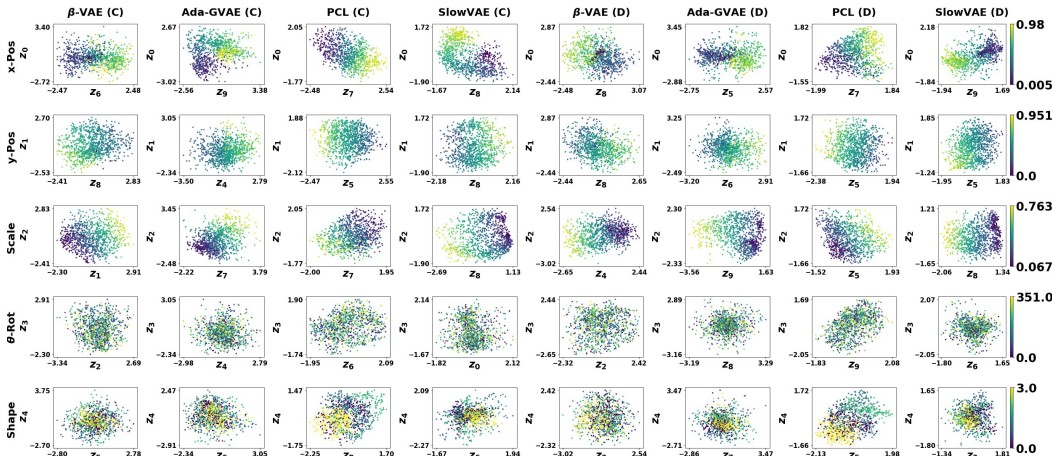

Figure 28: **Natural Sprites Latent Representations**. Best two latents selected from Fig 21. The left four columns denote the continuous (C) version of Natural Sprites, whereas the right four columns correspond to the discretized (D) version. Color-coded by ground truth.

