# OpenReview forum: "Towards Nonlinear Disentanglement in Natural Data with Temporal Sparse Coding"
_ICLR.cc/2021/Conference — ICLR 2021 Oral_

### Official Review · AnonReviewer2 · 2020-10-23
**Very well written and executed papers, good job!**

**Rating:** 9
**Confidence:** 3

**Review:**

The paper starts with an observation that temporal transitions in sequences of natural images are sparse, which is supported by data collected from two big datasets (youtube-vots and kitti-mots). This suggests using a sparse prior for temporal transitions of latent variables when modelling naturalistic scenes. The authors then introduce SlowVAE, a model for temporal independent component analysis (ICA), that depends on such a sparsity prior, and prove that latent variables are identifiable under this model up to permutation and sign flips, which, they say, is stronger than any previous result. Additionally, this work introduces a number of datasets of increasing complexity (and similarity to natural datasets) for testing disentanglement as well as it performs a large scale and detailed evaluation of the introduced model.

The paper is very well written and full of convincing arguments. The related work section (#2) is very well thought-through and extensively describes links to the relevant literature. The model (sections 3.1 and 3.3) is well described--I especially liked section 3.4. which describes assumption made by the theory, and how these assumptions can be violated in practice, which is then followed by extensive experimentation showing that the theory can work even when assumptions are validated. It would be of a great benefit to the community if a similar section was present in other papers.

I concede that I did not understand the proof sketch in sec 3.2 nor the corresponding figure 2, and I did not the full proof in the appendix, therefore I cannot speak to the correctness of the identifiability claim.

I have two remarks about eq. 4.
1) if \gamma != 0, this is not an ELBO, strictly speaking, which is contrary to what the text suggests.
2) We can read that the mean \mu(x_{t-1}) is used as a "single sample" to approximate the last term of that equation. Strictly speaking, this can lead to biased estimates of that term, specifically when the mapping from z_{t-1} to the statistics of p(z_t | . ) is non-linear. Additionally, in high-dimensional distribution, the mode itself is a very unlikely sample. Why is this ok to take the mean in the context of this equation?

The evaluation is extensive (two baselines, one of SOTA from ICA and disentangled representations literature, and 14 datasets). Figure 4, which shows plots of true generative factor vs matched latent variable, is superb. It is a really good method of visualising disentanglement--I agree with the authors that this is a much better way then showing latent traversals.

The paper seems to be very good, though I am no expert on ICA. I strongly recommend acceptance. The reason I am not giving a higher score is that the significance to the community seems not that high, but please correct me if I am wrong.

Some further questions:
1) in sec 3.1. you write "we assume that the noise [in g] is modeled indirectly as a latent variable". Does that mean that g is itself a latent-variable model?
2) The appendix includes a "broader impact" statement, which strongly suggests that this paper was previously rejected from NeurIPS. May I ask what the main points of criticism were?

Some further remarks:
* In the first paragraph of intro you write "although untrue in the literal sense". Why is that? It seems true to me.
* "this problem" a few lines below the above is unclear.
* Table 2 appears BEFORE Table 1 in the text. This is very confusing!
* It is unclear what "Natural" in Table 1 refers to. I assumed that is refers to the introduces "Natural Sprites" dataset, but maybe I am wrong?
* A hyphen in latex should be typed as "---" and there should be no spaces between the hyphen and the surrounding text; you have one e.g. in  the conclusion.

UPDATE: T`he authors' response addressed all my remarks. I've increased the score.

---

> ### Author Response · Authors · 2020-11-17
> **Addressing significance and proof intuition**
>
> Dear Reviewer 2,
>
> Thank you for your positive review of our submission and valuable feedback. We will address your review in full before the end of the rebuttal period. For now, we would like to offer a direct reply to your comment about the significance of our work. We agree that we did not stress the potential impact to the community enough, which we will rectify in the final version. We have also updated our proof sketch for your consideration and additional feedback.
>
> At a high level, disentanglement learning aims to formalize representation learning by directly comparing representations to underlying ground-truth state, as opposed to the indirect evaluation of benchmarking against heuristic downstream tasks (e.g. object recognition). As evidence for its relevance to all of machine learning, we note that at ICML 2019, the best paper award was given to the study of Locatello et al. (2018), which was a large-scale empirical study of disentanglement learning. Additionally, the field of linear and nonlinear ICA has been a core component of unsupervised representation learning for over 30 years, with a bounty of textbooks, highly-cited papers, and applications across a wide variety of engineering and medical fields. Our paper bridges these two important areas of research.
>
> As an unbiased argument for our own paper’s significance, we would like to refer you to comments made by the other reviewers. R1 found our paper “very topical” and “fully agree[d] with the authors that temporal information is essential for representation-learning, and hope this paper can help accelerate the field in that direction.” R4 found that our paper “addresses the important problem of unsupervised learning of disentangled representations” and provides “two new benchmarks of natural and unconstrained data… which would enable future works on more realistic scenarios.” We also note that the reviewers were unanimous in considering the paper “well-written” (R1, R2, R4) or “well-presented” (R3), which we believe will also increase the paper’s impact in the community.
>
> In our own opinion, our work is significant in that it takes a unique approach to solving identifiable disentanglement. Previous work has leveraged temporal information as an additional assumption that leads to identifiability/disentanglement. However, these assumptions do not cover natural video. We found that the observed sparsity in natural transitions is a sufficient assumption for disentanglement, providing the first work, to the best of our knowledge, which proposes a theoretically grounded solution that covers the observed statistics of real videos. Furthermore, we provided a stronger proof that recovers the sources up to permutations and sign flips, as opposed to arbitrary non-linear element-wise transformations. Finally, we provide an empirical contribution by leveraging large-scale natural video databases to provide datasets at the complexity of existing ones, but augmented with natural transitions.
>
> Please let us know whether this changes your assessment of the significance of our work. Regarding the clarity of the proof sketch, we modified both the description and the associated figure to give a better intuition. Please see the image in the link below. It is important to us that this section conveys the intuition to a wide audience, so we appreciate your criticism of its clarity.
>
> https://ibb.co/ZYqyhVg

---

> > ### Comment · AnonReviewer2 · 2020-11-23
> > **Thanks for the clarification.**
> >
> > Given the response, I agree now that the paper is potentially highly significant. The new proof intuition and updated figure both do a much better job at explaining how the proof works. I've increased my score.

---

> ### Author Response · Authors · 2020-11-20
> **Response to additional remarks and questions**
>
> Dear Reviewer 2,
>
> We appreciate your positive assessment of our work and agree that other papers would also benefit from an open discussion on the relationship between empirical study and theory. We provide a point-by-point response to individual concerns below, modulo the ones we covered in our earlier reply.
>
> Regarding the remarks about equation 4: We agree with your first remark and modified the text to make this distinction clear. In terms of the second remark, we agree that using the mean of q(q_{t-1} | x_{t-1}) as a single sample to estimate the expectation in the last term of Eq. (4) leads to a biased estimator. However, in practice we found that such an estimator works well. This is likely related to the observation that the variance of the optimised variational distributions, q(z_{t-1} | x_{t-1}) and q(z_t | x_t), is quite small, and therefore samples from such distributions are similar to their means. We use variational distributions factorized across time steps and dimensions (i.e. factors of variation), and such distributions are known to suffer from underestimated variational posterior variance when using the ELBO as an optimisation objective [1]. Such a setting is common in VAE-related models, since we are typically interested in the mean of the variational posterior distribution rather than in the full distribution. Using variational inference instead of point estimates (which we use in the end) allows us to compute KL terms, which proved to be a useful regularisation. Extending the existing models and applications to take advantage of the full variational posterior is an interesting direction of future work.
>
> Regarding your further questions: 1) g is a function satisfying the assumptions of Theorem 1, which we parametrize with a neural network. We agree that this sentence is confusing and we will revise it. 2) You are correct. The main point of criticism was a lack of additional comparisons, which we rectified in this submission. We also fully rewrote the paper to clarify our message and methods.
>
>
> We agree that it is an interesting philosophical question whether there is a generative model of the visual world. We have updated our text to avoid ruling this out. Regarding your second question, we will update the text to make it clear that “this problem” is referring to decomposing a signal into its underlying factors of variation. We additionally fixed the other minor suggestions you made. Thank you for carefully reading our paper and spotting those.
>
> [1]: http://www.cs.columbia.edu/~blei/fogm/2018F/materials/BleiKucukelbirMcAuliffe2017.pdf

---

### Official Review · AnonReviewer1 · 2020-10-24
**Recommendation to Accept**

**Rating:** 8
**Confidence:** 4

**Review:**


# Summary

This paper introduces a novel VAE-based model with the aim to improve unsupervised disentangling of latent factors in visual data.  This model differs from previous disentangling models in that it takes short (2-frame) videos as input instead of static images.  The model is equipped with a Laplace prior over the dynamics of the video to help it align its representation to axes of sparse temporal dynamics.  The intuition is that the temporal dynamics of natural visual stimuli vary sparsely according to some choice of factors, and that choice of factors is exactly what “disentangled” should refer to.  The authors show that their model achieves better disentangling than previous static-image methods according to a number of metrics.

# Pros

* The interpretation of disentangling as a basis in which the distribution of temporal dynamics of video is sparse is valuable.  Previous approaches to disentangling have been plagued by non-identifiability, and this new interpretation of disentangling is a natural and operational solution.  I fully agree with the authors that temporal information is essential for representation-learning, and hope this paper can help accelerate the field in that direction.
* The paper is clear and well-written.
* The experiments are very thorough in terms of comparisons to previous models and evaluation with previous metrics.
* The application of the Mean Correlation Coefficient (MCC) as a metric for disentangling is good --- I think it is simpler and clearer than many existing disentangling metrics.
* The latent embedding plots are a nice way to visualize latent representations.  While they don’t show what effects non-matched generative factors have on the latent coordinates, they offer valuable information about the latent embedding that is complementary to the commonly used latent traversals.

# Cons

My biggest suggestion is to include an ablation study.  Aside from PCL (which isn’t variational so lives in a different world), there are no existing disentangling models on videos to fairly compare the authors’ model to.  Consequently, it is very important to perform ablation experiments.  More specifically, after reading the paper I have a burning question:  How important is the Laplace prior over transitions?  In other words, can the model work with just a KL regularization between the posteriors for consecutive timesteps?  I think it’s quite possible the answer is “yes” --- simply having a KL regularization instead of the Laplace prior should disentangle better than static-image VAEs because the diagonal posterior will be pressured to align with the transition dynamics.  So I wouldn’t be surprised if the model does quite well with just a KL instead of the Laplace, and that would be a simpler model with two fewer hyperparameters.  So please do this experiment --- regardless of the outcome, the results will be very valuable for readers considering using your model.

Aside from that, I have only a couple minor suggestions:
* Figure 2 is a notationally confusing.  You use z subscripts to indicate time in the lower part of the figure, but in the top part of the figure z subscripts indicate component index.  Maybe make the component index a superscript, or at least make the z in the bottom part of the figure bold so the boldness distinguishes vector from scalar.
* Perhaps make a note that the MCC metric doesn’t work well for discrete latents like shape in dSprites.  For example, in Figure 4 the SlowVAE permutes the shape ordering (e.g. as compared to PCL) and gets a low MCC score for that, but that low score is a drawback of the metric, not the model.  This isn’t unique to MCC --- most other metrics (except MIG) would fall for this too.  But perhaps for discrete latents (ones where we don’t care about the ordering of a discrete set of values), MCC can take the highest over all permutations.  I don’t think it’s necessary to do this, but perhaps you can add a sentence that mentions it so readers understand why the shape score is low.

# Conclusion:

Overall, I recommend this paper to be accepted.  It is very topical, since disentangling has recently been receiving increasing attention, and by incorporating temporal cues into disentangling in VAEs this paper could help steer the field in a productive new direction.  I only hope the authors do the suggested ablation experiment (I think practitioners would appreciate those results).

---

> ### Author Response · Authors · 2020-11-20
> **Ablation results and suggested revisions**
>
> Dear Reviewer 1,
>
> Thank you for your review and feedback. We very much appreciate your assessment of our work as a "topical", “valuable”, and "well-written" paper. We address each of your suggestions for further improvement in turn below.
>
> As comparison models, you correctly remarked that PCL isn’t variational, which makes it somewhat dissimilar. However, we also compare to Ada-GVAE (Locatello et al. 2020) which is VAE-based, directly applicable to videos and, therefore, a fair comparison.
>
> Nonetheless, we found your recommended ablation experiment extremely valuable. We ran a new experiment with a control model that, as you suggested, replaces the Laplace prior with the posterior of the previous time step. Hence, we minimize KL(q(z_t)|q(z_{t-1})), i.e. a KL-divergence term between the posterior at time-step t and time-step t-1. We termed the model Posterior Matching-VAE (PM-VAE) (see these links for results tables: https://ibb.co/Xx7Z5hH, https://ibb.co/kDwZQCg; note mean delta_t=0.05s corresponds to delta_t=1 in the original submission, as well as mean delta_t=0.15s corresponds to delta_t=5 in the original submission; this update in labelling was made for clarity and will be thoroughly explained in the paper revision, which will be released soon). We report several hyperparameter values for PM-VAE and also note that we continue to use the same hyperparameter values for SlowVAE as in all other experiments. Interestingly, this setting is equivalent to a Gaussian transition prior (with variances in each direction scaled by the posterior of the previous time step), i.e. alpha=2, which, according to our theory, should not be identifiable. Confirming these predictions, we found that the disentanglement scores were overall reduced for the datasets we tested, which include all natural datasets. We will include this result (for all datasets and metrics) among our extended results in the appendix, as it covers a valuable control condition.
>
> Thank you for pointing out the notational inconsistency in Fig. 2. It is important to us that the proof sketch and figure conveys the intuition to a wide audience, so we appreciate your scrutiny. We updated the figure, as well as the associated text. Please follow this link (https://ibb.co/ZYqyhVg) to see the revision.
>
> We also added an extended discussion on metrics to the manuscript, which highlights the issue with categorical variables that you mentioned. We note that properly evaluating disentanglement is an ongoing area of research, with notable results from Higgins et al. (2019) and two submissions to this ICLR submission cycle (https://openreview.net/forum?id=YZ-NHPj6c6O, https://openreview.net/forum?id=cbdp6RLk2r7).
>
>
> Again, thank you very much for reviewing our paper and for your valuable suggestions!

---

### Official Review · AnonReviewer4 · 2020-10-28
**Generally good paper on disentanglement in more natural data, with extensive evaluation.**

**Rating:** 9
**Confidence:** 4

**Review:**

Summary:

The paper addresses the problem of disentangling the underlying generative factors from data with a particular focus on dynamic natural data. It provides evidence that transitions of objects in natural movies can be characterised by temporally sparse distributions. A novel proof based on a sparse prior on temporally adjacent observations is provided, allowing to recover true latent variables up to permutations and sign flips, and improving the disentangling performance over existing methods. Two new datasets with measured natural dynamics are also proposed to enable evaluation on more realistic scenarios.

##################################################################

Strengths:
- The paper addresses the important problem of unsupervised learning of disentangled representations. Overall, it is well written and easy to read.
- A new framework of non-linear ICA is introduced, showing evidence to the hypothesis that natural scenes have sparse temporal transitions.
- Two new benchmarks of natural and unconstrained data are presented which would enable future works on more realistic scenarios.
- The paper provides extensive experiments and comparisons to relevant baselines, including both qualitative analysis and quantitative results using different metrics.

##################################################################

Weaknesses:
- The existing datasets present in DisLib are considered unnatural since they assume data is i.i.d and define a prior on the number of factors to be changed. The datasets presented in this work represent incrementally more realistic scenarios where sparse transitions are imposed in addition to data with natural continuous generative factors and data with transitions from unstructured natural videos. In the Laplace transitions dataset, the rate of the Laplace prior λ is sampled from a uniform distribution λ ∼ U (1, 10) which may result in changes in many factors (when λ is small). Although this allows to have fair comparison with Locatello et al., (2020), this setting may sometimes result in non-sparse transitions between image pairs which is not aligned with the statistics of natural transitions as discussed throughout the paper. This would not be expected if one does not read Appendix 4. Hence, it would be good to clarify this particular point in the main paper.
- In the Natural Sprites dataset, pairs are produced by only varying the position and scale (with real transitions extracted from the YouTube measurements) while color, shape, and orientations are fixed. While fixing the color and shape can be understood to follow natural transitions of objects, it is unclear why the orientation is not varied in this case?
- For the Kitti masks dataset, continuous natural transitions are considered in all underlying factors. Table 2 shows a clear improvement when the temporal distance between sampled frames ∆t = 5 compared to ∆t = 1. What would be the effect of further varying this parameter?
- From Figure 4, authors raised up an interesting observation for the rotation factor, where SlowVAE shows three sinusoidal oscillations with different frequencies matching the three distinct rotational symmetries of the shapes present in the dataset. However, from the MCC measurements, we can notice that all methods including SlowVAE struggle in disentangling the rotation factor. Here, it would be interesting to further discuss this limitation.

##################################################################

Final update: Authors addressed all my comments in the rebuttal making significant improvements to the revision. I've increased my score.

---

> ### Author Response · Authors · 2020-11-20
> **Varying ∆t and addressing additional points**
>
> Dear Reviewer 4,
>
> Thank you for your careful review and feedback. We appreciate your praise of the paper being “well written and easy to read” as well as your acknowledgement of the importance of the problem we address. Please find below responses to each of the points you suggested for further improvement.
>
> Your question regarding the Delta-t parameter in KITTI Masks was very relevant. The definition of Delta-t in the manuscript was not clear. We have changed the variable name to max(Delta-frames)=N, which indicates that all pairs differ by at most N frames. Choosing an upper bound of N was done because the frame rates and sequence lengths in the KITTI dataset were variable. We have added a figure (https://ibb.co/hgz0n5T) to illustrate the relationship between max(Delta-frames) and Delta-t, in seconds. We changed all of the remaining figures and tables to be with respect to the mean time between consecutive frames in seconds (mean Delta-t). We ran the experiment you requested for a large range of mean Delta-t values. We found that model performance increased initially with larger temporal separation between data points, then plateaued (https://ibb.co/5Wfx5Pn). We also tested the temporal sparsity with increasing frame spacing and noticed an increase in alpha, but that it remained sparse (alpha < 1, https://ibb.co/9HgH43p).
>
> We agree that we did not motivate our design choice for fixing orientation in the Natural Sprites (NS) dataset well enough. We have corrected this in the paper. In summary, it is unclear how to accurately estimate object orientation from the masks (we tried measuring orientation from the principal directions of the mask but encountered a variety of failure cases). We also did not want to introduce artificial transitions in the NS dataset. Therefore, we decided to fix the orientation across transitions.
>
>
> The question of proper metrics for disentanglement has been top-of-mind for us and also Reviewer 1, who noted the failure of MCC to properly address categorical variables like shape. We do agree that it is important to bring awareness to the shortcomings of current metrics. To this end, we extended our discussion on the matter in the revised manuscript.
>
> Finally, thanks for pointing out the confusion with the lambda parameter in the LAP dataset. This phrasing was accidentally left in the paper from an earlier draft. In the current work, we use the UNI dataset to directly compare to Locatello et al., 2020. This alleviated the need to draw lambda from a uniform distribution, and so all of the experiments in the paper were performed with lambda=1. The lambda parameter changes the scale, not the distribution itself. In other words, the sparsity is unchanged, as the sparsity is controlled by the shape (alpha), but the scale of movements is changed. We will correct this in the revised manuscript.

---

### Official Review · AnonReviewer3 · 2020-10-28
**Does the motivation hold up more generally?**

**Rating:** 7
**Confidence:** 3

**Review:**

This paper introduces the SlowVAE to model transitions (of position, size, etc) in single-object videos. A Laplace prior conditioned on the latents at the previous step is used to learn the transitions, which the authors argue are naturally sparse.

My biggest concern regarding this work is the construction of datasets: the authors effectively “consider pairs of images [x_{t-1}, x_t], which differ effectively in only a few factors of variation due to the sparse prior.” While the marginal distributions of factor transitions may no doubt be sparse in natural settings, I would guess the sparsity is not independent across factors. For example: a video where a camera moves sporadically will show the position factors and size of an object changing sparsely but simultaneously. Could you augment Fig 1 and appendix H to show the joint distributions in addition to the marginals?

Moreover, the LAP procedure has the following rejection rule: “if all factors remain constant (no transition), the sample is rejected as the pair would not result in any temporal learning signal.” Surely there needs to be some static pairs if you want to be close to natural videos?

Given KITTI is the only dataset you show results on whose transitions are natural (albeit not 3D-natural; see caveat in question 2 below), it would be nice to see more evidence how the SlowVAE would generalize and cope in scenarios different from the ones you’ve constructed. If the LAP procedure indeed samples from marginal transition statistics rather than the joint distributions, I don't believe the samples would resemble "natural transitions". Learning disentangled representations from data which is engineered to show changes in one generative feature at a time seems like cheating because we usually don't have easy access to such data (unless an agent actively interacts with an object).

Strengths:
- Good literature review
- Comparison to relevant SOTA models, linking to prior work on disentanglement as well as nonlinear ICA, on a variety of metrics.
- Solid experiments and well-presented results

Questions:
1) Have you tried working with sequences rather than pairs of images? I assume your ELBO can be readily extended to sequential data. Are there additional challenges in working with sequences?
2) Do you think there's room for bias in the natural statistics you computed on Youtube-VOS and KITTI-MOTS? For instance, from the fact that object masks are projections of 3D objects onto 2D frames? Consider the bicycle example (row 1) at https://youtube-vos.org/. In 3D, the position of the bicycle changes smoothly as the cyclist carries it. But in 2D there is a noticeable jump in the center of mass of the bicycle's mask (relative say to the body of the rider). Could this possibly explain why you found a heavier-tailed Laplace distribution to be a good fit for the transitions in Youtube-VOS and KITTI-MOTS?
3) How do natural transition statistics look like in multi-object datasets? How do you envision extending your work to tackle those?
4) In 3.3, you suggest alpha = 1 helps break the rotational symmetry of the ELBO by making axis-aligned representations optimal. Could you substantiate this claim? Theorem 1 does not immediately suggest an identifiability advantage for alpha = 1.
5) On the Natural Sprites dataset, why is it that discrete transitions give SlowVAE an advantage over PCL (MCC score 52.6 versus 50.2 and all other metrics) whereas continuous transitions are disadvantageous (MCC score 49.1 versus 51.7)? Why work with discrete transitions at all?

---

> ### Author Response · Authors · 2020-11-12
> **Clarification concerning dataset construction and new results on joint dependencies**
>
> Dear Reviewer 3,
>
> Thank you for your extensive review and valuable feedback. While we will address your review in full before the end of the rebuttal period, we wanted to address some specific points you raised now to allow time to discuss additional feedback that you might have. Here, we will focus on what you named as your biggest concern: the construction of the datasets. We believe part of the concern may be because we did not completely clarify how we constructed them.
>
> You state that "learning disentangled representations from data which is engineered to show changes in one generative feature at a time seems like cheating”. We fully agree with this, which is what motivated us to extend previous work that used such artificial transitions (Locatello et al., 2020) by incorporating information from natural video data, namely with Natural Sprites (NS) and KITTI Masks (KM).
>
> YouTube-VOS is a previously published dataset (Xu et al. 2018; Yang et al. 2019) with 4,883 unique video instances computed from YouTube videos. Figure 1 shows YouTube-VOS transition statistics, which is aggregated across all 4.8k unique object instances. Therefore, our statistics are aggregated across unique objects in a multi-object dataset (addressing the first part of your question #3). Note that this applies similarly to KITTI-MOTS. NS is constructed using the position (x, y) and scale information measured from YouTube-VOS masks, and thus would exhibit any natural statistical structure present for those factors.
>
> You noted the importance of considering the joint dependencies among natural generative factors. We agree that it is valuable to know both the extent of these dependencies and how much they impact our empirical disentanglement performance. To answer this, we constructed modified datasets where time-pairs of factors are shuffled per-factor (e.g. combining the x transition from one clip with the y transition from a different clip). This destroys dependencies between the factors, while maintaining the sparse marginal distributions. In the attached figure (that we also added to the manuscript), we show 2D marginals before (blue) and after (orange) this shuffling. The additional density on the diagonals in the unshuffled data reveals dependencies between pairs of factors on both datasets. This confirms your intuition that the observed dependency is mismatched from the theoretical assumptions of our model. Per your suggestion, we tested how robust SlowVAE is to such a mismatch by training it on the shuffled data and re-evaluating disentanglement. We found our model performed better when trained with the shuffled data than with the original data, as expected since shuffling rectifies the mismatch (see results below, mean and s.d. across 10 random seeds). However, the gain is limited in size (it isn’t statistically significant, independent T-test, p<0.05), which is in line with our discussion on empirical performance despite theoretical violations.
>
> Figure here: https://ibb.co/R9WrPkw
>
> SlowVAE MCC on permuted Natural Sprites (Continuous): 52.9 (4.2)
>
> SlowVAE MCC on unaltered Natural Sprites (Continuous): 49.1 (4.0)

---

> > ### Comment · AnonReviewer3 · 2020-11-18
> > **Concern regarding UNI and LAP-sampled datasets holds**
> >
> > Thanks for your quick turnaround to allow for discussion. The (slightly) increased MCC score on permuted Natural Sprites and your plots of pairwise natural transitions do lend credibility to the concern that SlowVAE's disentanglement hinges on observing transitions in generative factors independently (e.g. one or a few at a time). Here is my full argument:
> >
> > From my reading, SlowVAE does no better than baseline models on natural transitions from Youtube-VOS (which are only deployed in Natural Sprites). Is that fair to say? I do appreciate there is a performance gain on KITTI Masks; the top-3 latent traversals do look great. But there isn't enough information in the results on KITTI Masks at the moment (e.g. comparisons with PCL and Ada-GVAE) to make a compelling case for SlowVAE. Beta-VAE isn't a fair baseline for a temporal model. Moreover, there is only one score and one number (no average over different runs) in Figure 5.
> >
> > That prompts the conclusion that to show a performance gain, you are relying on independently sampling transitions (per factor) via the UNI and LAP sampling schemes. Your results on SmallNORB and Cars3D prompt the same conclusion. (Could you also clarify what's the value of k i.e. the number of factors which change simultaneously in your UNI sampling for dSprites? And I assume LOC == UNI in Tables 15 and 16?)
> >
> > Unfortunately, the independence assumption in the UNI and LAP sampling schemes is totally inadequate when combined with sparse marginal distributions like the Laplace, an upper bound on how many factors can change at the same time, or a prohibition on static objects (i.e. no transition). These assumptions lead directly to the cheating via demonstration we discussed earlier.
> >
> > To make a convincingly case for SlowVAE, I would advise doing at least one of the following:
> > - focus on natural experiments (and perhaps leave all artificial-transition datasets for the appendix). Maybe you could apply Youtube-VOS transitions to MPI3D?
> > - add ablations to justify your model, especially the choice of prior (e.g. add comparisons with α = 1 and 2) and the effect of parameters like γ and λ
> >
> > I look forward to reviewing the effectiveness of SlowVAE independently of the artificial-transition datasets.

---

> > > ### Author Response · Authors · 2020-11-20
> > > **Results from requested experiments and clarification of contributions**
> > >
> > > Dear Reviewer 3,
> > >
> > > We understand your concerns and share your desire to see more progress towards real data. From this discussion, we have identified two important challenges when moving from the standard i.i.d. data in disentanglement research towards more natural data: modeling the sparse marginal transition distributions; and modeling the complex dependencies between natural factors. The purpose of our work was to focus on the former. We believe that this constitutes an important advance that sets the stage for solving further issues and eventually applying the technology to real-world problems. We acknowledge that our theory will have to be extended to deal with dependent factors, which we touch upon with empirical comparisons on the two proposed natural datasets and our permutation experiment (see below). We will add a disclaimer to our introduction to make this nuance more clear.
> > >
> > > Nevertheless, we have existing and new experiments that already shed some light on the challenge of modeling natural dependencies. We hope that this addresses the biggest concerns from your first and second responses:
> > >
> > > 1) _Natural dependencies across factors are not considered in our work._ Natural Sprites and KITTI datasets use transitions measured from natural videos, and thus any scores reported on those will reflect performance under natural dependencies. In Table 2, we do provide comparisons to PCL and Ada-GVAE on KITTI Masks, demonstrating that as we increase maximum temporal separation between frames, SlowVAE outperforms baselines with statistical significance. We also updated Figure 5 to show Ada-GVAE and PCL instead of beta-VAE (https://ibb.co/D53kpFh).
> > >
> > > 2) _SlowVAE's disentanglement hinges on observing transitions in generative factors independently._ We would like to highlight that the improvement of SlowVAE on the permuted (i.e. independent) continuous Natural Sprites was not significant (https://ibb.co/FgtRV8p). Moreover, when we performed the same experiment on discrete Natural Sprites (where we can measure standard disentanglement metrics), we found an overall improved score with non-permuted transitions (i.e. with dependencies), with 3/7 metrics showing a significant improvement (https://ibb.co/mJqS8FX). Our conclusion is that these preliminary results do not support the hypothesis that SlowVAE’s disentanglement depends on independent factors.
> > >
> > > You suggested a further experiment of setting alpha = 2 that would make a convincing case for SlowVAE. We replaced the Laplace prior with a KL-divergence term between the posteriors at time-step t and time-step t-1, which is equivalent to a Gaussian (alpha = 2) transition prior. We termed the model Posterior Matching-VAE (PM-VAE). As our theory predicts, disentanglement scores for this ablated model were overall reduced (https://ibb.co/Xx7Z5hH, https://ibb.co/kDwZQCg). We report several hyperparameter values for PM-VAE and also note that we continue to use the same hyperparameter values (selected without supervision) for SlowVAE as in all other experiments.
> > >
> > > We agree with you that the rejection rule in the LAP procedure is unnatural. As such, we have been retraining our networks without this modification. We do not find any significant difference in performance (preliminary results, note LAP-NC refers to no rejection rule: https://ibb.co/Qc3MjyW, https://ibb.co/pPFR26g, https://ibb.co/TvGPyVY). To lend more credibility to the construction of our LAP extension, we additionally sampled 10k transitions from each dataset with LAP transitions and computed the number of factors that had changed within a pair, explicitly showing most factors are changing in each transition (https://ibb.co/n8P5bRM). Therefore, we are not artificially only varying a single factor per transition (i.e., “cheating”). Quite the opposite, the LAP DisLib extension provides a bridge from i.i.d. data to natural data by explicitly modeling the observed sparse marginal transition distributions. Therefore, we firmly believe that the LAP dataset is an important step from previous datasets towards natural data and should not be dismissed.
> > >
> > > To summarize our claims: We present marginal (and now joint) statistics of natural video transitions and bring attention to the sparse marginals. Informed by this observation, we present a novel proof of identifiability that is stronger than previous results. We incorporate natural video transitions into datasets that are otherwise comparable to existing work to provide a valuable step towards disentanglement on natural video. Our implementation of our theory, SlowVAE, performs at or above state-of-the-art, even when the data does not match the model assumptions. Future work will have to investigate further the complex dependencies between natural factors.

---

> ### Author Response · Authors · 2020-11-21
> **(pt 1 of 2) Response to remaining remarks and questions**
>
> Dear Reviewer 3,
>
> In addition to the discussion in the other thread, we wanted to follow up with responses to each of your remaining questions and remarks:
>
> Q1 _Extension from pairs to sequences_: We have chosen to work with pairs of inputs as minimal sequences because we are interested in the first temporal derivative, more specifically in the sparsity of the transitions between pairs of images. Other methods that look at the second temporal derivative, such as work from Hénaff et al. [1] on straightening, would require triplets as minimal sequences. Extending our approach beyond this minimal requirement would, as you correctly stated, be simple in terms of the resulting ELBO (which would still factorise like in Eq. 4 because of the Markov property). The only additional complexity would be in the data and loss handling.
>
> Q2 _Potential bias of 3D to 2D projection_: Thank you for bringing this to our attention. We found the question compelling from a broader perspective of visual processing. Unfortunately, the KITTI-MOTS masks lack the associated depth data required to answer your question. Although we do think it is a compelling question in terms of natural scene statistics, we will have to reserve answering it for additional study in future work. Nonetheless, the statistics we compute are still relevant, given that most computer vision models and vision-based animals see the 3D world as projected onto their 2D receptor arrays.
>
> Q3 _Multi-object datasets_: We addressed the first part of this question in our earlier reply, namely that Youtube-VOS and KITTI-MOTS are multi-object datasets, although we consider each unique object (mask) separately. For the second part, multi-object representation learning and disentanglement are highly connected, in fact they have recently begun to be used interchangeably [2]. To briefly comment on possible extensions in this direction, we see no reason why our prior would not be beneficial to multi-object methods such as MONet [3] and IODINE [4]. We find this to be an interesting avenue for future work.
>
> Q4 _Further clarification about alpha = 1_: You are right, choosing alpha = 1 does not provide any additional identifiability advantage in comparison to any other alpha < 2. Theorem 1 guarantees identifiability if the data is generated by the ground-truth generative model (Eq. 2) for any alpha < 2 and the model’s alpha matches the ground-truth. As can be seen in Figs. 1 and 9, alpha ~= 0.5 provided the best fit to the ground-truth marginals. However, we chose alpha = 1 for SlowVAE, since it allowed us to derive a simple closed-form solution for the KL divergence in the ELBO. In the referenced paragraph in Section 3.3, we wanted to emphasize that choosing alpha = 1 allows for identifiability (as any alpha < 2), but it might be suboptimal in terms of a model mismatch between the ground-truth model and the VAE. We have revised the paragraph to make this point clear.
>
> Q5 _Discretisation of Natural Sprites_: The continuous NS has similar complexity to dSprites in terms of dimensionality and object shape while including natural transitions. Discretizing NS yields equal complexity in terms of the number of possible object states, but collected into pairs that exhibit natural transition statistics. This helps isolate the effect of including natural transitions from the effect of increasing data complexity.
>
> [1]: https://www.cns.nyu.edu/pub/lcv/henaff18-reprint.pdf
> [2]: https://arxiv.org/abs/2011.01758
> [3]: https://arxiv.org/abs/1901.11390
> [4]: https://arxiv.org/abs/1903.00450
> [5]: https://arxiv.org/abs/1811.12359
> [6]: https://arxiv.org/abs/2002.02886
> [7]: https://github.com/google-research/disentanglement_lib/blob/master/disentanglement_lib/methods/weak/train_weak_lib.py#L48
> [8]: https://github.com/deepmind/spriteworld
> [9]: https://arxiv.org/abs/1905.12614

---

> ### Author Response · Authors · 2020-11-21
> **(pt 2 of 2) Response to remaining remarks and questions**
>
> * The reason we report a single score in Fig. 5 is because the MCC matrix is only displayed for a single model, namely the best-performing model (the seed with maximum average score across evaluated metrics), for space considerations. We have updated the text to make this more clear. Per your suggestion, we swapped the beta-VAE model with the best-performing PCL and Ada-GVAE models in Fig. 5. We note that there are also PCL and Ada-GVAE visualisations in Figs. 4,11-24. We provide means and standard deviations for evaluation of 10 random seeds of SlowVAE, PCL, and Ada-GVAE in table 2.
>
> * Our UNI dataset variant is using the k = rand condition, so the number of factors is uniformly sampled from 1 to D-1, where D is the total number of factors. UNI is modeled after the scheme used in Locatello et al., (2020) [6], with key differences being (i) their code randomly (with 50% probability) sets k = 1 even in the k = rand setting [7] and (ii) we ensure that exactly k factors change. As such, they should be considered comparable but not exactly the same. We should also note that we do not use bolding to denote significance in those tables because some of the numbers are reported from Locatello et al., (2020) [6], and thus we can't run the significance test.
>
> * Unfortunately, we are limited in the available states for all of the DisLib datasets. As such, any application of naturally recorded statistics to a DisLib dataset (e.g. MPI3D, as you suggested) would introduce error corresponding to discretizing the natural measurements into the available states. This is why we constructed Natural Sprites using Spriteworld [8], which allowed us to render entirely new sprite objects at the precise position and scale as was measured from YouTube. This option is unavailable without the renderer for MPI3D.
>
> * SlowVAE uses α = 1, and we have provided a comparison to α = 2 in our previous response. Regarding varying the gamma and lambda parameters, Locatello et al. (2018) [5] showed that model performance varied highly with respect to hyperparameters. As such, we felt that a more appropriate decision was to select a single model parameter set (with UDR [9], which is unsupervised) for the entire study. This puts us at a disadvantage to comparison models (e.g. Ada-GVAE parameters were selected (without supervision) per dataset, as recommended by Locatello et al. (2020) [6], details will be clarified in sec. 5.2), but we nonetheless consider it a more appropriate method for benchmarking model performance.
>
> We made all other corrections you suggested and hope that we have addressed all of your points. Thank you again for providing a detailed review of our work.
>
> [1]: https://www.cns.nyu.edu/pub/lcv/henaff18-reprint.pdf
> [2]: https://arxiv.org/abs/2011.01758
> [3]: https://arxiv.org/abs/1901.11390
> [4]: https://arxiv.org/abs/1903.00450
> [5]: https://arxiv.org/abs/1811.12359
> [6]: https://arxiv.org/abs/2002.02886
> [7]: https://github.com/google-research/disentanglement_lib/blob/master/disentanglement_lib/methods/weak/train_weak_lib.py#L48
> [8]: https://github.com/deepmind/spriteworld
> [9]: https://arxiv.org/abs/1905.12614

---

### Author Response · Authors · 2020-11-24
**Summary of additional experiments and rebuttal discussion**

We would like to thank again all reviewers for their valuable inputs and positive feedback, summarizing our paper as “very topical” (R1), “full of convincing arguments” (R2), containing “solid experiments and well-presented results” (R3) and “well written and easy to read” (R4). Moreover, their suggestions have led to a number of exciting new results and updates:

- L2 transition prior experiment (R1, R3): With this rotationally symmetric prior the model becomes non-identifiable and performs worse. Thus, as predicted by our theory, we need a sparse transition prior to achieve good disentanglement.

- Natural dependencies experiment (R3): We have added a visualization that shows the complex dependencies between natural factors and demonstrated that breaking these dependencies (by shuffling) has no clear effect on our model performance.

- Delta t experiments (R4): We have clarified the effect of temporal separation in KITTI Masks and demonstrated that sampling pairs with larger temporal separation increases the model’s disentanglement performance.

- Improved proof intuition (R1, R2): Thanks to the reviewer’s feedback we have managed to make the proof sketch and intuition more accessible to a wide audience, which is very important to us.

Furthermore, we have retrained our models to include static transitions (R3), which did not significantly influence our results; demonstrated that our datasets are not "cheating", as multiple factors change in most transitions (R3); added clarifications about the LAP dataset and the limitations of current metrics (R4); and clarified the significance of our work (R2). Finally, we made several modifications throughout the paper and appendix to improve the clarity of our arguments based on points of discussion in these reviews.

---

### Decision · Program_Chairs · 2021-01-07
**Final Decision**

**Decision:**

Accept (Oral)

**Comment:**

This paper proposes a model for learning disentangled representations by assuming the slowness prior over transitions between two frames. The model is well justified theoretically, and evaluated extensively experimentally. The results are good, and all reviewers agree that this paper is among the top papers they have reviewed. For this reason, I am pleased to recommend this paper for an Oral.